# Psilocybin desynchronizes the human brain

Joshua S. Siegel[1✉], Subha Subramanian[2], Demetrius Perry[1], Benjamin P. Kay[3], Evan M. Gordon[4], Timothy O. Laumann[1], T. Rick Reneau[4], Nicholas V. Metcalf[3], Ravi V. Chacko[5], Caterina Gratton[6], Christine Horan[7], Samuel R. Krimmel[3], Joshua S. Shimony[4], Julie A. Schweiger[1], Dean F. Wong[4], David A. Bender[1], Kristen M. Scheidter[3], Forrest I. Whiting[3], Jonah A. Padawer-Curry[8], Russell T. Shinohara[9,10,11], Yong Chen[11], Julia Moser[12,13], Essa Yacoub[14], Steven M. Nelson[12,15], Luca Vizioli[14], Damien A. Fair[12,13,14,15], Eric J. Lenze[1], Robin Carhart-Harris[16,17], Charles L. Raison[18,19], Marcus E. Raichle[3,4,8,20,21], Abraham Z. Snyder[3,4], Ginger E. Nicol[1,23] & Nico U. F. Dosenbach[3,4,8,20,22,23]

A single dose of psilocybin, a psychedelic that acutely causes distortions of space–time perception and ego dissolution, produces rapid and persistent therapeutic effects in human clinical trials[1–4]. In animal models, psilocybin induces neuroplasticity in cortex and hippocampus[5–8]. It remains unclear how human brain network changes relate to subjective and lasting effects of psychedelics. Here we tracked individual-specific brain changes with longitudinal precision functional mapping (roughly 18 magnetic resonance imaging visits per participant). Healthy adults were tracked before, during and for 3 weeks after high-dose psilocybin (25 mg) and methylphenidate (40 mg), and brought back for an additional psilocybin dose 6–12 months later. Psilocybin massively disrupted functional connectivity (FC) in cortex and subcortex, acutely causing more than threefold greater change than methylphenidate. These FC changes were driven by brain desynchronization across spatial scales (areal, global), which dissolved network distinctions by reducing correlations within and anticorrelations between networks. Psilocybin-driven FC changes were strongest in the default mode network, which is connected to the anterior hippocampus and is thought to create our sense of space, time and self. Individual differences in FC changes were strongly linked to the subjective psychedelic experience. Performing a perceptual task reduced psilocybin-driven FC changes. Psilocybin caused persistent decrease in FC between the anterior hippocampus and default mode network, lasting for weeks. Persistent reduction of hippocampal-default mode network connectivity may represent a neuroanatomical and mechanistic correlate of the proplasticity and therapeutic effects of psychedelics.

Psychedelic drugs can reliably induce powerful acute changes in the perception of self, time and space by agonism of the serotonin 2A (5-HT$_{2A}$) receptor[9,10]. In clinical trials, a single high dose of psilocybin (25 mg) has demonstrated rapid and sustained symptom relief in depression[1–3,11–14], addiction[4,15] and end-of-life anxiety[13,14]. Together, these observations indicate that psychedelics should induce potent acute (lasting roughly 6 hours) and persistent (24 hours to 21 days) neurobiological changes.

In rodent models, transient activation of the 5-HT$_{2A}$ receptors by a psychedelic can alter neuronal communication in 5-HT$_{2A}$-rich regions (for example, the medial frontal lobe) and induce persistent plasticity-related phenomena[5–7]. Synaptogenesis in the medial frontal lobe and anterior hippocampus is thought to be key to the neurotrophic antidepressant effects of psilocybin[5,16,17]. Yet, inherent limitations of rodent models, and imperfect homology to the human 5-HT$_{2A}$ receptor[18], limit the generalizability of these assertions.

Understanding the effects of psychedelics on human brain networks is critical to unlocking their therapeutic mechanisms. In humans, during the roughly 6 hour duration of action, psilocybin increases glutamate signalling and glucose metabolism[19–21], broadly decreases the power of electrophysiological signals[22], reduces hemodynamic fluctuations[23] and decreases segregation between functional networks[24]. The drivers

[1]Department of Psychiatry, Washington University School of Medicine, St Louis, MO, USA. [2]Department of Psychiatry, Beth Israel Deaconess Medical Center, Boston, MA, USA. [3]Department of Neurology, Washington University School of Medicine, St Louis, MO, USA. [4]Mallinckrodt Institute of Radiology, Washington University School of Medicine, St Louis, MO, USA. [5]Department of Emergency Medicine, Advocate Christ Health Care, Oak Lawn, IL, USA. [6]Department of Psychology, Florida State University, Tallahassee, FL, USA. [7]Miami VA Medical Center, Miami, FL, USA. [8]Department of Biomedical Engineering, Washington University in St Louis, St Louis, MO, USA. [9]Center for Biomedical Image Computing and Analytics, University of Pennsylvania, Philadelphia, PA, USA. [10]Penn Statistics in Imaging and Visualization Endeavor, Perelman School of Medicine, University of Pennsylvania, Philadelphia, PA, USA. [11]Department of Biostatistics, Epidemiology and Informatics, Perelman School of Medicine, University of Pennsylvania, Philadelphia, PA, USA. [12]Masonic Institute for the Developing Brain, University of Minnesota, Minneapolis, MN, USA. [13]Institute of Child Development, University of Minnesota, Minneapolis, MN, USA. [14]Center for Magnetic Resonance Research (CMRR), University of Minnesota, Minneapolis, MN, USA. [15]Department of Pediatrics, University of Minnesota, Minneapolis, MN, USA. [16]Department of Neurology, University of California, San Francisco, CA, USA. [17]Centre for Psychedelic Research, Imperial College London, London, UK. [18]Usona Institute, Fitchburg, WI, USA. [19]Department of Psychiatry, University of Wisconsin School of Medicine & Public Health, Madison, WI, USA. [20]Department of Psychological and Brain Sciences, Washington University in St Louis, St Louis, MO, USA. [21]Department of Neuroscience, Washington University School of Medicine, St Louis, MO, USA. [22]Department of Pediatrics, Washington University School of Medicine, St Louis, MO, USA. [23]These authors contributed equally: Ginger E. Nicol, Nico U. F. Dosenbach. ✉e-mail: jssiegel@wustl.edu

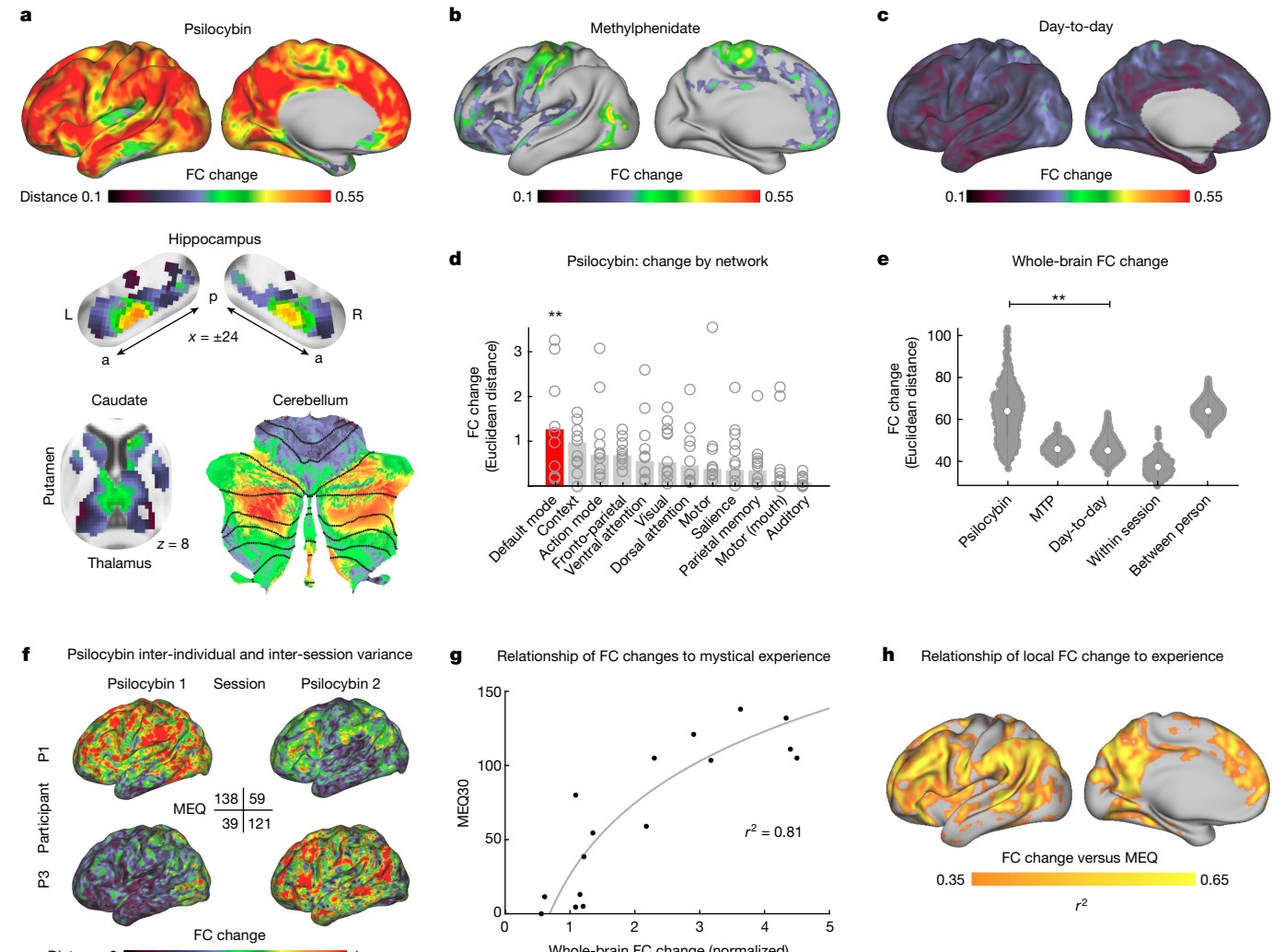

**Fig. 1 | Acute psilocybin effects on functional brain organization.** FC change (Euclidean distance) was calculated across the cortex and subcortical structures. Effects of drug condition were tested with an LME model in $n = 6$ longitudinally sampled participants over ten sessions with psilocybin and six sessions with methylphenidate (MTP) (**a** and **b** are thresholded at $P < 0.05$ based on permutation testing with TFCE; see unthresholded statistical maps in Extended Data Fig. 2). **a**, Psilocybin-associated FC change, including in subcortex. a, anterior; p, posterior; L, left; R, right. **b**, MTP-associated FC change. **c**, Typical day-to-day variability as a control to the drug conditions (unthresholded: not included in LME model). **d**, Average FC change in individual-defined networks. Open circles represent individual participants. FC change is larger in DMN than other networks. Rotation-based null model (spin test[62,97]): ten psilocybin doses, 1,000 permutations, one-sided $P_{spin} < 0.001$, ($P_{spin} > 0.05$ for all other networks). **P < 0.001, uncorrected. **e**, Whole-brain FC change (Euclidean distance from baseline) for all rest scans across conditions. FC change for MTP, psilocybin and day-to-day are in comparison to same-participant baseline. White dots indicate median, vertical lines indicate quartiles. LME model predicting whole-brain FC change: ten psilocybin doses (275 observations), estimate (95% CI) = 15.83 (13.50, 18.15), $t_{(266)} = 13.39$, $P = 1.36 \times 10^{-31}$, uncorrected. For the full FC distance matrix with session labels, see Extended Data Fig. 3. **f,g**, Comparison of the differences in FC change to differences in psychedelic experiences. **f**, Individual FC change maps and MEQ30 scores for two exemplars (see Extended Data Fig. 4 for all drug sessions). **g**, The relationship between whole-brain FC change and mystical experience rating is plotted for all drug sessions (psilocybin and MTP). The LME model demonstrated a significant relationship: 16 drug doses (ten psilocybin, six MTP), estimate (95% CI) = 69.78 (50.15, 89.41), $t_{(13)} = 7.68$, $P = 3.5 \times 10^{-6}$, uncorrected. **h**, The relationship between FC change and MEQ30 ($r^2$) is mapped across the cortical surface.

of these acute changes are poorly understood, particularly in the subcortex. Preliminary efforts to identify network changes in the weeks after psilocybin have yielded mixed results[25–27]. Persistent effects of psilocybin on clinically relevant circuits have yet to be characterized in humans.

The ventromedial prefrontal cortex and anterior and middle hippocampus are functionally connected to the default mode network (DMN)[28,29]. Increased FC between the hippocampus and DMN has been associated with depression symptoms[30] and decreased FC is associated with treatment[31,32]. These 5-HT$_{2A}$ receptor rich[33] and depression associated default mode regions[34–36] are candidates for mediating the neurotrophic antidepressant effects of psychedelics.

Precision functional mapping uses dense repeated functional magnetic resonance imaging (fMRI) sampling[37–41] to reveal the time course of individual-specific intervention-driven brain changes[42]. This approach accounts for inter-individual variability in brain networks[37] and capitalizes on the high stability of networks within individuals from day to day[38]. Using precision functional mapping, we observed individual-specific acute and persistent brain changes following a single high dose of psilocybin.

Healthy young adults received 25 mg psilocybin and 40 mg methylphenidate (MTP, generic name Ritalin, dose-matched for arousal effects) 1–2 weeks apart and underwent regular MRI sessions (roughly 18 per participant) before, during, between and after the two drug

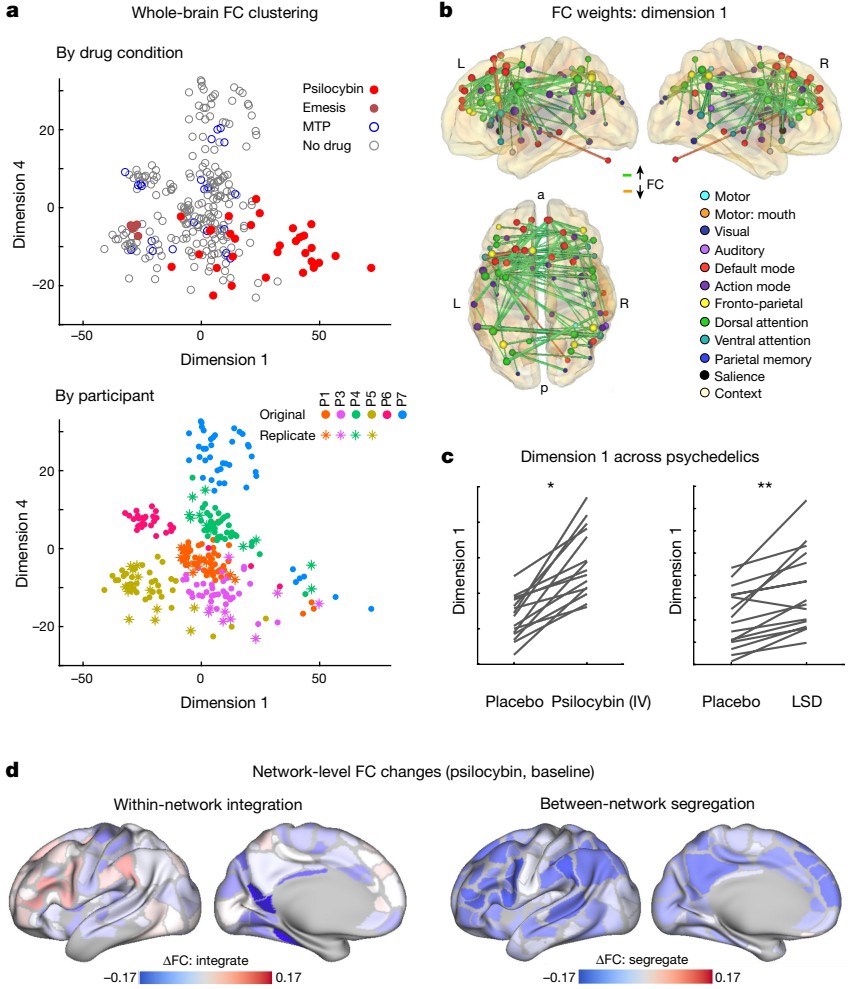

**Fig. 2 | Data-driven clustering of brain network variability.** MDS blind to session labels was used to assess brain changes across conditions. **a**, In the scatter plots, each point represents whole-brain FC from a single 15 min scan, plotted in a multidimensional space on the basis of the similarity between scans. Dimensions 1 and 4 showed strong effects of psilocybin. The top shows scans coded on the basis of drug condition. Dark red denotes that the participant had an episode of emesis shortly after taking psilocybin. The bottom shows scans coloured on the basis of participant identity. Dimension 1 separates psilocybin from non-drug and MTP scans in most cases. See Extended Data Fig. 5 for the dimension 1–4 weight matrices. **b**, Visualization of dimension 1 weights. The top 1% of edges (connections) are projected onto the brain (green indicates connections that are increased by psilocybin). Cerebellar connections are included although the structure is not shown. **c**, Re-analysis of dimension 1 in

extant datasets with intravenous psilocybin (left, ref. 55, paired two-sided *t*-test of change in dimension 1 score, $n = 9$, $t_{(8)} = 2.97$, $P = 0.0177$, uncorrected) and LSD (right, ref. 56, paired two-sided *t*-test: $n = 16$, $t_{(15)} = 4.58$, $P = 3.63 \times 10^{-4}$, uncorrected). *$P < 0.05$, **$P < 0.001$, uncorrected. **d**, Average effects of psilocybin on network FC, shown separately for within-network integration (left) and between network segregation (right). For network integration (left), blue indicates a loss of FC (correlations) between regions within the same network. For network segregation (right), blue indicates a loss of FC (anticorrelations) to all other regions in different networks; see Extended Data Fig. 6 for a full correlation matrix. Dissolution of functional brain organization corresponds to decreased within-network integration and decreased between network segregation.

doses (Extended Data Fig. 1, Supplementary Table 1 and Supplementary Video 1: data quality metrics for 129 total MRI visits). Dense predrug sampling familiarized participants with the scanner and established baseline variability.

## Psilocybin disrupts brain connectivity

Psilocybin acutely caused profound and widespread brain FC changes (Fig. 1a) across most of the cerebral cortex ($P < 0.05$ based on two-sided linear mixed-effects (LME) model and permutation testing), but most prominent in association networks (FC change mean (standard deviation, s.d.): association cortex 0.44 (0.03), primary cortex 0.36 (0.05)). In the subcortex the largest psilocybin-associated FC changes were seen in DMN connected parts of the thalamus, basal ganglia, cerebellum and hippocampus[29,43,44] (Fig. 1a and Extended Data Fig. 2). In the

hippocampus, foci of strong FC disruption were located in the anterior hippocampus (Montreal Neurological Institute coordinates −24, −22, −16 and 24, −18, −16). Other large FC disruptions were seen in medi-odorsal and paraventricular thalamus[45] and anteromedial caudate. In the cerebellum, the largest FC changes were seen in the DMN connected areas[44] (Fig. 1a).

By comparison, MTP-associated FC changes localized to sensorimotor systems (Fig. 1b and Extended Data Fig. 2) and paralleled the map of day-to-day variability (Fig. 1c) probably due to arousal effects[39]. Psilocybin-associated FC change was largest in the DMN (Fig. 1d and Supplementary Fig. 1; averaged across all psilocybin sessions; spin test, 1,000 permutations, one-sided $P_{spin} < 0.001$; $P_{spin} > 0.05$ for all other networks). However, MTP-associated FC change was largest in motor and action networks ($P_{spin} = 0.002$; $P_{spin} > 0.05$ for all other networks; Supplementary Fig. 1b).

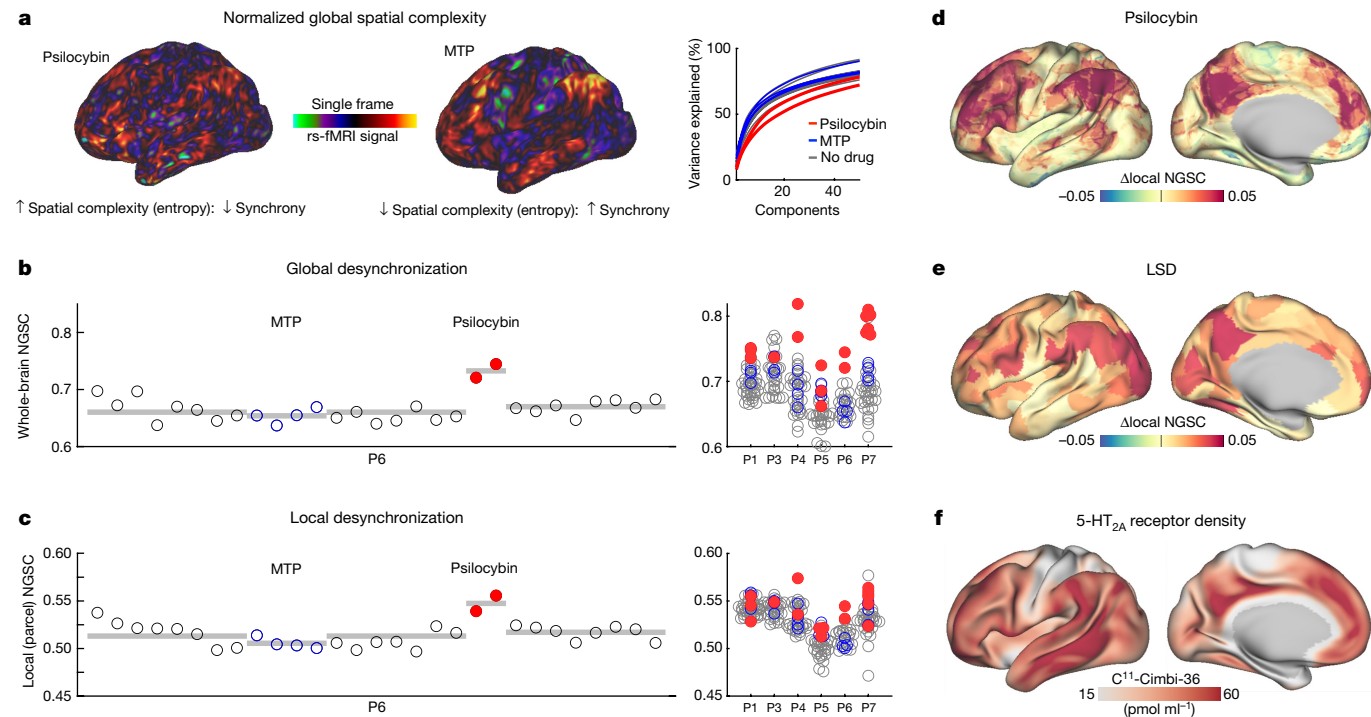

**Fig. 3 | Spatial desynchronization of cortical activity during psilocybin.**
**a**, NGSC captures the complexity of brain activity patterns. It is derived from the number of spatial principal components needed to explain the underlying structure. Higher entropy equals desynchronized activity. On the right is variance explained by subsequent principal components for psilocybin in red, MTP in blue and no drug in grey for P6. **b**, Whole-brain entropy (NGSC) is shown for every fMRI scan for a single participant (P6). At right, increases during psilocybin were present in all participants. Sample sizes are provided in Supplementary Table 1. Grey bars indicate condition means. **c**, Parcel entropy

(computed on individual-specific parcels) within functional brain areas shows similar psilocybin-driven increases as whole-brain entropy. **d**, Psilocybin-associated spatial entropy (individual-specific parcels, averaged across participants) is visualized on the cortical surface. Psilocybin-associated increases in entropy were largest in association cortex. **e**, LSD-associated increases in spatial entropy were similar to those induced by psilocybin (using data from ref. 56). **f**, Increases corresponded spatially to 5-HT$_{2A}$ receptor density[33]. In **b**–**d**, $n$ = 6 participants, 272 observations (scans). For **e**, $n$ = 16 participants.

Despite MTP and psilocybin causing similar increases in heart rate (Supplementary Fig. 2), the effects of psilocybin on FC were more than threefold larger than the effects of MTP (Fig. 1e; post hoc two-sided $t$-test; $P = 3.6 \times 10^{-6}$, uncorrected). The psilocybin effects also dwarfed those of other control conditions (Fig. 1e; day-to-day change (normalized) 1; task 1.22, MTP 1.10, high head motion 1.29, psilocybin 3.52, between person 3.53; Extended Data Fig. 3; these effects were robust to preprocessing choices: Supplementary Figs. 3 and 4). To put the effects of psilocybin into perspective, it helps to consider that the mean changes in brain organization caused by the drug were as large as the differences in brain organization between different people (Fig. 1e).

## The psychedelic experience

The large amount of data collected per participant, under the individual-specific imaging model, allowed us to move beyond group-analyses and compare the subjective psychedelic experience (30-item Mystical Experience Questionnaire, MEQ30)[46] to brain function data session-by-session (Fig. 1f). The MEQ30 is a self-assessment instrument used to measure the intensity and quality of mystical experiences, including feelings of connectedness, transcendence of time and space, and a sense of awe, with a maximum score of 150 (ref. 46). Across psilocybin sessions and participants, FC change tracked with the intensity of the subjective experience (Fig. 1f and Extended Data Fig. 4). Correlating the whole-brain FC change ($x$ axis) against the MEQ30 scores ($y$ axis) for all drug sessions (Fig. 1g) revealed an $r^2 = 0.81$ (LME model predicting MEQ30 score: effect of FC change, $t_{(13)} = 7.68$; $P = 3.5 \times 10^{-6}$, uncorrected). Head motion was not significantly correlated with MEQ30 scores (effect of framewise displacement, $t_{(13)} = -1.26$,

$P = 0.23$, uncorrected). Projecting the relationship between someone's mystical experience and the corresponding FC change onto the brain (Fig. 1h, vertex-wise) showed it to be driven by association cortex, relatively sparing primary motor and sensory regions. Of the four MEQ30 dimensions (mystical, positive mood, transcendence of time and space, and ineffability), the one most strongly correlating with brain change was transcendence (for example, 'loss of your usual sense of time or space', $r^2 = 0.86$, Supplementary Fig. 5), however, all dimensions were highly correlated ($r > 0.8$). Repeated sampling enabled us to determine that the inter-individual variability in the effects of psilocybin in the brain was more likely related to differences in drug effects than measurement error (likelihood ratio test of participant-specific response to psilocybin, $P = 0.00245$, uncorrected)[47,48].

## The psychedelic dimension

To examine the latent dimensions of brain network changes we performed multidimensional scaling (MDS) on the parcellated FC matrices from every fMRI scan[38]. MDS is blind to session labels (for example, drug, participant). Yet, dimension 1, which explained the largest amount of variability, separated psilocybin from other scans (Fig. 2a), apart from one session during which the participant (P5R) had emesis 30 minutes after taking psilocybin (dark red dots on the left of Fig. 2a). The higher score on dimension 1 associated with psilocybin, corresponded to reduced segregation between the DMN and other networks (fronto-parietal[49], dorsal attention[50], salience[51] and action-mode[52,53]) that are typically anticorrelated with it[54] (Fig. 2b and Extended Data Fig. 5). To determine whether this reflects a common effect of psilocybin that generalizes across datasets and psychedelics,

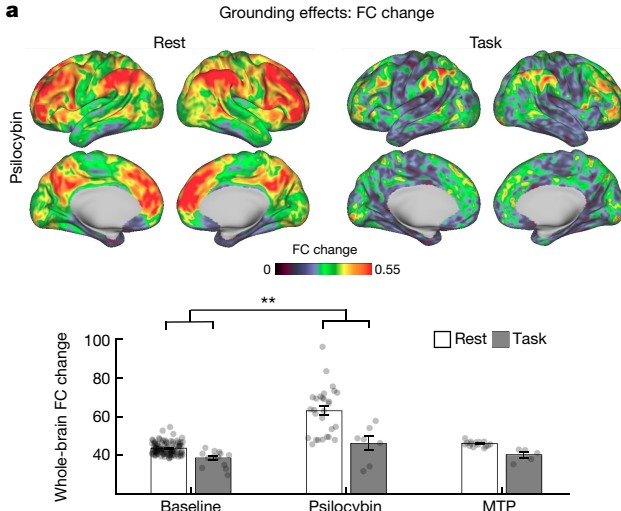

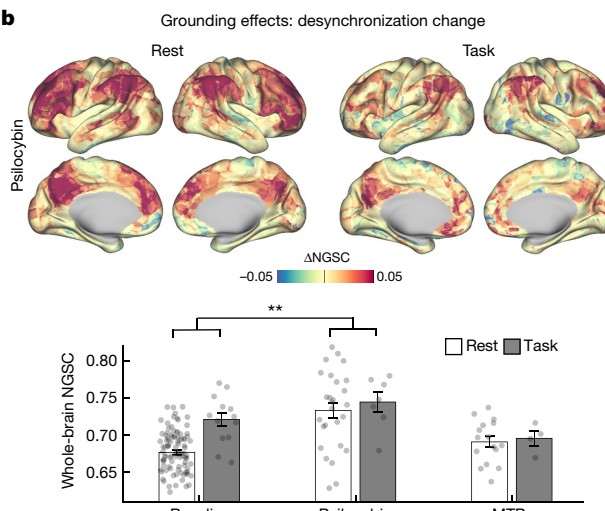

**Fig. 4 | Effects of perceptual task performance on psilocybin-associated FC change and desynchronization. a**, Psilocybin-associated FC change from resting scans (left) and from task scans (right). **b**, Regional NGSC change (psilocybin minus baseline) from rest scans (left) and from task scans (right). Bar graphs on the bottom indicate the corresponding whole-brain FC change (**a**) and whole-brain NGSC values (**b**) during rest and task for baseline and drug conditions. LME models indicated an interaction of task × psilocybin on FC change ($n = 7$ with task data on psilocybin, estimate (95% CI) = −6.48 (−9.59, −3.37), $t_{(265)} = -6.48$, $P = 5.49 \times 10^{-5}$, uncorrected) and an interaction of task × psilocybin on NGSC ($n = 7$ with task data on psilocybin, estimate (95% CI) = −0.042 (−0.056, −0.027), $t_{(265)} = -5.62$, $P = 4.82 \times 10^{-8}$, uncorrected). Bars indicate mean and error bars indicate s.e.m.. \*\*$P < 0.001$, uncorrected.

we calculated dimension 1 scores for extant datasets from participants receiving intravenous (i.v.) psilocybin[55] and lysergic acid diethylamide (LSD)[56]. Psychedelic treatment increased dimension 1 in nearly every participant in the psilocybin and LSD datasets (Fig. 2c), suggesting that this is a common effect across psychedelic drugs and individuals.

Subtraction of average FC (psilocybin minus baseline) revealed a pattern of FC change similar to dimension 1 (Fig. 2d and Extended Data Figs. 5 and 6). Consistent with previous psychedelics studies[24], psilocybin increased FC between networks (particularly fronto-parietal, default mode and dorsal attention), whereas FC within networks was relatively less affected. A similar pattern of loss of segregation between brain networks is produced by nitrous oxide and ketamine[57], suggesting that the psychedelic dimension observed here may generalize to psychedelic-like dissociative drugs.

By comparison, MTP decreased within-network FC in the sensory, motor and auditory regions (Extended Data Fig. 6), consistent with previous reports[58] and similar to the effects of caffeine[39]. To verify that observations in our sample ($n = 6$) were generalizable, we compared stimulant effects in our study to those in the Adolescent Brain Cognitive Development (ABCD) Study[59] ($n = 487$ taking stimulants). The effect of stimulant use in ABCD was consistent with MTP-associated FC changes in our dataset (Extended Data Fig. 6).

## Desynchronization explains FC change

Multi-unit recording studies suggest that agonism of 5-HT$_{2A}$ receptors by psychedelics desynchronizes populations of neurons that typically co-activate[60]. We proposed that this phenomenon, observed at a larger spatial scale, might account for psilocybin-associated FC change (Fig. 1). We observed that the typically stable spatial structure of resting fMRI fluctuations was disrupted and desynchronized by psilocybin (Supplementary Videos 2–7: brain activity time courses during drug sessions for each participant). Therefore, we quantified brain signal synchrony using normalized global spatial complexity (NGSC): a measure of spatial entropy that is independent of the number of signals[61]. NGSC calculates cumulative variance explained by subsequent spatiotemporal patterns (Fig. 3a). The lowest value of NGSC (0) means that the time course for every vertex and/or voxel is identical. The highest

value of NGSC (equal to one) means that the time course for every vertex and/or voxel is independent, indicating maximal desynchronization (or spatial entropy).

Psilocybin significantly increased NGSC acutely with values returning to predrug baseline by the following session (Fig. 3b,c). The increase in NGSC was observed at the whole-brain level (Fig. 3b; LME model, estimate (95% confidence interval (CI)) = 0.0510 (0.0343, 0.0676), $t_{(265)} = 6.8$, $P = 2.0 \times 10^{-6}$, uncorrected) and correlated with the subjective experience (MEQ30: Extended Data Fig. 7; $r = 0.80$, $P = 3.52 \times 10^{-4}$, uncorrected, after single outlier removal), whereas nuisance variables did not. Increased NGSC was also observed for individual-defined brain areas[62] (Fig. 3c; LME model, estimate (95% CI) = 0.0149 (0.0071, 0.0228), $t_{(265)} = 3.74$, $P = 2.30 \times 10^{-4}$, uncorrected), with the largest increases in association cortex and minimal changes in primary cortex (Fig. 3d). Global and local desynchronization replicated in an LSD dataset[56] (Fig. 3e) and the distribution of these effects correlated with 5-HT$_{2A}$ receptor density (Fig. 3f; bivariate correlation NGSC$_{psilocybin}$ to Cimbi-36 binding, $r = 0.39$, $P = 1.9 \times 10^{-13}$; NGSC$_{LSD}$ to Cimbi-36 binding, $r = 0.32$, $P = 4.5 \times 10^{-9}$, uncorrected)[33,63].

## Task engagement reduces desynchronization

To investigate how psilocybin-driven brain changes are influenced by task states, participants were asked to complete a simple auditory–visual matching task in the scanner (Methods, perceptual fMRI task). Participants performed this task with more than 80% accuracy during drug sessions (Extended Data Fig. 8a–c). Engagement in the task significantly decreased the magnitude of psilocybin-associated network disruption and desynchronization (Fig. 4; LME model interaction of task × psilocybin: FC change $P = 5.49 \times 10^{-5}$, NGSC $P = 4.82 \times 10^{-8}$, uncorrected). These results were robust to scan order effects (Supplementary Fig. 6) and regression of evoked responses (Supplementary Fig. 7).

The reduction of psilocybin-driven brain changes during task performance seems to parallel the psychological principle of 'grounding': directing one's attention externally as a means of alleviating intense or distressing thoughts or emotions. Grounding techniques are commonly used in psychedelic-associated psychotherapy to lessen overwhelming or distressing effects of psilocybin[64]. Task-related reductions in network desynchronization provide strong evidence for context-dependent

effects of psilocybin on brain activity and FC[65] and fill an important gap between preclinical studies of context dependence[66,67] and clinical observations[68].

Classical animal studies documented that psychedelics reduce optic tract responses to photic stimulation of the retina, indirectly reducing visual cortex activation[69,70]. We replicated these effects by documenting reduced task-evoked responses in primary visual cortex (Extended Data Fig. 8d–g). To assess whether psilocybin affects the magnitude of hemodynamic responses elsewhere, we analysed evoked responses during the perceptual task in other task-related regions of interest (Extended Data Fig. 8f,g). But the magnitudes of other evoked responses were not significantly changed by psilocybin (two-way analysis of variance of drug and participant; effect of drug: left V1 $P = 0.03$, right V1 $P = 0.02$, all other $P > 0.1$, uncorrected).

## Persistent decrease in hippocampal FC

To assess whether persistent neurotrophic and psychological effects of psychedelics might be associated with persistent FC changes after psilocybin, we compared FC changes 1–21 days post-psilocybin to pre-psilocybin. Whole-brain FC change scores were small (normalized FC change (range) of 1.05 (0.94, 1.27)), indicating that the brain's network structure had mostly returned to baseline (Extended Data Fig. 2).

Atypical cortico-hippocampal connectivity has been associated with affective symptoms[30] and hippocampus neurogenesis is observed after psilocybin[6]. Further, acute decreases in hippocampal glutamate after psilocybin correlate with decreased DMN connectivity and ego dissolution[21]. Thus, we investigated whether the same region of the anterior hippocampus that showed strong acute FC change also showed persistent FC change. We observed significant FC change in the 3 week post-drug period (Fig. 5a,b; LME mean change 0.095, $P_{pre–post-psilocybin} = 0.0033$, uncorrected). No persistent FC differences were observed post-MTP (Methods, section 'Persistent effects analysis'; LME 'FC change' 90% CI (−0.056, 0.080); equivalence $\delta = \pm 0.086$, $P_{pre–post-MTP} = 0.77$).

FC between the anterior hippocampus and DMN was decreased post-psilocybin (Fig. 5c,d). Time-course visualization, after aligning them so that psilocybin dose was day 0, suggests that connectivity is reduced for 3 weeks following a single psilocybin dose (Fig. 5d; AntHip-DMN FC mean (95% CI): pre-psilocybin was 0.180 (0.169, 0.192); post-psilocybin was 0.163 (0.150, 0.176)). AntHip-DMN FC values returned to pre-psilocybin baseline by the replication visit 6–12 months later, however, the smaller replication sample ($n = 4$ with one pre-psilocybin visit each) was not statistically powered to detect small changes. This observation is compelling, as it localized to the anterior hippocampus, a brain region showing substantial synaptogenesis following psilocybin[6]. Reduced hippocampal-cortical FC may reflect increased plasticity of self-oriented hippocampal circuits[31] (Fig. 5e).

## From micro- to macro-scale psychedelic effects

The synchronized patterns of cofluctuations during the resting state are believed to reflect the brain's perpetual task of modelling reality[71]. It follows that the stability of functional network organization across day, task, MTP and arousal levels (but not between individuals), reflects the subjective stability of waking consciousness. By contrast, the much larger changes induced by psilocybin fit with participants' subjective reports of a radical change in consciousness. The large magnitude of effects of psilocybin, in comparison to the effects of MTP, suggests that observed changes are not merely due to increased arousal or non-specific effects of monoaminergic stimulation[72].

Our observation that psychedelics desynchronize brain activity regionally and globally provides a bridge between previous findings at the micro- and macro-scales of neuroscience. Multi-unit recording studies suggest that agonism of 5-HT$_{2A}$ receptors by psychedelics does

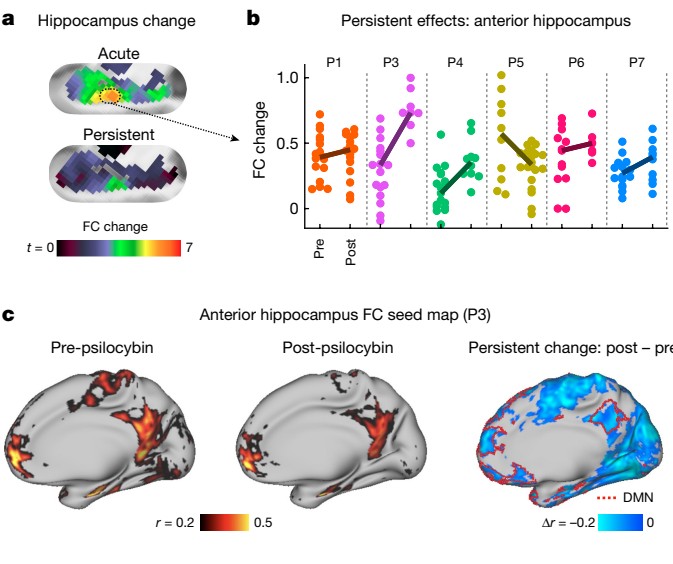

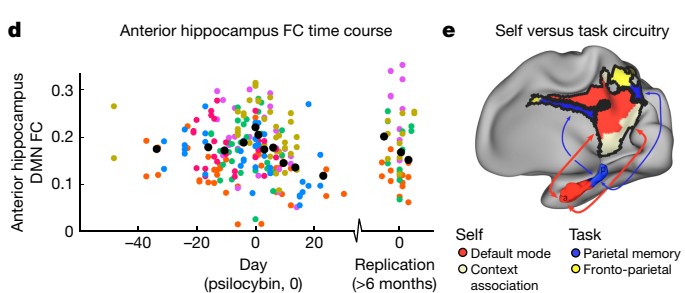

**Fig. 5 | Persistent effects of psilocybin on hippocampal-cortical FC.**
**a**, Hippocampus FC change maps (left hippocampus; unthresholded $t$-maps, as in Extended Data Fig. 2). Acute psilocybin FC change is shown on top and persistent FC change (3 weeks after psilocybin) on the bottom. **b**, Each dot represents the FC change score for the anterior hippocampus for a single scan before (left) and after (right) psilocybin for every participant (coloured as in Fig. 2). Participants showed a post-psilocybin increase in FC change in the anterior hippocampus (LME model, pre- versus post-psilocybin; $n = 6$ participants, 186 observations, estimate (95% CI) = 0.095 (0.032, 0.168), $t_{(182)} = 2.97$, $P = 0.0033$, uncorrected). **c**, Connectivity from an anterior hippocampus seed (Montreal Neurological Institute coordinates −24, −22, −16 and 24, −18, −16) pre-psilocybin (left), post-psilocybin (middle) and persistent change (post- minus pre-) for an exemplar participant (P3). The red border on the right-most brain outlines the individual-specific DMN. A decrease in hippocampal FC with parietal and frontal components of the DMN is seen. **d**, Time course of anterior hippocampus minus DMN for all participants and scans (participant colours as in **b**). A moving average is shown in black. **e**, Schematic of hippocampal-cortical circuits, reproduced from ref. 29, CC BY 4.0.

not uniformly increase or decrease firing of pyramidal neurons, but rather serves to desynchronize pairs or populations of neurons that co-activate under typical conditions[60]. Meanwhile, previous resting fMRI studies have reported a range of acute changes following ingestion of psilocybin[55,63], ayahuasca[73] and LSD[56,74], which broadly converge on a loss of network connectivity and an increase in global integration[24,75]. Disruption of synchronized activity at several scales may explain the paradoxical observation that psychedelics produce an increase in metabolic activity[19,20], a decrease in the power of local fluctuations[22,76] and a loss of the brain's segregated network structure[23,56]. This desynchronization of neural activity has been described as an increase in entropy or randomness of brain activity in the psychedelic state[77,78]. Our results support the hypothesis that these changes underpin the cognitive and perceptual changes associated with psychedelics.

## Desynchrony may drive persistent change

The dramatic departure from typical synchronized patterns of co-activity may be key to understanding the acute effects of psilocybin and also its persistent neurotrophic effects. Changes in resting activity are linked to shifts in glutamate-dependent signalling during psilocybin exposure[21,79,80]. This phenomenon, shared by ketamine and psychedelics, engages homeostatic plasticity mechanisms[81,82], a neurobiological response to large deviations in typical network activity patterns[83–85]. This response to novelty includes rapid upregulation in expression of *BDNF*, *MTOR*, *EEF2* and other plasticity-related immediate early genes[8,80], which are thought to have a key role in the antidepressant response[86]. Consistent with this notion, psilocybin produced the largest changes in the DMN, frequently associated with neuropsychiatric disorders[34,35,87–91], and in a region of the anterior and middle hippocampus associated with the self[29,92] and the present moment[93].

Psychedelics rapidly induce synaptogenesis in the hippocampus and cortex, effects that seem to be necessary for rapid antidepressant-like effects in animal models[7,17]. However, understanding the underpinnings of the behavioural effects of psychedelics requires human studies. Advances in precision functional mapping[37,94,95] and individual-level characterization enabled us to identify desynchronization of resting-state fMRI signals, connect these changes with subjective psychedelic effects and localize these changes to depression-relevant circuits (DMN, hippocampus). These analyses rely on precise characterization of an individuals' baseline brain organization (for example, individual definition of brain areas, networks and day-to-day variability) to understand how that organization is altered by an intervention. This precision drug mechanism study was conducted in non-depressed volunteers. Verification of the proposed antidepressant mechanism of psilocybin will require precision patient studies. New methods to measure neurotrophic markers in the human brain[96] will provide a critical link between mechanistic observations at the cellular, brain networks and psychological levels.

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

## Methods

### Regulatory approvals and registrations

Written informed consent was obtained from all participants in accordance with the Declaration of Helsinki and procedures established by the Washington University in Saint Louis Institutional Review Board. All participants were compensated for their time. All aspects of this study were approved by the Washington University School of Medicine (WUSOM) Internal Review Board, the Washington University Human Research Protection Office (WU HRPO), the Federal Drug Administration (IND no. 202002165) and the Missouri Drug Enforcement Agency (DEA) under a federal DEA schedule 1 research licence and registered with ClinicalTrials.gov identifier NCT04501653. Psilocybin was supplied by Usona Institute through Almac Clinical Services.

### Study design

Healthy young adults ($n = 7$, 18–45 years) were enrolled between April 2021 and March 2023 in a randomized cross-over precision functional brain mapping study at Washington University in Saint Louis (see Supplementary Methods for inclusion and exclusion criteria). The purpose of the study was to evaluate differences in individual-level connectomics before, during and after psilocybin exposure. Participants underwent imaging during drug sessions (with MRI starting 1 h after drug ingestion) with 25 mg psilocybin or 40 mg MTP, as well as non-drug imaging sessions. Drug condition categories were (1) baseline, (2) drug 1 (MTP or psilocybin), (3) between, (4) drug 2 and (5) after. Randomization allocation was conducted using REDCap and generated by team members who prepared study materials including drug or placebo but otherwise had no contact with participants. A minimum of three non-drug imaging sessions were completed during each non-drug window: baseline, between and after drug sessions. The number of non-drug MRI sessions was dependent on availability of the participant, scanner and scanner support staff. Dosing day imaging sessions were conducted 60–180 min following drug administration during peak blood concentration[98]. One participant (P2) was not able tolerate fMRI while on psilocybin, and had trouble staying awake on numerous fMRI visits after psilocybin and was thus excluded from analysis (except for data quality metrics in Extended Data Fig. 1).

MTP was selected as the active control condition to simulate the cardiovascular effects and physiological arousal (that is, controlling for dopaminergic effects) associated with psilocybin[99]. Usona Institute, a US non-profit medical research organization, provided good manufacturing practices for psilocybin.

Drug sessions were facilitated by two clinical research staff who completed an approved in-person or online facilitator training programme provided by Usona Institute, as part of the phase 2 study (ClinicalTrials.gov identifier NCT03866174). The role of the study facilitators was to build a therapeutic alliance with the participant throughout the study, prepare them for their drug dosing days and to observe and maintain participant safety during dosing day visits[64]. The pair consisted of an experienced clinician (lead clinical facilitator) and a trainee (cofacilitator).

The predefined primary outcome measure was precision functional mapping (numerous visits, very long scans to produce individual connectomes) examining the effects of psilocybin on cortical and cortico-subcortical brain networks that could explain its rapid and sustained behavioural effects. Predefined secondary outcome measures included (1) assessment of hemodynamic response to evaluate how 5-HT$_{2A}$ receptor agonism by psychedelics may alter neurovascular coupling, (2) assessment of acute psychological effects of psilocybin using the MEQ30 score (Supplementary Methods) and (3) assessment of personality change using the International Personality Item Pool-Five-Factor Model[100]. Changes in pulse rate and respiratory rate during psilocybin and placebo were later added as secondary outcome measures and personality change was abandoned because it was clear that we would not be powered to detect personality change.

### Replication protocol

Participants were invited to return 6–12 months after completing the initial cross-over study for a replication protocol. This included 1–2 baseline fMRIs, a psilocybin session (identical to the initial session, except for lack of blinding) and 1–2 'after' sessions within 4 days of the dose.

### Participants

Healthy adults aged 18–45 years were recruited by campus-wide advertisement and colleague referral. Participants ($n = 7$) were enrolled from March 2021 to May 2023. Participants were required to have had at least one previous lifetime psychedelic exposure (for example, psilocybin, mescaline, ayahuasca, LSD), but no psychedelics exposure within the past 6 months. Individuals with psychiatric illness (depression, psychosis or addiction) based on the DSM-5 were excluded. Demographics and data summary details are provided in Supplementary Table 1. One of the authors (N.U.F.D.) was a study participant.

### MRI

Participants were scanned roughly every other day over the course of the experiment (Extended Data Fig. 1). Imaging was performed at a consistent time of day to minimize diurnal effects in FC[101]. Neuroimaging was performed on a Siemens Prisma scanner (Siemens) in the neuroimaging laboratories at the Washington University Medical Center.

Structural scans (T1w and T2w) were acquired for each participant at 0.9 mm isotropic resolution, with real-time motion correction. Structural scans from different sessions were averaged together for the purposes of Freesurfer segmentation and nonlinear atlas registrations.

To capture high-resolution images of blood oxygenation level-dependent (BOLD) signal, we used an echo-planar imaging sequence[102] with 2 mm isotropic voxels, multiband 6, multi-echo 5 (times to echo: 14.20, 38.93, 63.66, 88.39, 113.12 ms)[103], repetition or relaxation time: 1,761 ms, flip angle of 68° and in-plane acceleration[104] (IPAT or grappa) of 2. This sequence acquired 72 axial slices (144 mm coverage). Each resting scan included 510 frames (lasting 15 min, 49 s) as well as three frames at the end used to provide estimate electronic noise.

Every session included two 15-min resting-state fMRI (rs-fMRI) scans, during which participants were instructed to hold still and look at a white fixation crosshair presented on a black background. Head motion was tracked in real time using Framewise Integrated Real-time MRI Monitoring software (FIRMM)[105]. An eye-tracking camera (EyeLink) was used to monitor participants for drowsiness.

### Perceptual (matching) fMRI task

Participants also completed a previously validated event-related fMRI task. This was a suprathreshold auditory–visual matching task in which participants were presented with a naturalistic visual image (duration 500 ms) and coincident spoken English phrase, and were asked to respond with a button press to indicate whether the image and phrase were 'congruent' (for example, an image of a beach and the spoken word 'beach') or 'incongruent'. Both accuracy and response time of button presses were recorded. Each trial was followed by a jittered inter-stimulus interval optimized for event-related designs. In a subset of imaging sessions, two task fMRI scans were completed following the two resting scans. Task fMRI scans used the same sequence used in resting fMRI, included 48 trials (24 congruent, 24 incongruent) and lasted a total of 410 s. In analyses, high motion frames were censored[106] and the two task scans were concatenated to better match the length of the rs-fMRI scans. Note the stimulus order in the two trials did not vary across session. The order of rest and task scans was not counterbalanced across sessions to avoid concern that task scans may influence subsequent rest scans.

## Resting fMRI processing and resting-state network definition

Resting fMRI data were preprocessed using an in-house processing pipeline. In brief, this included removal of thermal noise using NORDIC denoising[107–109], correction for slice timing and field distortions, alignment, optimal combination of many echoes by weighted summation[110], normalization, nonlinear registration, bandpass filtering and scrubbing at a movement threshold of 0.3 mm to remove reduce the influence of confounds[111]. Tissue-based regressors were computed in volume (white matter, ventricles, extra-axial cerebrospinal fluid)[112] and applied following projection to surface. Task-based regressors were only applied when indicated. Details on rs-fMRI preprocessing are provided in Supplementary Methods. Visualizations of motion, physiological traces and signal across the brain ('grayplots') before and after processing[113] are provided in Supplementary Video 1.

## Surface generation and brain areal parcellation

Surface generation and processing of functional data followed similar procedures to Glasser et al.[114]. To compare FC and resting-state networks across participants, we used a group-based surface parcellation and community assignments generated previously[62].

For subcortical regions, we used a set of regions of interest[115] generated to achieve full coverage and optimal region homogeneity. A subcortical limbic network was defined on the basis of neuroanatomy: amygdala, anteromedial thalamus, nucleus accumbens, anterior hippocampus and posterior hippocampus[116,117]. These regions were expanded to cover anatomical structures (for example, anterior hippocampus)[31].

To generate region-wise connectivity matrices, time courses of all surface vertices or subcortical voxels within a region were averaged. FC was then computed between each region timeseries using a bivariate correlation and then Fisher z-transformed for group comparison.

## Individualized network and brain area mapping

We identified canonical large-scale networks using the individual-specific network matching approach described previously[43,44,62]. In brief, cortical surface and subcortical volume assignments were derived using the graph-theory-based Infomap algorithm[118]. In this approach, we calculated the correlation matrix from all cortical vertices and subcortical voxels, concatenated across all a participant's scans. Correlations between vertices within 30 mm of each other were set to zero. The Infomap algorithm was applied to each participant's correlation matrix thresholded at a range of edge densities spanning from 0.01 to 2%. At each threshold, the algorithm returned community identities for each vertex and voxel. Communities were labelled by matching them at each threshold to a set of independent group average networks described previously[62]. In each individual and in the average, a 'consensus' network assignment was derived by collapsing assignments across thresholds, giving each node the assignment it had at the sparsest possible threshold at which it was successfully assigned to one of the known group networks. See Extended Data Fig. 4 and Supplementary Fig. 1 for individual and group mode assignments, respectively. The following networks were included: the association networks including the DMN, fronto-parietal, dorsal attention, parietal memory, ventral attention, action-mode, salience and context networks; and the primary networks including the visual, somato-motor, somato-motor face and auditory networks.

To compute local (areal) desynchronization, we also defined brain areas at the individual level using a previously described areal parcellation approach[39]. In brief, for each participant, vertex-wise FC was averaged across all sessions to generate a dense connectome. Then, abrupt transitions in FC values across neighbouring vertices were used to identify boundaries between distinct functional areas.

## LME model

To take advantage of the multilevel precision functional mapping study design, a LME model was used. Every scan was labelled on the following dimensions: participant identity (ID), MRI visit, task (task or rest), drug condition (prepsilocybin, psilocybin, MTP, postpsilocybin) and head motion (average framewise displacement). The rs-fMRI metrics (described below) were set as the dependent variable, drug (drug condition), task, framewise displacement (motion) and drug × task were defined as fixed effects, and participant ID and MRI session were random effects.

Let $y_{ij}$ be the rs-fMRI metric (for example, FC change score at a given vertex) for the $j$th observation (15 min fMRI scan) within the $i$th participant. The LME model can be written as:

$$y_{ij} = \beta_0 + \beta_{drug} \cdot drug_{ij} + \beta_{FD} \cdot FD_{ij} + \beta_{task} \cdot task_{ij} \\ + \beta_{task\text{-}by\text{-}drug} \cdot task_{ij} \cdot drug_{ij} + u_{0i} + v_{0j} + \varepsilon_{ij} \tag{1}$$

- $\beta_0$ is the intercept term.
- $\beta_{drug}$, $\beta_{FD}$, $\beta_{task}$ and $\beta_{task\text{-}by\text{-}drug}$ are the coefficients for the fixed effects predictors.
- $drug_{ij}$, frame displacement$_{ij}$ ($FD_{ij}$) and task$_{ij}$ are the values of the fixed effects predictors for the $j$th observation within the $i$th group.
- $u_{0i}$ represents the random intercept for the $i$th participant, accounting for individual-specific variability.
- $v_{0j}$ represents the random intercept for the $j$th observation within the $i$th participant, capturing scan-specific variability.
- $\varepsilon_{ij}$ is the error term representing unobserved random variation.

In MATLAB (Wilkinsonian notation), this model is expressed for every vertex Y(vertex) = fitlme(groupd, FC_Change(vertex) ~ drug + framewise displacement + task + task-by-drug + (1 |SubID) + (1 |session)).

To compensate for the implementations of this LME model on many rs-fMRI-related dependent variables, differences were highlighted when $P < 0.001$. All $P$ values reported are not corrected for multiple comparisons.

## Vertex-wise FC change

FC change ('distance') was calculated at the vertex level to generate FC change maps and a LME model (equation (1)) was used in combination with wild bootstrapping[119,120] and threshold-free cluster enhancement (TFCE)[95,121] to estimate $P$ values for $t$-statistic maps resulting from the model (Figs. 1a–d and 4). Wild bootstrapping is an approach to permutation testing that was designed for models that are not independent and identically distributed, and are heteroscedastic.

First, a FC change map was generated for every scan by computing, for each vertex, the average distance between its FC seedmap and the FC seedmap for each of that participant's baseline scans. As each participant had several baseline visits, FC change was computed for baseline scans by computing distance from all other baseline scans (excluding scans within the same visit). This provided a measure of day-to-day variability. Second, the distance value was used as the dependent variable $y_{ij}$ in the LME model to generate a $t$-statistic. Third, a wild bootstrapping procedure was implemented as follows. Several bootstrap samples ($B = 1,000$) were generated using the Rademacher procedure[120], in which the residuals were randomly inverted. Specifically, a Rademacher vector was generated by randomly assigning −1 or 1 values with equal probability to the residual of each observation. By element-wise multiplication of the original residuals with the Rademacher vector, bootstrap samples were created to capture the variability in the data.

For the observed $t$-statistic-map and each bootstrap sample, the TFCE algorithm was applied to enhance the sensitivity to clusters of significant voxels or regions while controlling for multiple comparisons. The value of the enhanced cluster statistic derived from the bootstrap samples was used to create a null distribution under the null hypothesis.

By comparing the original observed cluster statistic with the null distribution, *P* values were derived to quantify the statistical significance of the observed effect. The *P* values were obtained on the basis of the proportion of bootstrap samples that produced a maximum cluster statistic exceeding the observed cluster statistic.

The combined approach of wild bootstrapping with the Rademacher procedure and TFCE provided the method to estimate *P* values for our multilevel (drug condition, participant, session, task) design. This methodology accounted for the complex correlation structure, effectively controlled for multiple comparisons and accommodated potential autocorrelation in the residuals through the Rademacher procedure. By incorporating these techniques, association with psilocybin and other conditions was reliably identified amid noise and spatial dependencies.

### Whole-brain FC change
For analyses in Figs. 1e,g, 2 and 4a (bottom), Extended Data Fig. 3 and Supplementary Figs. 3, 4 and 6, distance calculations were computed on the FC matrix using *z*-transformed bivariate correlation of time courses from parcellated brain areas[62]. The effects of day-to-day, drug condition, task and framewise displacement and drug × task were directly examined by calculating the distance between functional network matrices generated from each scan. Root-mean-squared Euclidean distance was computed between the linearized upper triangles of the parcellated FC matrix between each pair of 15 min fMRI scans, creating a second-order distance matrix (Extended Data Fig. 3). Subsequently, the average distance (reported as 'whole-brain FC change') was examined for FC matrices that were from the same individual within a single session, from the same individual across days ('day-to-day'), from the same participant between drug and baseline (for example, psilocybin), from the same individual but different tasks ('task:rest'), from the same individual between highest motion scans and baseline ('hi:lo motion'), from different individuals ('between person'). In the 'high head motion' comparison ('hi:lo motion' in Supplementary Fig. 3), the two non-drug scans with the highest average framewise displacement were labelled and compared against all other baseline scans.

A LME model (equation (1)) and post hoc *t*-tests were used to assess statistical differences between drug conditions. A related approach using *z*-transformed bivariate correlation ('similarity' rather than distance) was also taken and results were unchanged (Supplementary Fig. 3c).

### Likelihood ratio test of participant-specific response
To test whether variability in participant-specific response to psilocybin was larger than would be expected by chance, we used a likelihood ratio test for variance of random slopes for a participant-specific response to psilocybin[48]. The difference in log likelihood ratios was compared to a null distribution of 1 million draws from a mixture of chi-squared distributions with degrees of freedom 1 and 2. We note that the likelihood ratio test of variance components is a non-standard problem[47] as the covariance matrix of the random effects is positive definite and the variances of random effects are non-negative. Finally, the test statistic for the likelihood ratio in this LME model was compared against a 50/50 mixture of two independent chi-squared distributions, each with one and two degrees of freedom, respectively.

### Assessing subjective experience
Subjective experience was assessed for drug sessions using the MEQ30[46] (Supplementary Methods). The MEQ30 is designed to capture the core domains of the subjective effects of psychedelics (as compared to the altered states of consciousness rating scales that more broadly assess effects of psychoactive drugs[122]) and is related to the therapeutic benefits of psychedelics. We applied a LME model across all drug sessions, similar to the one described above, but with MEQ30 total score as the dependent variable. Whole-brain FC change

and framewise displacement were modelled as fixed effects, and participant was modelled as a random effect. The same model was solved using FC change from every vertex to generate a vertex-wise map of the FC change versus MEQ30.

### Normalized FC change
The conditions above were compared by calculating normalized FC change scores using the following procedure: we (1) determined FC change for each condition compared to baseline as described above, (2) subtracted within-session distance for all conditions (such that within-session FC change was 0), (3) divided all conditions by day-to-day distance (such that day-to-day FC change was equal to 1). Thus, normalized whole-brain FC change values (for example, psilocybin versus base was 3.52) could be thought of as proportional to day-to-day variability.

### Data-driven MDS
We used a classical MDS approach to cluster parcellated connectomes across fMRI scans, as previously described[38]. This data-driven approach was used to identify how different parameters (for example, task, drug, individual) affect similarity and/or distance between networks. MDS places data in multidimensional space on the basis of the dissimilarity (Euclidean distance) among data points, which in this case means a data point represents the linearized upper triangle of a FC matrix. Every matrix was entered into the classical MDS algorithm (implemented using MATLAB 2019, cmdscale.m). Many dimensions of the data were explored. The eigenvectors were multiplied by the original FC matrices to generate a matrix of eigenweights that corresponded to each dimension. These eigenweights were also applied to other rs-fMRI psychedelics datasets to generate dimensions scores (section 'Other datasets').

### Rotation-based null model (spin test) for network specificity
To assess network specificity of FC change values, we calculated average FC change of matched null networks consisting of randomly rotated networks with preserved size, shape and relative position to each other[62,97]. To create matched random networks, we rotated each hemisphere of the original networks a random amount around the *x*, *y* and *z* axes on the spherical expansion of the cortical surface[62]. This procedure randomly relocated each network while maintaining networks' sizes, shapes and relative positions to each other. Random rotation followed by computation of network-average FC change score was repeated 1,000 times to generate null distributions of FC change scores. Vertices rotated into the medial wall were not included in the calculation. Actual psilocybin FC change was then compared to null rotation permutations to generate a *P* value for the 12 networks that were consistently present across every participant's Infomap parcellation. For bar graph visualization (Fig. 1 and Supplementary Fig. 1b), networks with greater change (*P* < 0.05 based on null rotation permutations) are shown in their respective colour and other networks are shown in grey.

### NGSC
We used an approach previously validated to assess spatial complexity (termed entropy) or neural signals[61]. Temporal principal component analysis was conducted on the full BOLD dense timeseries, which yielded *m* principal components (*m* roughly 80 K surface vertices and subcortical voxels) and associated eigenvalues. The normalized eigenvalue of the *i*th principal component was calculated as

$$\lambda_i' = \frac{\lambda_i}{\sum_{i=1}^{m} \lambda_i'} \tag{2}$$

where *m* is the number of principal components, and $\lambda_i$ and $\lambda_i'$ represent the eigenvalue and the normalized eigenvalue of the *i*th principal

component, respectively. Last, the NGSC, defined as the normalized entropy of normalized eigenvalues, was computed using the equation:

$$\text{NGSC} = -\frac{\sum_{i=1}^{m} \lambda_i' \log \lambda_i^i}{\log m} \tag{3}$$

The NGSC computed above attains values from the interval 0 to 1. The lowest value NGSC = 0 would mean the brain-wide BOLD signal consisted of exactly one principal component or spatial mode, and there is maximum global FC between all vertices. The highest value NGSC = 1 would mean the total data variance is uniformly distributed across all $m$ principal components, and a maximum spatial complexity or a lowest FC is found.

NGSC was additionally calculated at the 'parcel level'. To respect areal boundaries, this was done by first generating a set of individual-specific parcels in every participant (on all available resting fMRI sessions concatenated) using procedures described oreviously[39,62].

NGSC maps were compared to PET-based 5-HT$_{2A}$ receptor binding maps published in ref. 33. Similarity was assessed by computing the bivariate correlation between NGSC values and 5-HT$_{2A}$ binding across 324 cortical parcels from the Gordon–Laumann parcellation.

## Persistent effects analysis

To assess the persistent effects of psilocybin, we compared FC changes 1–21 days postpsilocybin to predrug baseline. The FC change analysis (described above) indicated that connectivity at the whole-brain level did not change following psilocybin (Supplementary Fig. 1). A screen was conducted with $P < 0.05$ threshold to identify brain networks or areas showing persistent effects. This analysis identified the anterior hippocampus as a candidate region of interest for persistent FC change (section 'Baseline/after psilocybin FC change analysis' in Supplementary Methods).

We assessed change in anterior hippocampus 'FC change' pre- versus postpsilocybin using the LME model described previously. In this model, all sessions before psilocybin (irrespective or cross-over order) were labelled as prepsilocybin and all sessions within 21 days after psilocybin were labelled as postpsilocybin.

As a control, we tested anterior hippocampus FC change pre- versus post-MTP using both the LME model, and an equivalence test. To control for potential persistent psilocybin effects, only the block of scans immediately before and after MTP were used (for example, if a participant took MTP as drug 1, then all baseline scans were labelled as 'pre-MTP' and all scans between drugs 1 and 2 were labelled 'post-MTP').

Equivalence testing (to conclude no change in anterior hippocampus after MTP) was accomplished by setting $\delta = 0.5$ standard deviation of FC change across pre-MTP sessions. We computed the 90% CI of change in FC change between pre- and post-MTP sessions. If the bounds of the 90% CI were within $\pm\delta$, then equivalence was determined[123].

## Other datasets

Raw fMRI and structural data published previously[55,56] were run through our in-house registration and processing pipeline described above. These datasets were used for replication, external validation and generalization to another classic psychedelic (that is, LSD) for the measures described above (for example, NGSC and the MDS-derived psilocybin FC dimension, dimension 1).

Using the data from ref. 55: $n = 15$ healthy adults (five women, mean age 34.1 years, s.d. 8.2) completed two scanning sessions (psilocybin and saline) that included an eyes-closed resting-state BOLD scan for 6 min before and following i.v. infusion of drug. fMRI data were acquired using a gradient-echo-planar imaging sequence, TR and TE of 3,000 and 35 ms, field-of-view 192 mm, 64 × 64 acquisition matrix, parallel acceleration factor of 2 and 90° flip angle.

Using the data from ref. 56: healthy adults completed two scanning sessions (LSD and saline), which included an eyes-closed resting-state BOLD scan acquired for 22 min following i.v. drug infusion lasting 12 min. $n = 20$ participants completed the protocol, but data were used for $n = 15$ (four women; mean age 30.5, standard deviation 8.0) deemed suitable for BOLD analyses. fMRI data were acquired using a gradient-echo-planar imaging sequence, TR and TE of 2,000 and 35 ms, field-of-view 220 mm, 64 × 64 acquisition matrix, parallel acceleration factor of 2, 90° flip angle and 3.4 mm isotropic voxels.

The ABCD database resting-state functional MRI[59] (annual release v.2.0, https://doi.org/10.15154/1503209) was used to replicate the effects of stimulant use on FC. Preprocessing included framewise censoring with a criterion of frame displacement less than or equal to 0.2 mm in addition the standard predefined preprocessing procedures[124]. Participants with fewer than 600 frames (equivalent to 8 min of data after censoring) were excluded from the analysis. Parcel-wise group-averaged FC matrices were constructed for each participant as described above for 385 regions on inter-test in the brain.

Use of a stimulant (for example, MTP, amphetamine salts, lisdexamfetamine) in the last 24 h was assessed by parental report. Participants with missing data were excluded. Regression analysis was used to assess the relationship between FC (edges) and stimulant use in the last 24 h. Framewise displacement (averaged over frames remaining after censoring) was used as a covariate to account for motion-related effects. The $t$-values that reflect the relationship between stimulant use and FC were visualized on a colour scale from −5 to +5 to provide a qualitative information about effect of stimulant use on FC.

## Reporting summary

Further information on research design is available in the Nature Portfolio Reporting Summary linked to this article.

## Data availability

All data from individual participants P1–P7 are available at https://wustl.box.com/v/PsilocybinPFM, with a password available on completion of a data use agreement. The ABCD data used in this report came from ABCD the Annual Release 2.0, https://doi.org/10.15154/1503209. The ABCD data repository grows and changes over time (https://nda.nih.gov/abcd). The Imperial College London psilocybin and LSD datasets are available upon request.

## Code availability

Data processing code for the psilocybin precision functional mapping data can be found at https://wustl.box.com/s/dmj5s3h9pxt9b-cw9mm3ai9c15y756o79. Code specific to analyses can be found at https://gitlab.com/siegelandthebrain1/Psilocybin_PFM/. Data processing code for the ABCD data can be found at https://github.com/DCAN-Labs/abcd-hcp-pipeline. Matching task stimuli are available at https://gitlab.com/siegelandthebrain1/Psilocybin_PFM/-/blob/main/image_task_clean.zip. Software packages incorporated into the above pipelines for data analysis included: MATLAB R2019b, https://www.mathworks.com/ (including Psychtoolbox v.2.0 and Statistics and Machine Learning Toolbox v.11.6); Connectome Workbench v.1.5; http://www.humanconnectome.org/software/connectome-workbench.html; Freesurfer v.6.2, https://surfer.nmr.mgh.harvard.edu/; FSL v.6.0, https://fsl.fmrib.ox.ac.uk/fsl/fslwiki; 4dfp tools, https://4dfp.readthedocs.io/en/latest/; Infomap, https://www.mapequation.org; Cifti MATLAB utilities (including spin test): https://github.com/MidnightScanClub/SCAN and 4dfp tools, https://4dfp.readthedocs.io/en/latest/. MRI pulse sequences used to acquire the data are provided at https://gitlab.com/siegelandthebrain1/Psilocybin_PFM/-/blob/main/NP1161_MRI_sequence.pdf.

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

**Acknowledgements** This work was supported by the Taylor Family Institute Fund for Innovative Psychiatric Research (J. S. Siegel, T.O.L., G.E.N.); the McDonnell Center for Systems Neuroscience (J. S. Siegel, G.E.N.); the Institute of Clinical and Translational Science (G.E.N.); National Institutes of Health (NIH) grants MH112473 (J. S. Siegel, S.S., D.A.B., C.H., G.E.N.), T32 DA007261 (J. S. Siegel), NS123345 (B.P.K.), MH129616 (T.O.L.), MH121276 (N.U.F.D., E.M.G., D.A.F.), MH118370 (C.G.), NS124738 (C.G.), MH096773 (D.A.F., N.U.F.D.), MH122066 (D.A.F., E.M.G., N.U.F.D.), MH124567 (D.A.F., E.M.G., N.U.F.D.), NS129521 (E.M.G., D.A.F., N.U.F.D.) and NS088590 (N.U.F.D.); the National Spasmodic Dysphonia Association (E.M.G.); the Ralph Metzner Professorship and the Tianqiao and Chrissy Chen Institute (R.C.-H.); the Intellectual and Developmental Disabilities Research Center (N.U.F.D.); by the Kiwanis Foundation (N.U.F.D.); the Washington University Hope Center for Neurological Disorders (E.M.G., N.U.F.D.) and by Mallinckrodt Institute of Radiology pilot funding (E.M.G., N.U.F.D.). Furthermore, this study used data from the ABCD study, supported by National Institutes of Health grant no. U01DA041120. We give a special thanks to our study participants, who completed a demanding protocol with grace for the benefit of scientific inquiry. Data used in the preparation of this article were obtained from the Adolescent Brain Cognitive Development (ABCD) Study (https://abcdstudy.org), held in the NIMH Data Archive (NDA). This is a multisite, longitudinal study designed to recruit more than 10,000 children age 9-10 and follow them over 10 years into early adulthood. The ABCD Study is supported by the National Institutes of Health and additional federal partners under award numbers U01DA041048, U01DA050989, U01DA051016, U01DA041022, U01DA051018, U01DA051037, U01DA050987, U01DA041174, U01DA041106, U01DA041117, U01DA041028, U01DA041134, U01DA050988, U01DA051039, U01DA041156, U01DA041025, U01DA041120, U01DA051038, U01DA041148, U01DA041093, U01DA041089, U24DA041123, U24DA041147. A full list of supporters is available at https://abcdstudy.org/federal-partners.html. A listing of participating sites and a complete listing of the study investigators can be found at https://abcdstudy.org/consortium_members/. ABCD consortium investigators designed and implemented the study and/or provided data but did not necessarily participate in the analysis or writing of this report. This manuscript reflects the views of the authors and may not reflect the opinions or views of the NIH or ABCD consortium investigators.

**Author contributions** The concept came from J. S. Siegel and G.E.N. The study was designed by J. S. Siegel, S.S., T.O.L., C.L.R., E.J.L., A.Z.S. and G.E.N. Data acquisition and processing were done by J. S. Siegel, S.S., T.R.R., D.P., C.H., J. S. Shimony, J.A.S., D.A.B., K.M.S., F.I.W., J.M., E.Y., S.M.N., L.V., D.A.F. and A.Z.S.. Data analysis and interpretation were carried out by J. S. Siegel, B.P.K., E.M.G., T.O.L., N.V.M., C.G., R.V.C., S.R.K., D.F.W., J.A.P.-C., R.T.S., Y.C., R.C.-H., M.E.R., G.E.N. and N.U.F.D. The paper was written and revised by J. S. Siegel, S.S., M.E.R., A.Z.S., G.E.N. and N.U.F.D. Participant 7 was author N.U.F.D.

**Competing interests** Within the past year, J. S. Siegel has been an employee of Sumitomo Pharma America and received consulting fees from Longitude Capital. J. S. Siegel, N.U.F.D., T.O.L. and E.M.G. have submitted a provisional patent (patent no. 020949/US 15060-1787) for the use of precision functional mapping for measuring target engagement by experimental therapeutics. R.T.S. has received consulting compensation from Octave Bioscience and compensation for reviewership duties from the American Medical Association. C.L.R. serves as a consultant to Usona Institute and Novartis and receives research support from the Tiny Blue Dot Foundation. G.E.N. has received research support from Usona Institute (drug only). She has served as a paid consultant for Carelon, Alkermes, Inc., Sunovion Pharmaceuticals, Inc. and Novartis Pharmaceuticals Corp. T.O.L. holds a patent for taskless mapping of brain activity licenced to Sora Neurosciences and a patent for optimizing targets for neuromodulation, implant localization and ablation is pending. J. S. Siegel is a consultant and received stock options in Sora Neuroscience, and company that focuses on resting-state analysis. D.A.F. and N.U.F.D. are cofounders of Turing Medical Inc, have financial interest, may benefit financially if the company is successful in marketing FIRMM motion monitoring software products, may receive royalty income based on FIRMM technology developed at WUSOM and licenced to Turing Medical Inc. S.M.N., E.M.G. and T.O.L. have received consulting fees from Turing Medical Inc. D.F.W. is a consultant for Engrail Therapeutics and receives contract funds for WUSOM research studies from Eisai, Anavex and Roche. These potential conflicts of interest have been reviewed and are managed by WUSOM. The other authors declare no competing interests. All authors report no financial interest in psychedelics companies.

**Additional information**
**Correspondence and requests for materials** should be addressed to Joshua S. Siegel.

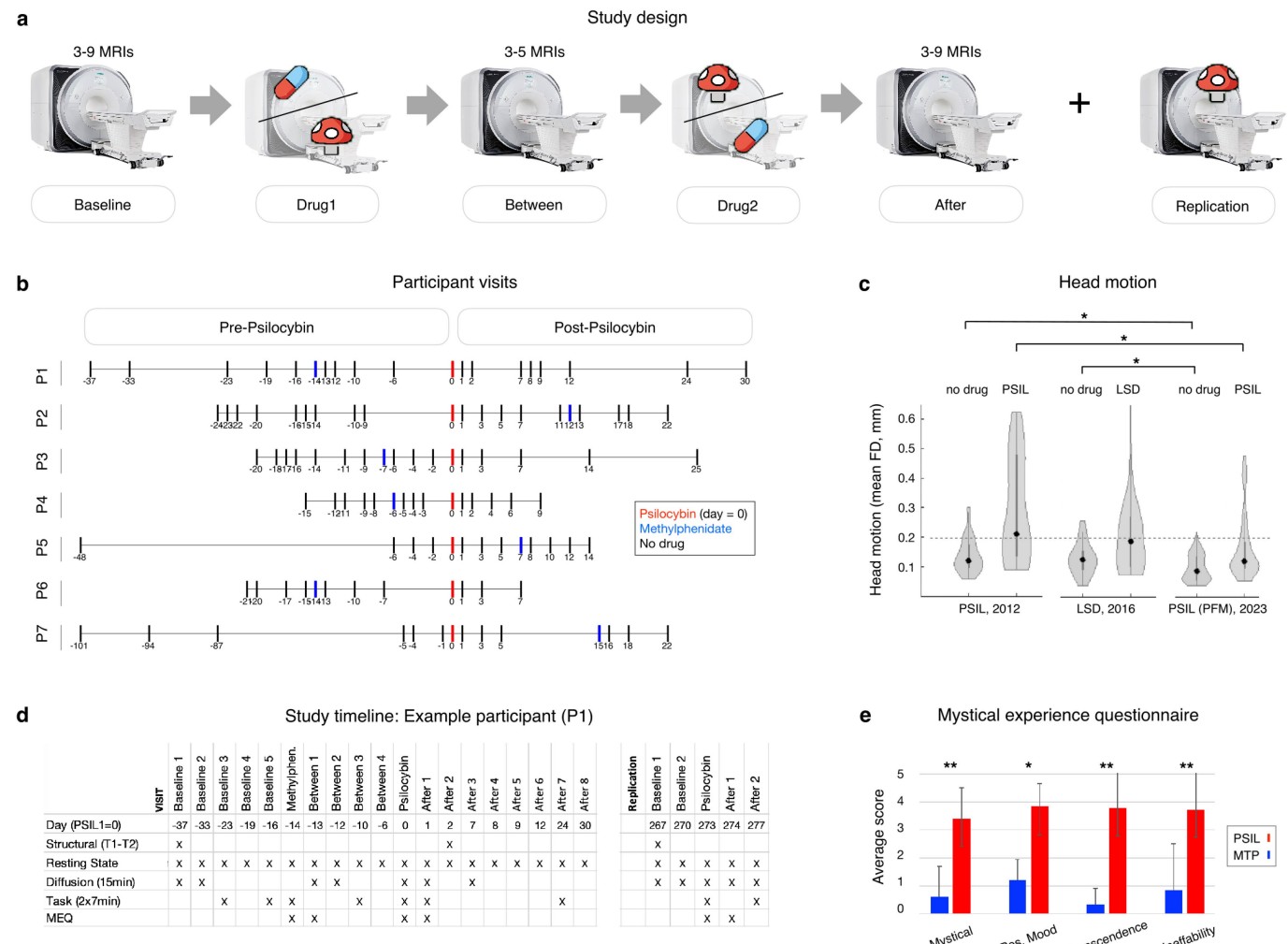

**Extended Data Fig. 1 | Quantifying psilocybin effects with precision functional mapping: design. a)** Schematic illustrating the study protocol of the individual-specific precision functional mapping study of acute and persistent effects of psilocybin (single dose: 25 mg). Repeated longitudinal study visits enabled high-fidelity individual brain mapping, measurement of day-to-day variance, and acclimation to the scanner. The open label replication protocol 6-12 months later included one or two scans each of baseline, psilocybin, and after drug. **b)** Timeline of imaging visits for 7 participants. **c)** Head motion comparisons across psychedelics studies[55,56]. Average head motion (FD, framewise displacement, in mm) off and on drug compared between our dataset and prior psychedelic fMRI studies. Unpaired two-sided $t$-test:

$n_{PSIL\,2012} = 15$, $n_{LSD,2016} = 20$, $n_{PSIL(PFM)} = 7$; off drug PSIL2012-PSIL(PFM) $t_{(274)} = -4.57$, $P_{uncorr} = 7.33 \times 10^{-5}$; off drug LSD2016-PSIL(PFM) $t_{(286)} = -4.03$, $P_{uncorr} = 7.34 \times 10^{-6}$; on drug PSIL2012-PSIL(PFM) $t_{(46)} = -1.80$, $P_{uncorr} = 0.079$; on drug LSD2016-PSIL(PFM) $t_{(88)} = -0.73$, $P_{uncorr} = 0.46$. Dotted line at FD of 0.2 mm. Dark gray bars indicate quartiles, light gray violins indicate distribution. *$P_{uncorr} < 0.05$, +$P < 0.1$. **d)** Timeline for an example participant (P1). **e)** Participants reported significantly higher scores on all dimensions of the mystical experience questionnaire during psilocybin (red) than placebo (40 mg methylphenidate; blue). Paired two-tailed $t$-test, $n = 7$; Mystical $t_{(6)} = -3.64$, $P_{uncorr} = 0.011$; Positive Mood $t_{(6)} = -5.44$, $P_{uncorr} = 0.0016$; Transcendence $t_{(6)} = -4.98$, $P_{uncorr} = 0.0025$; Ineffability $t_{(6)} = -2.54$, $P_{uncorr} = 0.044$. Error bars indicate SEM.

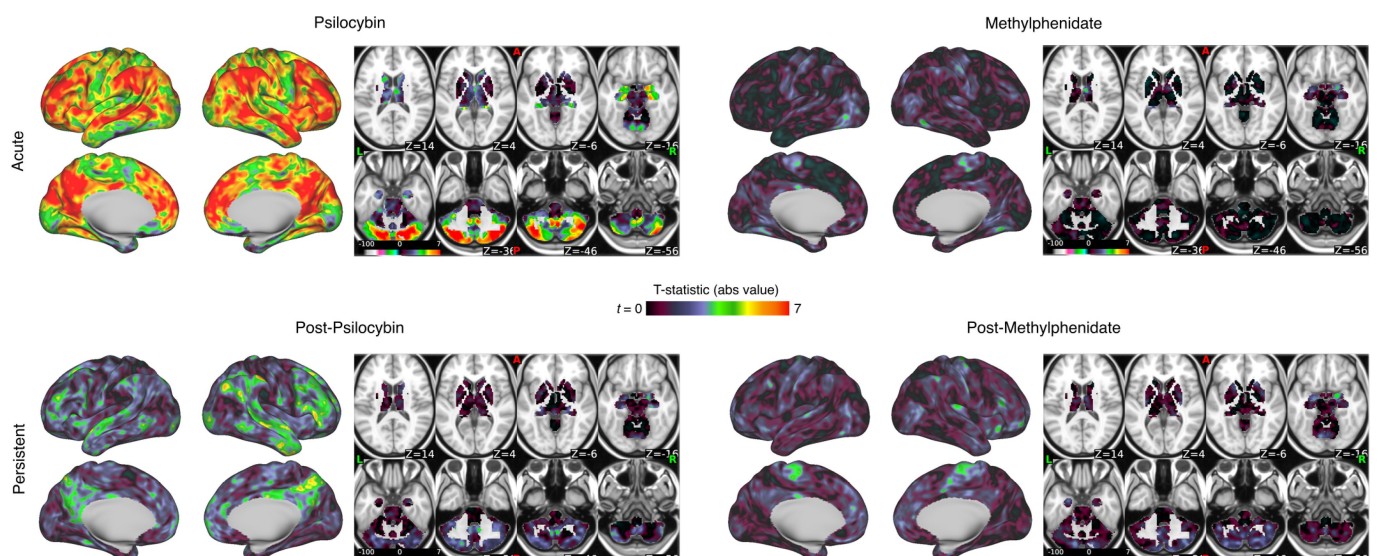

**Extended Data Fig. 2 | Unthresholded vertex-wise FC change maps.**
T-statistic maps, resulting from the linear mixed effects (LME) model based on vertex-wise FC change (Euclidean distance from baseline scans) across the cortex and subcortical structures for every scan. Higher t values indicate a larger change from baseline (pre-drug) scans. Effects of drug condition (baseline, psilocybin, methylphenidate, post-psilocybin, post-methylphenidate), were modeled as fixed effects. For example, if drug 1 was psilocybin and drug 2 was methylphenidate, then scans between drug visits were labeled post-psilocybin and scans after drug 2 were labeled post-methylphenidate.

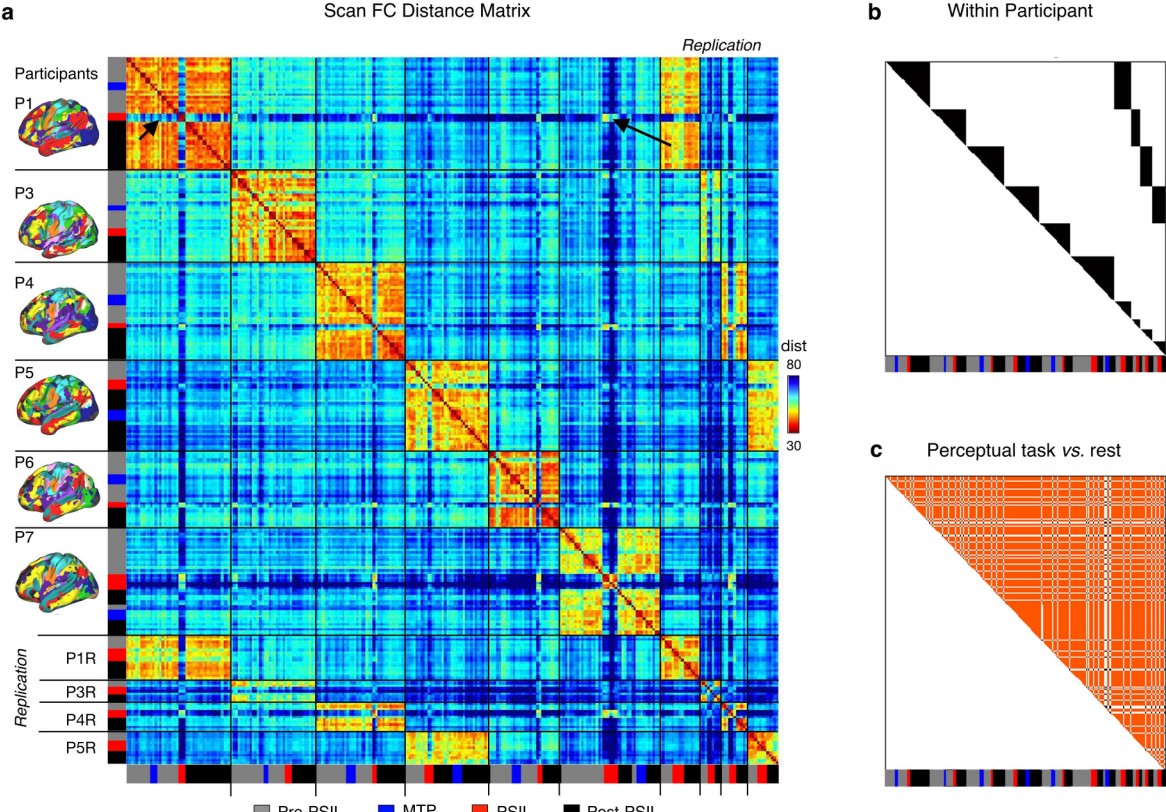

**a** Scan FC Distance Matrix

*Replication*

Participants
P1
P3
P4
P5
P6
P7

*Replication*
P1R
P3R
P4R
P5R

dist
80
30

Pre-PSIL  MTP  PSIL  Post-PSIL

**b** Within Participant

**c** Perceptual task *vs.* rest

**Extended Data Fig. 3 | Functional connectivity (FC) distance and condition matrices for all fMRI scans.** Following Gratton et al.[38], we compared FC matrices between rs-fMRI sessions to quantify contributors to variability in whole-brain FC. Under this approach, the effects of group, individual, session, and drug (as well as their interactions) are examined by first calculating the Euclidean distance among every pair of FC matrices (i.e., distance among the linearized upper triangles). **a)** In the resulting second-order distance matrix, each row and column show whole-brain FC from a single study visit. The colours in the matrix indicate distance between functional networks for a pair of visits (i.e., Euclidean distance between the linearized upper triangles of two FC matrices). Panels **b** and **c** demonstrate how the distance matrix was subdivided to compare different conditions. **b)** Black triangles represent distinct individuals. Replication protocol visits are listed at the end. **c)** Task and rest scans are shown in white and orange, respectively. Note that psilocybin scans (black arrows pointing to P1 psilocybin scans in panel **a** are very dissimilar to no-drug scans from the same individual (left arrow; in **a**) but have heightened similarity to psilocybin scans from other individuals (right arrow in **a**).

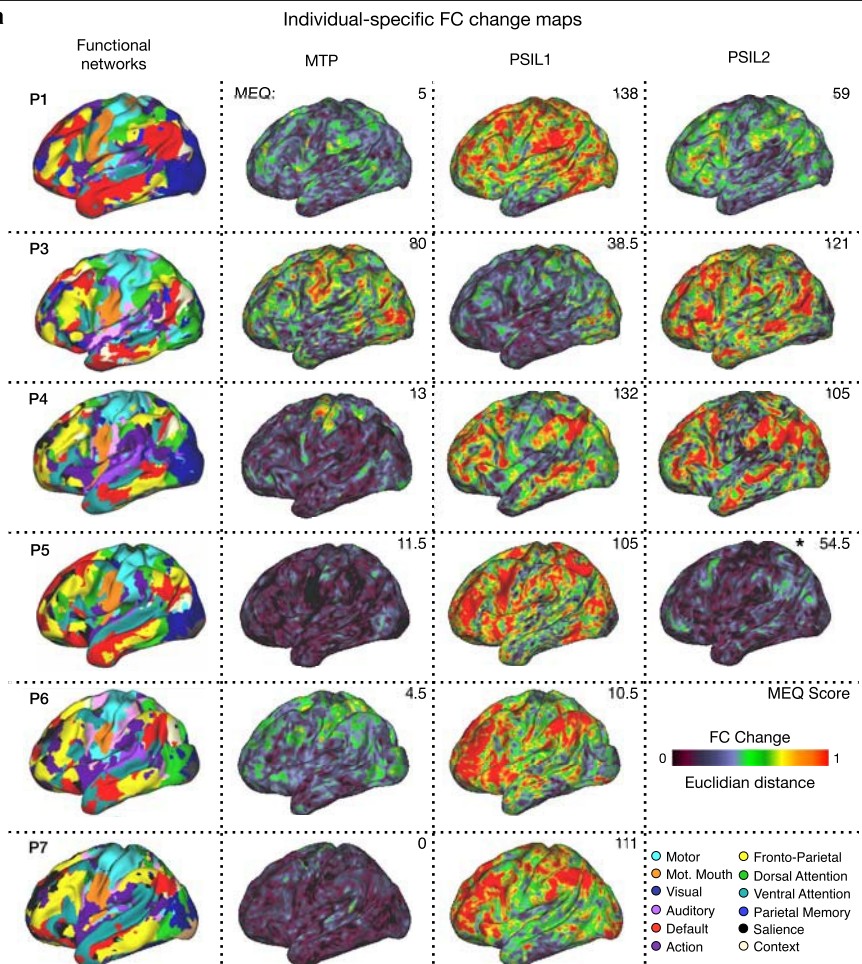

**Extended Data Fig. 4 | Participant-specific FC change maps for drug sessions.** Individual participant methylphenidate (MTP) and psilocybin (PSIL) FC change maps. Left most column shows individuals' functional networks. Right 3 columns show FC change maps, generated by calculating Euclidean distance from baseline seedmaps for each vertex. For each session the total score on the Mystical Experience Questionnaire (MEQ30: out of a maximum of 150) is given in the upper right corner. *P5 had an episode of emesis 30 minutes after drug ingestion during PSIL2.

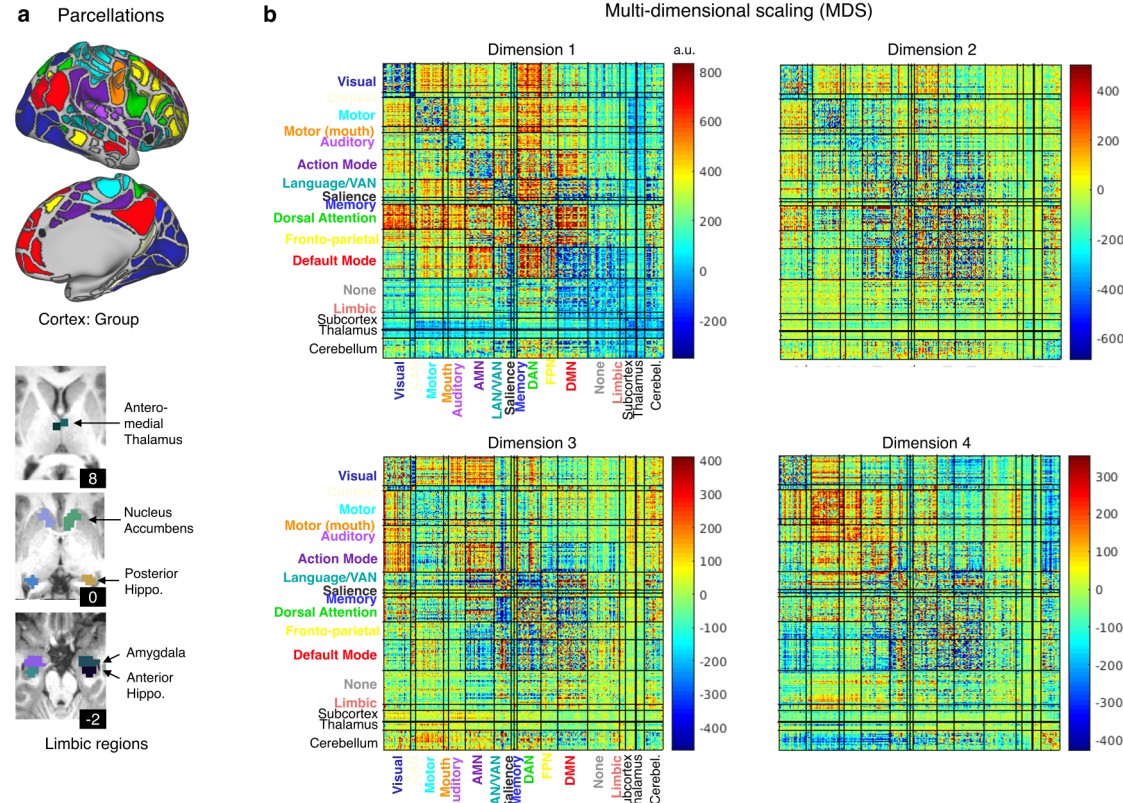

**Extended Data Fig. 5 | Multi-dimensional scaling, dimension edge weights. a)** Group parcellation (324 cortical and 61 subcortical parcels)[31] **b)** Weights from the first 4 dimensions generated by multi-dimensional scaling of the full dataset. The color of each pixel in the plot represents the weight of a given edge. Dimension 1 captures the loss of network integration (on diagonal boxes) and segregation (off diagonal boxes) of psilocybin. Dimensions 2 and 3 primarily explain individual differences and do not show network patterns as clearly. Dimension 4 captures shared effects of psilocybin (PSIL) and methylphenidate (MTP) on sensorimotor systems (suspected arousal effects).

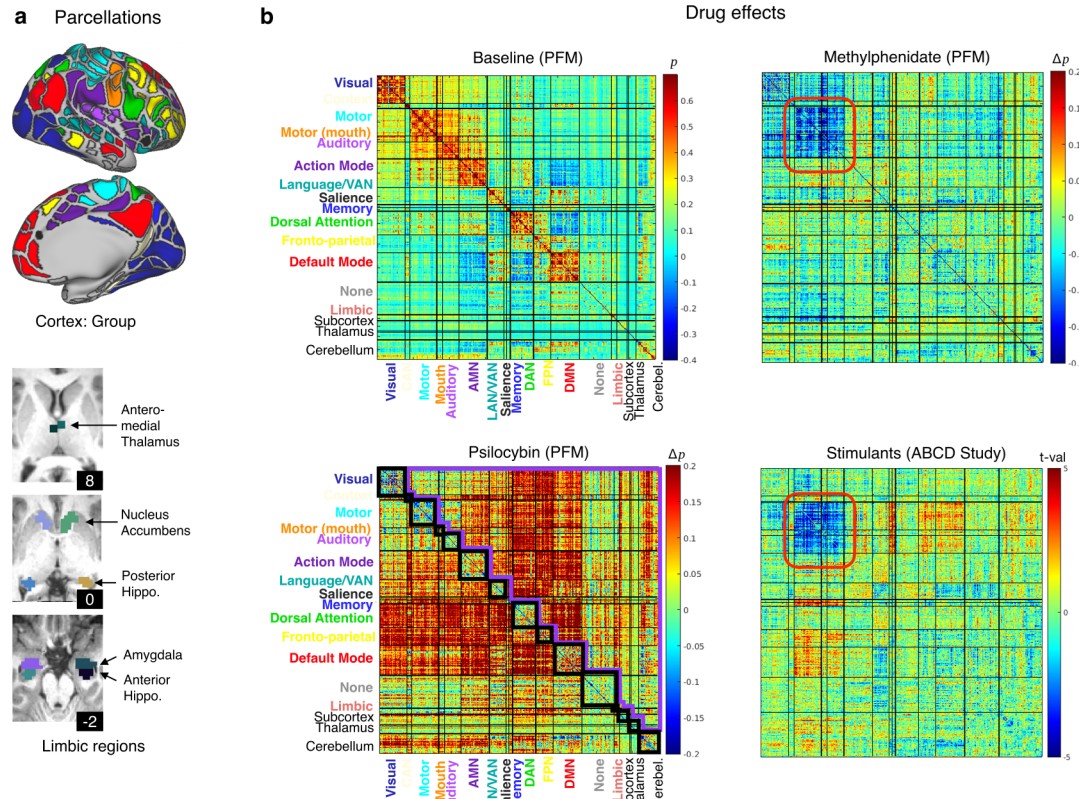

**Extended Data Fig. 6 | Average functional connectivity (FC) matrices by condition. a)** Group parcellation (324 cortical and 61 subcortical parcels)[31]. **b)** Average FC matrices and condition differences. Top left shows the group average FC adjacency matrix. Bottom left shows the effect of psilocybin, e.g. increased correlation between dorsal attention, fronto-parietal, and default mode network to each other and to other cortical, limbic, and cerebellar systems. Top right shows effect of methylphenidate. For comparison and validation, we compared methylphenidate to the main effect of stimulant use within the last 24 hours (bottom right, $n$ = 487 yes, $n$ = 7992 no) in ABCD rs-fMRI data (bottom right).

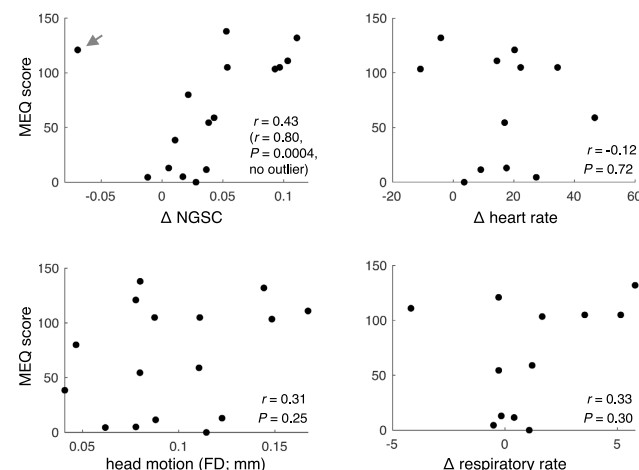

**Extended Data Fig. 7 | Correlations with mystical experience scores.**
Comparison of MEQ30 score (y-axes) to global desynchronization (top left;
NGSC change, drug minus baseline), head motion (bottom left; framewise
displacement (FD) in mm), heart rate change (top right; drug minus baseline),
and respiratory rate change (bottom right; drug minus baseline), for all drug
sessions. Statistics (*rho*, *P*) are based on bivariate correlation, two-sided,
uncorrected. In the case of Δ NGSC, statistics are reported before and after
the removal of an outlier point (> 2 SD lower than mean, indicated by the gray
arrow).

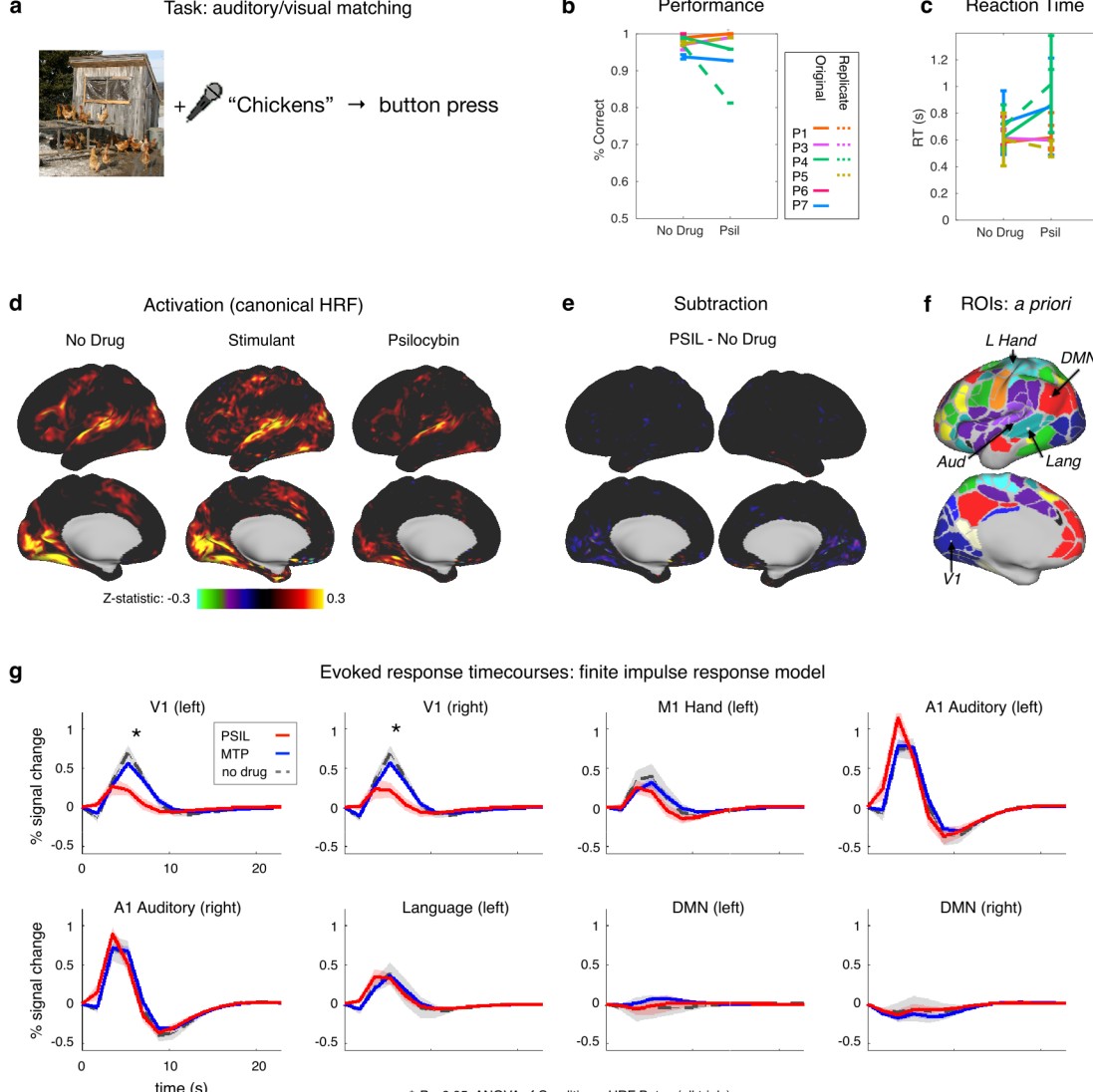

**Extended Data Fig. 8 | Auditory-visual matching fMRI task. a)** Schematic of auditory/visual matching task design. **b)** Comparison of performance ('No Drug' and psilocybin conditions are at ceiling). Lines indicate means and standard deviation across sessions. Number of task sessions are indicated in Supplementary Table 1. **c)** Comparison of reaction time (RT). Lines indicate mean and standard deviation across all trials (48 trials per session). **d)** Task fMRI activation maps (beta weights) and **e)** contrasts (simple subtraction) using the canonical hemodynamic response function (HRF). **f)** Eight a priori regions of interest for timecourse analyses. **g)** Average timecourses from the regions of interest shown in panel **f**, calculated using finite impulse response model over 13 TR x 1.761 s/TR = 22.89 seconds, for all trials. Shaded area around each line indicates SEM. ANOVAN of Condition x HRF Beta (Main effect of all trials) magnitude testing effect of drug, two-sided: Left V1, $F_{(2,40)} = 3.91$, $P = 0.030$; Right V1, $F_{(2,40)} = 4.40$, $P = 0.020$; Left M1 hand, $F_{(2,40)} = 0.40$, $P = 0.68$; Left Auditory A1, $F_{(2,40)} = 0.22$, $P = 0.81$; Right Auditory A1, $F_{(2,40)} = 0.77$, $P = 0.47$; Left Language, $F_{(2,40)} = 0.025$, $P = 0.98$; Left DMN, $F_{(2,40)} = 1.15$, $P = 0.33$; Right DMN, $F_{(2,40)} = 0.14$, $P = 0.87$. *$P < 0.05$. P-values are uncorrected for multiple comparisons.

# Reporting Summary

## Statistics

For all statistical analyses, confirm that the following items are present in the figure legend, table legend, main text, or Methods section.

| n/a | Confirmed | |
|---|---|---|
| ☐ | ☒ | The exact sample size (*n*) for each experimental group/condition, given as a discrete number and unit of measurement |
| ☐ | ☒ | A statement on whether measurements were taken from distinct samples or whether the same sample was measured repeatedly |
| ☐ | ☒ | The statistical test(s) used AND whether they are one- or two-sided *Only common tests should be described solely by name; describe more complex techniques in the Methods section.* |
| ☐ | ☒ | A description of all covariates tested |
| ☐ | ☒ | A description of any assumptions or corrections, such as tests of normality and adjustment for multiple comparisons |
| ☐ | ☒ | A full description of the statistical parameters including central tendency (e.g. means) or other basic estimates (e.g. regression coefficient) AND variation (e.g. standard deviation) or associated estimates of uncertainty (e.g. confidence intervals) |
| ☐ | ☒ | For null hypothesis testing, the test statistic (e.g. *F*, *t*, *r*) with confidence intervals, effect sizes, degrees of freedom and *P* value noted *Give P values as exact values whenever suitable.* |
| ☒ | ☐ | For Bayesian analysis, information on the choice of priors and Markov chain Monte Carlo settings |
| ☐ | ☒ | For hierarchical and complex designs, identification of the appropriate level for tests and full reporting of outcomes |
| ☐ | ☒ | Estimates of effect sizes (e.g. Cohen's *d*, Pearson's *r*), indicating how they were calculated |

*Our web collection on statistics for biologists contains articles on many of the points above.*

## Software and code

Policy information about availability of computer code

| Data collection | Task stimuli were presented using matlab psychtoolbox and custom code. Matching task fMRI code and stimuli are available at https://gitlab.com/siegelandthebrain1/Psilocybin_PFM/-/blob/main/image_task_clean.zip |
|---|---|
| Data analysis | Data processing code for the psilocybin PFM data (and Imperial College London LSD/psilocybin datasets) can be found here: http://128.252.217.203/repo/Pipeline/ (SVN Repository)<br>Code specific to psilocybin PFM task task analysis and manuscript analyses can be found here: https://gitlab.com/siegelandthebrain1/Psilocybin_PFM<br>Data processing code for the ABCD data can be found here: https://github.com/DCAN-Labs/abcd-hcp-pipeline<br>Software packages incorporated into the above pipelines for data analysis included:<br>Matlab R2019b, https://www.mathworks.com/ (including Psychtoolbox version 2.0 and Statistics and Machine Learning Toolbox version 11.6)<br>Cifti matlab utilities (including spin test): https://github.com/MidnightScanClub/SCAN<br>Connectome Workbench 1.5, http://www.humanconnectome.org/software/connectome-workbench.html<br>Freesurfer v6.2, https://surfer.nmr.mgh.harvard.edu/<br>FSL 6.0, https://fsl.fmrib.ox.ac.uk/fsl/fslwiki<br>4dfp tools, https://4dfp.readthedocs.io/en/latest/<br>Infomap, www.mapequation.org<br>CARET: http://brainvis.wustl.edu/ |

For manuscripts utilizing custom algorithms or software that are central to the research but not yet described in published literature, software must be made available to editors and reviewers. We strongly encourage code deposition in a community repository (e.g. GitHub). See the Nature Portfolio guidelines for submitting code & software for further information.

# Data

Policy information about availability of data

All manuscripts must include a data availability statement. This statement should provide the following information, where applicable:
- Accession codes, unique identifiers, or web links for publicly available datasets
- A description of any restrictions on data availability
- For clinical datasets or third party data, please ensure that the statement adheres to our policy

> All data from individual subjects P1-P7 is available upon completion of a data use agreement: https://wustl.app.box.com/folder/139716285911
>
> The ABCD data used in this report came from ABCD the Annual Release 2.0, DOI 10.15154/1503209.
>
> The Imperial College London psilocybin and LSD datasets (Carhart-Harris et al., 2012, Carhart-Harris et al., 2016) are available upon request.

# Research involving human participants, their data, or biological material

Policy information about studies with human participants or human data. See also policy information about sex, gender (identity/presentation), and sexual orientation and race, ethnicity and racism.

| | |
|---|---|
| Reporting on sex and gender | See supplemental Table 1 for P1-P7 characteristics<br>Male=4, Female=3<br>Imperial College LSD dataset: N=4/15 females (see Carhart-Harris et al, 2016)<br>Imperial College Psilocybin dataset: N=2/15 females (see Carhart-Harris et al, 2016)<br>ABCD: 49%M, 51%F |
| Reporting on race, ethnicity, or other socially relevant groupings | Not relevant/reported |
| Population characteristics | Details on age/education/past psychedelics use/personality are provided in Supplemental Table 1<br>Participant P1 P2 P3 P4 P5 P6 P7<br>Sex (self-report) M M F F M F M<br>Age Range (years) 41-45 36-40 36-40 36-40 18-20 21-25 41-45<br>Weight (lbs) 227 151 148 173 169 224 215<br><br>Carhart-Harris et al., 2012: n = 15 healthy adults (five females, mean age 34.1, SD 8.2)<br>Carhart-Harris et al., 2016: n = 20 participants completed the protocol but data were used for n = 15 (four females; mean age 30.5, SD 8.0)<br>ABCD: 9-10 year old males and females |
| Recruitment | WU P1-P7: Healthy adult participants were recruited from the Washington University community via word of mouth.<br><br>Imperial College Psilocybin/LSD: participants were recruited via word of mouth and provided written informed consent to participate.<br><br>ABCD: A very important motivation for the ABCD study is that its sample should reflect, as best as possible, the sociodemographic variation of the US population. The ABCD cohort recruitment emulates a multi-stage probability sample of eligible children: A nationally distributed set of 21 primary stage study sites, a probability sampling of schools within the defined catchment areas for each site, and recruitment of eligible children in each sample school. The major departure from traditional probability sampling of U.S. children originates in how participating neuroimaging sites were chosen for the study. Although the 21 ABCD study sites are well-distributed nationally the selection of collaborating sites is not a true probability sample of primary sampling units (PSUs) but was constrained by the grant review selection process and the requirement that selected locations have both the research expertise and the neuroimaging equipment needed for the study protocol. |
| Ethics oversight | P1-P7: All aspects of this study were approved by the Washington University School of Medicine Human Studies Committee and Institutional Review Board, the Food and Drug Administration (Center for Drug Evaluation and Research), and the Drug Enforcement Agency (which reviews all research protocol for the use of a Schedule 1 drug in humans).<br><br>Imperial College Psilocybin/LSD datasets: approved by the National Research Ethics Service committee London-West London and was conducted in accordance with the revised declaration of Helsinki (2000), the International Committee on Harmonization Good Clinical Practice guidelines, and National Health Service Research Governance Framework.<br><br>ABCD: The ABCD Study obtained centralized institutional review board approval from the University of California, San Diego, and each of the 21 study sites obtained local institutional review board approval. |

Note that full information on the approval of the study protocol must also be provided in the manuscript.

# Field-specific reporting

Please select the one below that is the best fit for your research. If you are not sure, read the appropriate sections before making your selection.

☒ Life sciences ☐ Behavioural & social sciences ☐ Ecological, evolutionary & environmental sciences

For a reference copy of the document with all sections, see nature.com/documents/nr-reporting-summary-flat.pdf

# Life sciences study design

All studies must disclose on these points even when the disclosure is negative.

| | |
|---|---|
| Sample size | Precision Functional Mapping focuses on within- rather than across-individual analysis, the following considerations were used to estimate sample size to measure persistent drug effects:<br><br>Conservative sample size calculation for single subject statistics: Based on priors from ketamine data, in a group of depression subjects imaged at two different baseline timepoints, the within-subject standard deviation of connectivity between the was 0.085. The average change in connectivity between sgACC and DMN two weeks after ketamine treatment was -0.04. This is an effect size of 0.47. The effect size for change in limbic FC was very similar, with within-subject SD = 0.072 and FC change after ketamine = -0.034. Given that this effect is measured on subjects was 12 minutes of data per imaging session, this is a lower limit of expected effect size. Based on this effect size (0.47), in order to have an 80% power of finding an effect with an error probability q <0.05 using a two-tailed t-test, we would need a total sample size of 38 fcMRI scan (19 before psilocybin, 19 after) in a single subject.<br><br>Optimistic sample size calculation for single subject statistics: The above power calculation is based on group-averaged data and standard regions of interest. By testing each hypothesis within-subject, we are able to substantially increase power by removing neurobiological variability (Gordon et al., 2017). It also gives the ability to more clearly identify individual variability in the effects of psilocybin. To our knowledge, only one published study has used this approach to examine the effects of temporary arm immobilization (casting) on brain plasticity (Newbold et al., 2020). They had N=3 subjects visit for 42-46 consecutive days and they acquired 30 minutes of resting fcMRI data per visit. They found, within 24 hours of casting, a very large effects of motor connectivity (left to right M1), with FC change of -0.23, -0.86, and -0.61, and in their three subjects (Fig. 3). Baseline within-subject motor FC variance was 0.0093, 0.0025, and 0.0058 for the same subjects. This yielded effect sizes of 2.4, 17.2, and 8.0, respectively. Assuming the smallest effect size of 2.4, we would require 5 MRI sessions per condition to have a 95% probability (power, 1-β err prob) to detect a significant effect.<br><br>While estimation of individual effects would be ideal, the primary endpoint will use a mixed-effects model in order to assess group effects. That statistical power of the primary endpoint should be approximately √N higher than the single subject t-test described above. Therefore, using the conservative effect size above, in order to have an 95% power of finding the hypothesized persisting effect of psilocybin at the group level, with an error probability a <0.05, we would need N=5 subjects. |
| Data exclusions | As stated in the methods, "One participant (P2) was not able tolerate fMRI while on psilocybin and had trouble staying awake on numerous fMRI visits after psilocybin and was excluded from analysis (except for data quality metrics in Extended Data Figure 1)." Otherwise, predetermined motion exclusion criteria were used to exclude motion-confounded fMRI data (see methods).<br><br>ABCD: 8,479 participants were selected as the participants with data available on stimulant use in the last 24 hours and at least 8 minutes of low-motion data, a pre-established criterion. |
| Replication | Experimental findings were replicated both within the original study cohort (4/7 participants returned 6-12 months after completion for a replication protocol) and across independently collected datasets from outside institutions (ABCD, Imperial College Psilocybin/LSD datasets) |
| Randomization | This was a randomized cross-over design. Participants underwent imaging during drug sessions with psilocybin (PSIL) 25mg, or methylphenidate (MTP) 40mg as well as non-drug imaging sessions. Randomization allocation was conducted via REDCap which displays randomization assignment to research team members who prepare study materials including drug or placebo but otherwise have no contact with participants. |
| Blinding | Participants and study staff (except for the study statistician running the randomization, and the study staff putting study medication into the capsule and bottle) were blinded to drug order. Active drug and placebo were in matching capsules. |

# Reporting for specific materials, systems and methods

We require information from authors about some types of materials, experimental systems and methods used in many studies. Here, indicate whether each material, system or method listed is relevant to your study. If you are not sure if a list item applies to your research, read the appropriate section before selecting a response.

## Materials & experimental systems

| n/a | Involved in the study |
|-----|----------------------|
| ☒ ☐ | Antibodies |
| ☒ ☐ | Eukaryotic cell lines |
| ☒ ☐ | Palaeontology and archaeology |
| ☒ ☐ | Animals and other organisms |
| ☐ ☒ | Clinical data |
| ☒ ☐ | Dual use research of concern |
| ☒ ☐ | Plants |

## Methods

| n/a | Involved in the study |
|-----|----------------------|
| ☒ ☐ | ChIP-seq |
| ☒ ☐ | Flow cytometry |
| ☐ ☒ | MRI-based neuroimaging |

# Clinical data

Policy information about clinical studies

All manuscripts should comply with the ICMJE guidelines for publication of clinical research and a completed CONSORT checklist must be included with all submissions.

| | |
|---|---|
| Clinical trial registration | NCT04501653 |
| Study protocol | Abbreviated study protocol is avaiable at https://clinicaltrials.gov/study/NCT04501653. Full study IND protocol is available on request. |
| Data collection | Healthy young adults (N = 7, 18-45 years) were enrolled between April 2021 and March 2023 in a randomized cross-over precision functional brain mapping study at Washington University in Saint Louis. |
| Outcomes | Primary Objective (Predefined in IND protocol):  Our overall goal is to use a Functional Connectivity (numerous visits, very long scans to produce individual connectomes) to examine the effects of psilocybin on cortical and cortico-subcortical brain networks that could explain its rapid and sustained behavioral effects. 1. Persisting changes Functional Connectivity [Time Frame: 2 weeks], measured with neuroimaging procedures outlined in the study methods.  Secondary Objective: Examine changes in blood flow, brain activity, and connectivity immediately following psilocybin. We will use methylphenidate and baseline fMRIs as control conditions. 1. Acute changes in functional connectivity and the hemodynamic response to neural activity [Time Frame: Immediate] will be measured with neuroimaging as outlined elsewhere in this protocol. 2. Participants will report a Mystical Experiences with psilocybin [Time Frame: 2-week], measured using Persisting Effects Questionnaire. |

# Plants

| | |
|---|---|
| Seed stocks | *Report on the source of all seed stocks or other plant material used. If applicable, state the seed stock centre and catalogue number. If plant specimens were collected from the field, describe the collection location, date and sampling procedures.* |
| Novel plant genotypes | *Describe the methods by which all novel plant genotypes were produced. This includes those generated by transgenic approaches, gene editing, chemical/radiation-based mutagenesis and hybridization. For transgenic lines, describe the transformation method, the number of independent lines analyzed and the generation upon which experiments were performed. For gene-edited lines, describe the editor used, the endogenous sequence targeted for editing, the targeting guide RNA sequence (if applicable) and how the editor was applied.* |
| Authentication | *Describe any authentication procedures for each seed stock used or novel genotype generated. Describe any experiments used to assess the effect of a mutation and, where applicable, how potential secondary effects (e.g. second site T-DNA insertions, mosiacism, off-target gene editing) were examined.* |

# Magnetic resonance imaging

## Experimental design

| | |
|---|---|
| Design type | Stuctural sequences, resting state, and event-related task |
| Design specifications | Participants were scanned roughly every other day over the course of the experiment (Extended Data Fig. 1). Imaging was performed at a consistent time of day to minimize diurnal effects in functional connectivity. Neuroimaging was performed on a Siemens Prisma scanner (Siemens, Erlangen, Germany) in the neuroimaging labs (NIL) at the Washington University Medical Center.  Perceptual fMRI task: Participants also completed a previously validated event-related fMRI task. This was a suprathreshold auditory-visual matching task in which participants are presented with a naturalistic visual image (duration 500 ms) and coincident spoken English phrase and are asked to respond with a button press to indicate if the image and phrase are 'congruent' (for example, an image of a beach, and the spoken word beach) or 'incongruent'. |

Both accuracy and response time of button presses were recorded. Each trial was followed by a jittered inter-stimulus interval optimized for event-related designs. Two task fMRI scans were completed during a subset of imaging sessions. Task fMRI scans employed the same sequence used in resting fMRI, included 48 trials (24 congruent, 24 incongruent), and lasted a total of 410s. In analyses, the two task scans were concatenated to better match the length of the resting-state fMRI scans.

| | |
|---|---|
| Behavioral performance measures | Performance was measure during the perceptual fMRI task. |

## Acquisition

| | |
|---|---|
| Imaging type(s) | Structural (T1-w and T2w), Diffusion, Functional |
| Field strength | 3.0T |
| Sequence & imaging parameters | MRI pulse sequences used to acquire the data are provided at https://gitlab.com/siegelandthebrain1/Psilocybin_PFM/-/blob/main/NP1161_MRI_sequence.pdf.<br>Structural scans (T1w and T2w) were acquired for each participant at 0.9 mm isotropic resolution, with real-time motion correction. Structural scans from different sessions were averaged together for the purposes of Freesurfer segmentation and nonlinear atlas registrations.<br>For functional scans, to capture high resolution images of blood oxygenation level-dependent (BOLD) signal, we used an echo-planar imaging sequence with 2mm isotropic voxels, multi-band 6, multi-echo 5 (TEs: 14.20 ms, 38.93 ms, 63.66 ms, 88.39 ms, 113.12 ms)90, TR 1761ms, flip angle = 68 degrees, and in-plane acceleration91 (IPAT/grappa) = 2. This sequence acquired 72 axial slices (144mm coverage). Each resting scan included 510 frames (lasting 15:49 minutes) as well as 3 frames at the end used to provide estimate electronic noise.<br>Every session included at least two 15-minute resting-state fMRI scans during which participants were instructed to hold still and look at a white fixation crosshair presented on a black background. Head motion was tracked in real time using Framewise Integrated Real-time MRI Monitoring software (FIRMM)92 . An eye-tracking camera (EyeLink, Ottawa) was used to monitor participants for drowsiness. |
| Area of acquisition | Whole-brain |
| Diffusion MRI | ☐ Used ☒ Not used |

## Preprocessing

| | |
|---|---|
| Preprocessing software | FSL 6.0 software tools used: FAST, Eddy, Topup, DTIFit, FEAT<br>Freesurfer versions 5.0, 5.3, and 6.0, recon-all pipelines for brain segmentation<br>Connectome workbench v1.0 and 1.5<br>4dfp tools (https://4dfp.readthedocs.io/)<br>Processing pipelines used:<br>https://github.com/DCAN-Labs/abcd-hcp-pipelines<br>https://github.com/DCAN-Labs/nhp-abcd-bids-pipeline<br>Smoothing kernels employed: from 2mm to 6mm FWHM in humans (geodesic on cortical surface) |
| Normalization | WU P1-P7: BOLD->T2 rigid body linear, T2->T1 rigid body linear, T1-> atlas nonlinear |
| Normalization template | MNI152 |
| Noise and artifact removal | The acquisition and preprocessing of data were optimised for noise and artifact removal. This included:<br>- Preprocessing of fMRI data included: 1) removal of thermal noise using NORDIC (a local PCA approach in which temporal components of an fMRI signal that are indistinguishable from Gaussian noise are eliminated)111; 2) compensation for asynchronous slice acquisition using sinc interpolation; 3) compute affine spatial registration of all volumes within a run; 4) elimination of odd/even slice intensity differences resulting from interleaved acquisition (debanding); 5) compute affine spatial registration across fMRI runs; 6) compute an run volume mean (of all low-noise volumes); 7) computation of field distortion on the basis of a spin echo field maps using FSL top-up131; and 8) gain field correction using FSL fast132 (computed on the run volume mean);<br>- Resampling in MNI152 2 mm3 atlas space was accomplished for all echoes in one step combining (i) motion correction; (ii) distortion correction; (iii) gain field correction; (iv) linear registration of average volumes across visits; and (v) non-linear MNI152 atlas registration via the fsl fnirt133. Optimal combination of echoes was then computed in MNI152 space using the weighted summation approach (as described in Posse et al114 equations 6 and 7). Finally, the voxel-wise intensities were adjusted (one scalar per run) to obtain a mode value of 1000 in the distribution of intensities summed over all voxels and volumes.<br>- Following cross-modal registration, data were passed through several additional preprocessing steps: (i) tissue-based regressors were computed based on FreeSurfer segmentation134; (ii) temporal filtering to retain frequencies in the 0.009–0.08Hz band; and (iii) frame censoring (iv) removal by regression of the following signals that contain spurious variance: (a) six parameters obtained by rigid body correction of head motion, (b) signal from white matter, ventricles and extra-axial sources of noise (nuisance regressors were also bandpass filtered to match timecourse frequencies). Where indicated, respiratory and pulse-oximetry traces were used to generate additional physiological regressors using the PhysIO software package135.<br>- Individualized cortical surfaces and subcortical volumes were generated for each participant's T1 MRI using FreeSurfer automated segmentation . Segmentation errors were manually corrected. Following preprocessing, BOLD data were sampled to each participant's individual cortical surface and subcortical volume using Connectome Workbench140.<br>- In an alternative analysis, multi-echo ICA (ME-ICA) denoising designed to isolate spatially structured T2*- (neurobiological; "BOLD-like") and S0-dependent (non-neurobiological; "not BOLD-like") signals was performed using a modified version of the |

"tedana.py" workflow (https://tedana.readthedocs.io/en/latest/). We found that this tool did not produce substantially different results from our processing pipeline.

| Volume censoring | High motion time-points were censored using a frame-wise displacement (FD) threshold of 0.3mm. |

## Statistical modeling & inference

| Model type and settings | Task fMRI was analyzed using a two-level approach. First solving a GLM for each session, second an ANOVA was used to test if evoked responses differed significantly between drug conditions (no drug, MTP, PSIL). fMRI data were preprocessed similar to resting data, with the exception of no nuisance regression and an FD threshold of 0.7 mm111. A generalized linear model was computed in two different ways: 1) vertexwise GLM, using a assumed hemodynamic response function to visualize the magnitude of task-evoked responses. 2) parcel-wise GLM, using a finite impulse response model to model evoked response for 19.37s after each trial. A set of a priori regions of interest (ROIs) relevant to the task were selected from the Gordon-Laumann parcellation. These included: left/right calcarine sulcus (V1), left/right auditory cortex (A1), left language (Wernicke's area), left hand knob, left angular gyrus, and right angular gyrus (default mode). Trial conditions (congruent, incongruent; button press, no button press) were collapsed to model a main effect of task. Additional demean and detrend terms and 6 movement parameters111 were added to the generate a general linear model (GLM). This GLM was solved to estimate beta weights separately for each task visit.<br><br>In level 2 analysis, a two-way ANOVA was conducted using the anovan function in MATLAB. This analysis allowed us to account for the effects of the drug (as a primary factor) as well as individual subjects (as a secondary factor). A p-value associated with the 'drug' factor of $P<0.05$, would indicate that the drug has a significant effect on evoked response. |

| Effect(s) tested | Main effects of psilocybin, effect of methylphenidate, and effect of pre- vs post-psilocybin, and interaction of task*psilocybin were tested on FC change, global normalized global spatial complexity, local global normalized global spatial complexity, anterior hippocampal FC change, heart rate, and respiratory rate were tested using a linear mixed effects model.<br><br>Linear mixed effects model takes advance of the nested levels in our multi-level precision functional mapping study design. Every scan was labeled on the following dimensions: Subject ID, MRI visit, task (task/rest), drug condition (pre-psilocybin, PSIL, MTP, post-psilocybin), and head motion (average framewise displacement, FD). rs-fMRI metrics (described below) were set as the dependent variable, drug (drug condition), task, FD (motion), and drug*task were defined as fixed effects, and Subject ID and MRI session were random effects.<br><br>Subjective experience was assessed for drug sessions using the mystical experience questionnaire (MEQ; see Supplementary methods). We applied a LME model across all drug sessions, similar to the one described above, but with MEQ total score as the dependent variable. Whole-brain FC change and framewise displacement (FD) were modeled as fixed effects, and participant was modeled as a random effect. The same model was solved using FC change from every vertex to generate a vertexwise map of the FC change versus MEQ. |

Specify type of analysis: ☐ Whole brain ☐ ROI-based ☒ Both

| Anatomical location(s) | MNI152 atlas was used to determined anatomical locations (e.g, MNI coordinates: anterior hippocampus loci -24, -22, -16 and 24,-18, -16 |

Statistic type for inference

(See Eklund et al. 2016)

FC change ('distance') was calculated at the vertex level to generate FC change maps and a linear mixed effects model (Eq. 1) was employed in combination with wild bootstrapping100,101 and threshold-free cluster enhancement (TFCE)86,102 to estimate P values for t-statistic maps resulting from the model (Fig. 1a-d, Fig. 4).

First, a FC change map was generated for every scan by computing, for each vertex, the average distance between its FC seedmap and the FC seedmap for each of that subject's baseline scans. Since each participant had multiple baseline visits, FC change was computed for baseline scans by computing distance from all other baseline scans (excluding scans within the same visit). This provided a measure of day-to-day variability. Second, the distance value was used as the dependent variable $y_{ij}$ in the LME model to generate a t-statistic. Third, a wild bootstrapping procedure was implemented as follows. A large number of bootstrap samples (B = 1,000) were generated using the Rademacher procedure101, where the residuals were randomly inverted. Specifically, a Rademacher vector was generated by randomly assigning -1 or 1 values with equal probability to each observation's residual. By element-wise multiplication of the original residuals with the Rademacher vector, bootstrap samples were created to capture the variability in the data.

For the observed t-statistic-map, and each bootstrap sample, the TFCE algorithm was applied to enhance the sensitivity to clusters of significant voxels or regions while controlling for multiple comparisons. The value of the enhanced cluster statistic, derived from the bootstrap samples, was used to create a null distribution under the null hypothesis. By comparing the original observed cluster statistic with the null distribution, P values were derived to quantify the statistical significance of the observed effect. The P values were obtained based on the proportion of bootstrap samples that produced a maximum cluster statistic exceeding the observed cluster statistic.

The combined approach of wild bootstrapping with the Rademacher procedure and TFCE provided method to estimate p-values for our multi-level (drug condition, subject, session, task) design. This methodology accounted for the complex correlation structure, effectively controlled for multiple comparisons, and accommodated potential autocorrelation in the residuals through the Rademacher procedure. By incorporating these techniques, association with psilocybin and other conditions were reliably identified amidst noise and spatial dependencies.

Task fMRI: ROI-wise t-values converted to Z-scores were used for inference on task evoked response.

| Correction | see above |

## Models & analysis

| n/a | Involved in the study |
|-----|----------------------|
| ☐ ☒ | Functional and/or effective connectivity |
| ☒ ☐ | Graph analysis |
| ☒ ☐ | Multivariate modeling or predictive analysis |

Functional and/or effective connectivity

For P1-P7 particpants, a vertex/voxelwise functional connectivity matrix was calculated from the resting-state fMRI data for each 15-minute scan as the Fisher-transformed pairwise correlation of the timeseries of all vertices/voxels in the brain.

For whole brain and region-of-interest (ROI) based analysis, we used 385 surface/subcortical regions of interest. For cortical regions, we used a parcellation and community assignments generated by Gordon & Laumann and colleagues. For subcortical regions, we used a set of regions of interest generated to achieves full coverage and optimal region homogeneity. A subcortical limbic network was defined based on neuroanatomy: amygdala, antero-medial thalamus, nucleus accumbens, anterior hippocampus, posterior hippocampu. These regions were expanded to cover anatomical structures (e.g. anterior hippocampus). To generate region-wise connectivity matrices, time courses of all surface vertices or subcortical voxels within a region were averaged. Functional connectivity (FC) was then computed between each region timeseries using Fisher z-transformed bivariate correlation.

The same ROI based approach was used on Imperial College London LSD/Psilocybin.

In the ABCD, parcel-wise group-averaged functional connectivity matrices were constructed for each participant by first calculating the parcel-wise functional connectivity within each participant as the Fisher-transformed pairwise correlation of the timeseries of all 385 regions on interest in the brain.

