## [Peer Review file · Nature]

Manuscript Title: Psilocybin desynchronizes human brain networks

Reviewer Comments & Author Rebuttals

Reviewer Reports on the Initial Version:

Referee #1 (Remarks to the Author):

This manuscript by Siegel et al. describes a well-designed (within-subject control, replication protocol) and ambitious set of experiments aimed at characterizing both the acute and long-term effects of Psilocybin (PSIL) on functional brain networks at the individual-subject level using a precision functional mapping and extended data acquisition strategy (18+ MRI visits over 6+ months). The paper is well-written, and the work is novel. The authors on this paper are leading experts on individual-focused fMRI, and the topic of this investigation is timely, and should be of interest to a wide audience of scientists, clinicians, and lay people.

My main concern is regarding the author's interpretation of the acute effects of PSIL. The authors report a dramatic disruption in the spatial organization of spontaneous BOLD fMRI signals (operationalized as "desynchronization"; Figures 1-4) in 4 of the 6 individuals at T1 and 3 of the 4 individuals scanned at T2 (Extended Data Figure 2). To my eyes, this effect (which is illustrated most clearly in the Supplementary Video) has all the hallmarks of an artifact ("speckling", tightly mixed and extreme positive and negative values, most pronounced at the edges of brain such as the operculum and in low signal regions). Because this finding and related follow-up analyses represent the bulk of the manuscript's focus (4 of the 5 main text figures), it is important that the author's thoroughly rule out the alternative explanation – that the effect is either a direct or indirect (i.e., arising from a complex interaction with the denoising procedures used) consequence of increased head motion (Extended Data Figure 1c) or physiological signals (Supplemental Figure 3) when on PSIL. Note that this critique does not apply to the author's findings on the long-term effects of PSIL on cortico-hippocampal FC (the findings reported in Figure 5), which appear to be solid, and potentially clinically relevant.

Below, I unpack by concern regarding the author's interpretation of the acute effects of PSIL as "desynchronization" in more detail and list what I would need to see in order to be convinced that this effect is not an artifact. In addition, I have several technical and relatively minor critiques and clarifications, which should be more straight forward to address.

Major concerns

Artifacts. Most of the results and visualizations (with the exception of the Supplementary Video) presented are several steps away from the data itself. This high level of abstraction is natural for a paper like this, but it also makes it difficult to gauge if an artifact is baked into the analyses (which is my strong intuition after watching the Supplementary Video). I am requesting that the author's provide the following visualizations to help determine if artifact(s) in their data are contributing the FC change and desynchronization findings.

1) “Grayplots” (e.g., Power 2017 Neuroimage, also sometimes referred to as carpet plots) for each scan, condition (No Drug, PSIL, MTP), and subject. These grayplots must include the head motion traces and physiology traces (respiration, pulse oximeter). Separate grayplots stacked underneath and temporally aligned with the head motion and physiological data) are needed for each stage of data processing (i.e., minimally preprocessed with and without NORDIC, denoised with and without PhysIO, etc.). These images can be converted into a supplementary video that would help a reader form judgements about whether the non-specific effects of PSIL (changes breathing or heart rate, and increased head motion) are either not being removed fully or interacting with specific preprocessing or denoising procedures (I am particularly concerned about NORDIC and PhysIO) in unintended ways. Note that the fMRI signals within each grayplot must be demeaned.

2) Additional Supplementary Videos for all subjects (including replication visits). These videos should capture each data processing stage (minimally preprocessed, denoised with and without PhysIO) and condition (no drug, PSIL). The goal is to see if the “speckling” pattern is introduced after a certain preprocessing or denoising step. Again, fMRI signals mapped to cortical surface should be demeaned.

Individual FC change maps. Figure 1A shows a large and global FC change (on average across the 6 subjects with PSIL scans) when on PSIL. Extended Data Figure 2 shows that this effect is observed in 4 of the 6 subjects with PSIL scans, and the 2 subjects (P1, P3) that did not exhibit the effect at T1 do at T2 (the replication protocol). Some explanation for why P1 and P3 show a large FC change at T2 (but not T1) is needed. For example, was their subjective experience or on PSIL different across the time points? Extended Data Figure 1E shows group-average scores on the Mystical experience questionnaire, but not for individual subjects at the different time-points, and this would be useful to help answer this question. Or was the amount of head motion or changes in some aspects of their physiology (breathing, heart rate) different at T1 vs T2?

Supplemental Figure 1. The effect of global signal regression on the magnitude of FC change on PSIL (vs no drug) is pronounced (Supplemental Figure 1E; distance drops ~40% from 4.58 with no GSR to 2.77 with GSR). I raise this point because it is suggestive of increased global signal fluctuations (which are primarily driven by changes in respiration, but also heart rate, modulating cerebral blood flow or volume and in turn the fMRI signal) when on PSIL, and consistent with my concern raised earlier about the role that altered physiology could have on the fMRI signals.

Opting to not perform multi-echo denoising. The authors collected multi-echo fMRI data, but did not perform multi-echo denoising, which is curious as it is particularly well-suited for this kind of study where the effect of interest (PSIL vs. no drug) will covary with various kinds of artifacts (increased head motion, physiological artifacts). The multi-echo denoising approach is unique in its ability to disentangle BOLD and non-BOLD (artifactual) signals (see Kundu et al. 2017 Neuroimage for review). Can the authors confirm that the effects reported in Figures 1-4 remain when using multi-echo ICA denoising (“ME-ICA”; currently implemented within the Tedana python library; <https://github.com/ME-ICA/tedana>) with or without subsequent global signal regression, instead of regressing the motion parameters, signals from nuisance compartments (CSF, WM), and the physiological recordings via the PhysIO toolbox. If the main findings (specifically, Figure 1A, Figure 3A-C, Extended Data Figure 2A and Extended Data Figure 3A) remain when using ME-ICA or ME-ICA + GSR alone (and to be clear, I mean without NORDIC, or regression of physiology or any other

nuisance variables, e.g., motion parameters), I would be satisfied that that the FC change and desynchronization findings are not artifactual or introduced by their particular denoising procedures.

Technical concerns & minor issues

Auditory-visual matching task. Was this task was administered before or after the resting-state scan? Is it possible that the reduced FC change (Figure 4A) is due to subjects acclimating to the drug?

Physiology. P1 and P3 had only “partial” physiological recordings. It is not clear what this means. For example, did they not have any physiology recorded during their T1 PSIL scans? Coincidentally, these are the two subjects that did not show the effect at T1 but did at T2. I am trying to gauge the likelihood that regression physiological recordings (which are likely more “erratic” when subjects are on PSIL) could introduce the kinds of effects reported in this paper.

Subject IDs. The subject identifiers in Supplemental Figure 3 to not match those in the main text. This information is needed to help understand if the subjects showing the strongest effects in Extended Data Figure 2 are also individuals with largest changes in breathing or heart rate. Also, there are 8 total subjects in Supplemental Figure 3, but only 7 reported elsewhere. Please clarify this discrepancy.

Multi-echo fMRI acquisition and processing. The fMRI processing description in supplement do not adequately describe how the multi-echo fMRI data were handled.

For example, were the preprocessing steps currently listed (correcting for odd/even slice intensity differences, whole brain intensity normalization, removal of thermal noise via NORDIC) are applied before or after the optimal combination of echoes. This is important information because for the multi-echo modeling (fitting T2* decay at each voxel to obtain weights for optimal combination) to be valid, one must avoid rescaling values within the different echoes in a manner that does not maintain the relative differences in signal intensity.

Treatment guess. In the supplemental methods and results section, it is reported that subjects were asked to guess if they had received PSIL or MTP. I do not see any data in the main text or supplement showing what percentage of subjects correctly guessed PSIL vs MTP.

Referee #2 (Remarks to the Author):

In this study, Siegel and colleagues studied acute and persistent changes in brain network organization after a single dose of psilocybin. They found that psilocybin changes functional connectivity 3 times more than methylphenidate. In particular, psilocybin led to desynchronization of brain networks throughout association cortex, with effects strongest in the default network. Performing a perceptual task reduced psilocybin-driven FC changes and desynchronization. Psilocybin induced persistent decreases in functional connectivity between the anterior hippocampus and cortex. Overall I think this is a very strong study.

Major comments

1) Maybe I am misreading the supplement, but the main text states that “Psilocybin-associated FC change was greatest in the default mode network ($P_{spin} < 0.001$; $P_{spin} > 0.05$ for all other networks; Fig. 1e)”. I assume this is only referring to the cortex since the spin test was used? However, according to Supplementary Figure 1b, it seems that during PSIL1 (which is presumably first psilocybin session), the largest cortical FC change was not DMN, although largest subcortical FC changes do correspond to DMN (Supplementary Figure 1c)? On the other hand, the largest cortical FC change during PSIL2 is DMN (Supplementary Figure 1b). I am confused by this inconsistency.

2) “Participants were required to have had at least one previous lifetime psychedelic exposure (e.g., psilocybin, mescaline, ayahuasca, LSD)” - What is the rationale for this?

3) I found the description of the LME model hard to follow. For example, what does “i-th group” refers to? Is it referring to the i-th subject? Another example is whether “j-observation” refers to the “j-th visit” or something else? Since there are 4 drug conditions (“pre-psilocybin, PSIL, MTP, post-psilocybin”), are these coded using 3 one-hot vectors or something else?

4) Furthermore, I am also not sure about the asymmetry in which PSIL and MTP was treated. For example, drug condition were split into “pre-psilocybin, PSIL, MTP, post-psilocybin”, but there was no pre-MTP and post-MTP. I am not sure how the asymmetry influences the result. For example, if drug conditions were coded as “pre-MTP, PSIL, MTP, post-MTP” instead, how will the results (Figure 1) change?

5) Page 4 (word document) - “At the group level, psilocybin caused large acute FC change across most of the cerebral cortex ($P < 0.05$ based on linear mixed effects model and permutation testing)” - Unclear how and what permutations were performed for the LME modeling. There was a lot of focus on the wild bootstrap in the methods section, but I don’t see an explanation of the permutation test. What was permuted?

6) “distribution of these effects closely matched PET-based maps of 5HT-2A receptor density (serotonin 2A) (Fig. 3e)” -> Is there any quantification of the similarity?

7) In Extended Data Figure 5, we actually see increased FC between many pairs of brain regions (many more red color than blue color), especially involving association networks. Given the

increased FC, how do we reconcile this with the “desynchronization” narrative, which suggests a decrease in FC?

8) I also think that portion of Extended Data Figure 5 should be brought into the main text. Right now, the main figures show some rather derived (downstream) measures like multidimensional distance (Figure 2), overall FC changes at each location (Figure 1) and normalized global spatial complexity (Figure 3) that do not provide any insights into intra and inter-network FC changes. Extended Data Figure 5 is important because it shows the underlying change in brain FC that give rise to the “derived (downstream)” measures.

9) If I understand correct, whole brain FC was computed using group-level ROIs instead of individualized ROIs? This is probably ok, but seems a bit of a waste given the amount of data collected per participant.

10) “Normalized global spatial complexity” should yield one number per brain. I am not sure how the spatial maps (Figures 3d and Figure 3e) were obtained.

11) Perhaps the most important question the study seeks to address is the persistent effects of psilocybin. Yet, I am worried about circularity/double dipping when looking at persistent changes due to psilocybin. More specifically, in the supplemental material, it states that “This analysis indicated that the limbic system was the only system to show FC change after psilocybin that exceeded chance ($P < 0.05$, uncorrected)... Following this observation, we probed FC change in the constituent parts of the limbic system. We separated the limbic system into five bilateral regions of interest - anterior hippocampus, posterior hippocampus, ventromedial thalamus, amygdala, and nucleus accumbens – and repeated the FC change permutation testing approach 500 times. Here, the anterior hippocampus was the only region for which pre- and post-drug FC change exceeded all 500 label permutations.” - Since was no correction for multiple comparisons in the first round of “system selection”, this should result in potential double dipping / circularity in the second round of analysis focusing on the regions within the limbic system.

Minor comments:

1) In Supplementary Figure 1b (second column), are you actually showing dissimilarity rather than “similarity: $z(r)$ ”? Because if it’s similarity, then shouldn’t larger values indicate less change, so I would expect a rough reverse ordering between first column (Euclidean distance) and second column (Similarity: $z(r)$)? But ordering in the two columns seem somewhat consistent.

2) Figure 1 captions: “Euclidian “ -> “Euclidean”

3) Page 16 (word document) - “To compensate for the implementations of this LME model on multiple rs-fMRI-related dependent variables. differences were highlighted when $P < 0.005$. “ -> “To compensate for the implementations of this LME model on multiple rs-fMRI-related dependent variables, differences were highlighted when $P < 0.005$. “

3) "Extended Data Figure 5" - "Right shows the effect of psilocybin" - should be bottom left. "Bottom left shows effect of methylphenidate" - should be right.

4) Page 19 (word document) - Under "Normalized global spatial complexity" in the methods section, "We used a n approach previously" should be "We used an approach previously"

5) Page 20 (word document) - "We assessed change in anterior hippocampus 'FC change' per- versus post-psilocybin" -> Should be "pre-" not "per-"

6) The extended data figures do not match the ordering of the main text.

Referee #3 (Remarks to the Author):

In this manuscript by Siegel, et al the authors expand on previous work that has examined acute changes in signal power and functional connectivity (FC) during acute psychedelic administration. They do this by longitudinally imaging subjects dosed with psilocybin using a method they previously developed, precision functional mapping (Gordon, et al 2017), that utilizes extensive within-individual data collection to characterize brain function in specific individuals. Six participants were dosed psilocybin and methylphenidate 1-2 weeks apart and underwent several MRI sessions (~18/participant). The main findings of the paper are that: 1) psilocybin causes profound changes in FC in cortex and in subcortical areas, and mostly in the default mode network (DMN) 2) reduced segregation between the DMN and other networks explained the variance between psilocybin and other groups, and that this held consistent for other data analyzed with in humans dosed with other psychedelics 3) that brain activity across regions is overall less synchronous with psilocybin 4) participation in a perceptual task weakens these effects and 5) some effects are persistent over time.

The longitudinal precision functional mapping approach is potentially powerful here, as it should allow for a detailed understanding of individual differences with psilocybin use. In addition, longitudinal imaging with psychedelics seems important given long lasting effects on humans. With this approach, however, most of the analyses presented in the paper focused on group differences, rather than individual ones. The main finding in both figures 1 and 2, corroborates previous results, that psilocybin desynchronizes cortical and subcortical activity and is therefore not entirely novel. In figure 3, I am not sure the authors' conclusion that they have presented evidence for 'grounding.' What they have shown is that at this dose of psilocybin, participants can still engage task associated networks, when performance is maintained at a high rate but this does not seem entirely surprising – higher doses of the drug might be helpful here to prove this point. Finally, and what is perhaps the most novel finding of the paper is the persistent decrease of FC of the anterior hippocampus and DMN. The authors assert that this their study is the first to study persistent effects of psilocybin however I can think of at least 1 other (Barrett, et al 2020).

Minor points:

Discussion is repetitive and includes a lot of points that were also included in the introduction. It might be good to have more thorough citations and review of literature in the introduction and use the discussion for interpretation of their results and next steps/outlook.

The layout of the figures and the reference to them in the text is often confusing. This is especially the case for figure 5. It was unclear to me what was plotted in panel 5b – and I could not find reference to this in the main text. It seems the authors are plotting numerous scans of each patient of the FC connectivity of the area highlighted in figure 5a? It was also unclear why this is going up in 5b but is a negative value in 5c. A clearer description of the methods used to quantify FC change in these panels as well as an explicit description of what is plotted in each subpanel.

Referee #4 (Remarks to the Author):

In this study, Siegel et al. use precision functional mapping to study within-subject acute and persistent changes in functional connectivity (FC) induced by 25 mg of oral psilocybin versus a psychoactive placebo (40 mg of oral methylphenidate) in 7 healthy adults (6 with useable fMRI data with psilocybin) administered in a blinded cross-over fashion. Their resting-state fMRI data collection protocol is rigorous and includes: (1) repeated scanning over multiple days to collect hours of data for establishing stable FC baselines for each participant, (2) scanning at approximately the same time each day to reduce diurnal variation in FC, (3) multi-echo data collection to reduce head motion, (4) real-time monitoring software, (5) field map unwarping, and (6) cardiovascular as well as respiratory monitoring to remove physiological noise from their fMRI data.

To characterize the acute effects of psilocybin, the team scanned participants during peak drug intoxication. During the scan, subjects passively stared at a fixation cross for 30 to 120 minutes and also completed an auditory-visual matching task. This was followed by hours of resting-state scanning collected in the days following psilocybin administration to evaluate for any persistent effects on FC. 4 of the participants also completed an open-label replication protocol with psilocybin 6-12 months after the blinded portion of the study.

Using this approach, Siegel et al. made a number of findings: (F-1) psilocybin acutely disrupts normal FC patterns globally, but most pronounced, in the default mode network (DMN), (F-2) psilocybin acutely reduces segregation between the DMN and other networks (fronto-parietal, dorsal attention, salience, and cingulo-opercular) that are normally anti-correlated with it, (F-3) psilocybin-associated disruptions in FC are spatially associated with 5HT_{2A} receptor density in humans, (F-4) individuals are still able to effectively engage task-positive networks on 25 mg of oral psilocybin and their FC patterns appear less de-synchronized while engaged in a task compared to rest, and (F-5) they observed persistent decreases in functional connectivity between the anterior hippocampus and DMN for up to 3 weeks following psilocybin.

This was a well-conceived resting-state fMRI study with oral psilocybin in healthy volunteers and serves to corroborate a number of important findings that have already been made and reported in the literature. For example, Preller et al. (*Biological Psychiatry* 2020;88:197-207)¹, Mason et al. (*Neuropsychopharmacology* 2020;45:2003-11).², and Madsen et al. (*European Neuropsychopharmacology* 2021;50:121-32)³ have previously demonstrated acute-phase findings (F-1), (F-2), and (F-3) with oral psilocybin in healthy volunteers. Persistent effects on FC in the weeks and months following oral psilocybin administration in healthy volunteers has also been examined and reported on by Barrett et al. (*Scientific Reports* 2020;10:2214)⁴ and McCulloch et al. (*Journal of Psychopharmacology* 2022;36:74-84).⁵ These papers should be discussed in the manuscript and serve as points of comparison to the findings reported here.

While this work is an important replication, it is not adequately novel to justify publication in *Nature*.

Moreover, I would like to provide some comments that I hope will clarify and strengthen the manuscript.

Major Comments:

- Supplementary Methods (lines 112-114): Low-pass filtering the motion time courses at 0.1 Hz prior to computing framewise displacement allows data collected during excessive head motion to enter your analyses and can lead to spurious findings. It would be prudent to repeat the main analyses of the manuscript without the low-pass filter (i.e. after removing all data containing excessive head motion) and ensure that similar results are still obtained. The results without the low pass filter should be reported.
- Supplementary Methods (lines 169-170): It is not clear why the impulse response was modeled 19.37 seconds after each trial. Moreover, it appears that all other task-related variance was unmodeled – specifically stimulus duration, response, conflict, error, etc. Without regressing out the full effect of the task, it is not possible to appreciate the underlying FC.
- Main Text (lines 222-228): I disagree with the conclusion being drawn here. At 25 mg of oral psilocybin most individuals maintain a connection to reality as is evidenced by the 100% performance on the auditory-visual matching task. What I believe the results are showing is that individuals are still able to effectively engage task-positive networks following ingestion of 25 mg of oral psilocybin, which appears as less de-synchronized brain activity compared to the brain at rest. I would not equate this to discovering a neurobiological basis for ‘grounding’.
- Expansion of Discussion: The manuscript would benefit significantly from an expanded discussion section that compares the results of this study to the work done by others who have studied the acute (Preller et al.1, Mason et al.2, Madsen et al.3) and persistent (Barrett et al.4 and McCulloch et al.5) effects of oral psilocybin in healthy volunteers.

Minor Comments:

- Summary (lines 35-36): The way this sentence currently reads implies that everyone experiences distortions in space-time perception and ego-dissolution on psilocybin. These subjective effects are dose dependent, and even at the highest doses, do not unanimously occur in all individuals who consume psilocybin.
- Summary (line 36): In addition to having persistent anti-depressant effects, psilocybin has been shown to have anti-addictive properties⁶ and more. Perhaps the term “transdiagnostic” better captures the effects of psilocybin?
- Summary (lines 37-39): Preller et al.1 demonstrated reduced global connectivity in associative brain regions in their resting-state fMRI study on the acute effects of oral psilocybin in healthy volunteers. Mason et al.2 and Madsen et al.3 also demonstrated reduced DMN integrity in their resting-state fMRI studies on the acute effects of oral psilocybin in healthy volunteers. These studies should be cited given their similarities to the present work.
- Summary (lines 39-40): It is still unclear how brain network changes underlie the variety(ies) of subjective psychedelic experience(s) (ego dissolution, sense of unity, feelings of love, synesthesia, oceanic boundlessness, etc.)
- Summary (lines 40): There have been some studies evaluating enduring effects on functional connectivity in healthy volunteers following oral psilocybin administration. For example, Barrett et al.4 scanned 12 healthy volunteers 1 week and 1 month after oral psilocybin ingestion and McCulloch et al. 5 scanned 10 healthy volunteers 1 week and 3 months after oral psilocybin ingestion.

- Summary (line 47): The anterior hippocampus is not considered part of the default mode network.
- Summary (48-50): You have demonstrated that individuals are still able to activate task-positive networks on 25 mg of oral psilocybin. I would not equate this to discovering a neurobiological basis for grounding.
- Summary (52-53): It is pure speculation that persistent reduction in hippocampal-DMN connectivity represents a mechanistic correlate of psilocybin's anti-depressant effect. I think this should be removed from the summary.
- Main Text (line 59): Consider citing Bogenschutz et al.6 as an example of psilocybin's anti-addictive potential.
- Main Text (line 69): Change network to networks
- Main Text (line 72): Remove citation 28 – Timmerman et al. studied DMT, not psilocybin.
- Main Text (line 73-74): This is not true – there have been at least two studies evaluating persisting effects of oral psilocybin on FC in healthy volunteers, namely Barret et al.4 and McCulloch et al.5
- Main Text (line 75-76): The anterior/middle hippocampus are not considered core parts of the default mode network. However, there is some evidence suggesting they are functionally connected.
- Main Text (lines 109-111): "default mode network parts of the thalamus, basal ganglia, cerebellum, and hippocampus" is very confusing. Most imagers and neuroscientists consider the DMN to consist of 3 parts: 1. medial prefrontal cortex, 2. posterior cingulate cortex / precuneus, and 3. inferior parietal lobule. It might be more clarifying to say something like -- regions of subcortical structures that have been shown to be functionally connected with the DMN.
- Main Text (lines 114-115): As above, "In the cerebellum the largest FC changes were seen in the default mode network" is very confusing. Consider rephrasing.
- Main Text (lines 119): Can you clarify high head motion = 1.5.
- Main Text (line 123): It might be helpful to define the DMN since you've been referring to many structures in the paper that aren't traditionally considered part of this network.
- Main Text (line 192): Preller et al.1 have already shown that psilocybin induced disruptions in FC are spatially associated with 5HT2A receptor density in humans and this manuscript should be cited in the main text.
- Main Text (line 202): I believe you mean to write Fig 3b-c not Fig 3b-d?
- Main Text (line 204): I believe you mean to write Fig 3d not Fig 3b-d?
- Main Text (line 205): I believe you mean to write Fig 3e not Fig 3d?
- Main Text (line 206): I believe you mean to write Fig 3f not Fig 3e?
- Main Text (line 274): Atypical cortico-hippocampal connectivity has been associated with affective symptoms in patients with multiple sclerosis. The relevant patient population should be mentioned.
- Main Text (line 294): Consider changing the header of this section – it is unclear to me how the 2 paragraphs (lines 295-309) relate to the linking of acute and persistent effects.
- Main Text (line 300): Should this header read "linking acute and persistent effects"?
- Main Text (lines 321-322): Please add citations to corroborate these statements.
- Main Text (line 326): The DMN and hippocampus are also highly relevant to addiction and significant work is being done to study psilocybin for the treatment of alcohol abuse and other addictive disorders. It might be worthwhile to add a sentence or two expanding the discussion on the relevance of your finding to this line of work.
- Methods (line 382): Please provide some additional information as to how the multi-echo data was analyzed.
- Methods (line 393): Add a reference for the previously validated event-related fMRI task.

- Methods (line 400): Were stimuli presented in random or pseudorandom order?
- Methods (line 409): Change areal to area
- Methods (line 413): Change achieves to achieve
- Methods (line 416): Change hippocampu to hippocampus
- Methods (line 420): Did you mean that you calculated the correlation between time series and then transformed each correlation using fisher z for group comparisons?
- Methods (line 440): Change advance to advantage
- Methods (line 442): Was there a post-MTP label?
- Methods (line 448): Please define γ – presumably this is the session/subject specific connectome?
- Methods (line 459): Change ‘.’ to ‘,’
- Methods (line 560-561): This sentence should be in the results, not the methods.
- References (line 631 and 651): Citation 3 and citation 14 are both Carhart-Harris et al. Psilocybin versus Escitalopram for Depression. Please consolidate them to a single citation.
- Extended Data Figure 1: Please ensure that the range of MRIs depicted in panel (a) matches the imaging visits in panel (b). For example, I don't see any participant who had 9 'after' MRI visits in panel (b) – P1 comes close at 8 'after' visits. I only see a max of 4 'between' MRI visits in panel (b) but the range in (a) says up to 5.
- Extended Data Figure 1: It would be helpful to be consistent with your nomenclature for days. For example, in panel (b) psilo = day 0, however, in panel (d) baseline 1 = day 0. Furthermore, based on panel (b) it appears that P1 has 8 'after' visits, but in panel (d) there are only 7 listed. Please clarify.
- Extended Data Figure 1: The open-label replication protocol is described as “included one scan each of baseline, psilocybin, and after drug” in the figure legend but in panel (d) participant P1 appears to have 2 baseline and 2 after scans. Please clarify.
- Extended Data Figure 1: It might make sense to depict individual-specific MEQ scores in panel (e) since the rest of the paper reports within-subject effects.
- Extended Data Figure 2: Add Euclidian distance to the color spectrum in panel (a) to clarify units.
- Extended Data Figure 2: Consider adding an asterisk to the FC change brain for P3 on MTP to explain why their change map looks more pronounced on MTP than PSIL1.
- Extended Data Figure 6: Consider re-labeling extended figure 6 to 4 to match the ordering presented in the manuscript. And visa-versa.
- Supplemental Table 1: Why is the age of each participant depicted as a 2- or 4-year range?
- Supplemental Table 1: Are you reporting baseline personality characteristics of each participant? Please clarify since Mini-IPIP was collected at 3 timepoints.
- Supplemental Table 1: I would clarify that '+' in MRI visits represents the additional scans done as part of the replication protocol.
- Supplemental Table 1: Why is the rsfMRI data with methylphenidate not reported?
- Supplemental Table 1: Why are 2 participants missing respiratory and pulse data?

Works Cited

- 1 Preller, K. H. et al. Psilocybin Induces Time-Dependent Changes in Global Functional Connectivity. *Biol Psychiatry* 88, 197-207 (2020). <https://doi.org/10.1016/j.biopsych.2019.12.027>
- 2 Mason, N. L. et al. Me, myself, bye: regional alterations in glutamate and the experience of ego dissolution with psilocybin. *Neuropsychopharmacology* 45, 2003-2011 (2020).

<https://doi.org/10.1038/s41386-020-0718-8>

3 Madsen, M. K. et al. Psilocybin-induced changes in brain network integrity and segregation correlate with plasma psilocin level and psychedelic experience. *Eur Neuropsychopharmacol* 50, 121-132 (2021). <https://doi.org/10.1016/j.euroneuro.2021.06.001>

4 Barrett, F. S., Doss, M. K., Sepeda, N. D., Pekar, J. J. & Griffiths, R. R. Emotions and brain function are altered up to one month after a single high dose of psilocybin. *Sci Rep* 10, 2214 (2020). <https://doi.org/10.1038/s41598-020-59282-y>

5 McCulloch, D. E. et al. Lasting effects of a single psilocybin dose on resting-state functional connectivity in healthy individuals. *J Psychopharmacol* 36, 74-84 (2022).

<https://doi.org/10.1177/026988112111026454>

6 Bogenschutz, M. P. et al. Percentage of Heavy Drinking Days Following Psilocybin-Assisted Psychotherapy vs Placebo in the Treatment of Adult Patients With Alcohol Use Disorder: A Randomized Clinical Trial. *JAMA Psychiatry* 79, 953-962 (2022).

<https://doi.org/10.1001/jamapsychiatry.2022.2096>

Author Rebuttals to Initial Comments:

Referees' comments:

Referee #1 (Remarks to the Author):

This manuscript by Siegel et al. describes a well-designed (within-subject control, replication protocol) and ambitious set of experiments aimed at characterizing both the acute and long-term effects of Psilocybin (PSIL) on functional brain networks at the individual-subject level using a precision functional mapping and extended data acquisition strategy (18+ MRI visits over 6+ months). The paper is well-written, and the work is novel. The authors on this paper are leading experts on individual-focused fMRI, and the topic of this investigation is timely, and should be of interest to a wide audience of scientists, clinicians, and lay people.

Thank you for these positive comments regarding the study design and aims, the novelty of our work, and writing quality!

My main concern is regarding the author's interpretation of the acute effects of PSIL. The authors report a dramatic disruption in the spatial organization of spontaneous BOLD fMRI signals (operationalized as "desynchronization"; Figures 1-4) in 4 of the 6 individuals at T1 and 3 of the 4 individuals scanned at T2 (Extended Data Figure 2). To my eyes, this effect (which is illustrated most clearly in the Supplementary Video) has all the hallmarks of an artifact ("speckling", tightly mixed and extreme positive and negative values, most pronounced at the edges of brain such as the operculum and in low signal regions). Because this finding and related follow-up analyses represent the bulk of the manuscript's focus (4 of the 5 main text figures), it is important that the author's thoroughly rule out the alternative explanation – that the effect is either a direct or indirect (i.e., arising from a complex interaction with the denoising procedures used) consequence of increased head motion (Extended Data Figure 1c) or physiological signals (Supplemental Figure 3) when on PSIL. Note that this critique does not apply to the author's findings on the long-term effects of PSIL on cortico-hippocampal FC (the findings reported in Figure 5), which appear to be solid, and potentially clinically relevant.

Thank you so much for these helpful suggestions. We implemented all of them to 1) assess if potential sources of noise or variations in processing impacted the findings, 2) clearly illustrate measures of data quality (e.g. supplemental videos with gray plots for every fMRI session and movies of resting fMRI data for every subject), and 3) provide further clarity both in the description of methods (see revised "Resting-state functional MRI processing, and surface projection" in the supplemental materials). This included implementing ME-ICA as well as a number of other approaches to assess if the results reported might be explained by some confound. The analyses

presented below clearly and consistently point away from the primary results being the consequence of artifacts.

Below, I unpack by concern regarding the author's interpretation of the acute effects of PSIL as "desynchronization" in more detail and list what I would need to see in order to be convinced that this effect is not an artifact. In addition, I have several technical and relatively minor critiques and clarifications, which should be more straight forward to address.

Major concerns

Artifacts. Most of the results and visualizations (with the exception of the Supplementary Video) presented are several steps away from the data itself. This high level of abstraction is natural for a paper like this, but it also makes it difficult to gauge if an artifact is baked into the analyses (which is my strong intuition after watching the Supplementary Video). I am requesting that the author's provide the following visualizations to help determine if artifact(s) in their data are contributing the FC change and desynchronization findings.

The supplemental video from the initial submission showed a raw timecourse for P6 (Psilocybin) prior to nuisance regression, frame censoring, or any smoothing. Standard preprocessing steps remove this speckling (as is seen in the snapshot below from raw versus pre-processed timecourses). We apologize for not being clearer in the initial submission. As we describe below, we now include grayplots and physiological traces for every session and rsfMRI timecourses for MTP and PSIL sessions as supplemental videos.

snapshot from P6 with minimal preprocessing (prior to bandpass, nuisance regression and smoothing)

snapshot from P6 fully processed (bandpass filter, nuisance regression and smoothing)

R2R Figure 1. Single frame examples from timecourse videos before and after standard rsfMRI preprocessing.

1) “Grayplots” (e.g., Power 2017 Neuroimage, also sometimes referred to as carpet plots) for each scan, condition (No Drug, PSIL, MTP), and subject. These grayplots must include the head motion traces and physiology traces (respiration, pulse oximeter). Separate grayplots stacked underneath and temporally aligned with the head motion and physiological data) are needed for each stage of data processing (i.e., minimally preprocessed with and without NORDIC, denoised with and without PhysIO, etc.). These images can be converted into a supplementary video that would help a reader form judgements about whether the non-specific effects of PSIL (changes breathing or heart rate, and increased head motion) are either not being removed fully or interacting with specific preprocessing or denoising procedures (I am particularly concerned about NORDIC and PhysIO) in unintended ways. Note that the fMRI signals within each grayplot must be demeaned.

Grayplots (before and after bandpass filtering, nuisance regression and smoothing) for all data collected are now provided as a supplemental video in the revised submission. Since removal of thermal noise using the NORDIC algorithm was executed prior to any other registration/distortion correction/etc., visualizing gray plots before NORDIC would not be possible (as gray plots typically show voxel timecourses after alignment). However, as can be seen below further evaluation of NORDIC (R2R Figure 7), and data reprocessed using ME-ICA (R2R Figure 4) both indicate that NORDIC denoising is not adding any systematic confounds to the data.

2) Additional Supplementary Videos for all subjects (including replication visits). These videos should capture each data processing stage (minimally preprocessed, denoised with and without PhysIO) and condition (no drug, PSIL). The goal is to see if the “speckling” pattern is introduced after a certain preprocessing or denoising step. Again, fMRI signals mapped to cortical surface should be demeaned.

All requested videos have been included as supplemental videos in the revised manuscript.

The supplemental video from the initial submission shows a timecourse for P6 Psilocybin PRIOR TO nuisance regression, frame censoring, or any smoothing. After nuisance regression and gentle smoothing (4mm FWHM), the speckling pattern (which the reviewer astutely points out as a hallmark of artifact) is removed.

For the initial submission, we arbitrarily showed the timecourse for P6. P6 provided high quality data overall, but moved more or psilocybin. We now include videos for every subject comparing methylphenidate and psilocybin rsfMRI timecourses. As illustrated below, P1 showed no increase in head motion during methylphenidate or psilocybin (relative to no drug) – yet, as is apparent from the video and manuscript results, still showed dramatic changes in local and global synchrony and functional connectivity on psilocybin.

R2R Figure 2. Average head motion (FD) from every scan.

3) Individual FC change maps. Figure 1A shows a large and global FC change (on average across the 6 subjects with PSIL scans) when on PSIL. Extended Data Figure 2 shows that this effect is observed in 4 of the 6 subjects with PSIL scans, and the 2 subjects (P1, P3) that did not exhibit the effect at T1 do at T2 (the replication protocol). Some explanation for why P1 and P3 show a large FC change at T2 (but not T1) is needed. For example, was their subjective experience or on PSIL different across the time points? Extended Data Figure 1E shows group-average scores on the Mystical experience

questionnaire, but not for individual subjects at the different time-points, and this would be useful to help answer this question. Or was the amount of head motion or changes in some aspects of their physiology (breathing, heart rate) different at T1 vs T2?

We sincerely thank the reviewer for posing this question/hypothesis – it has directly improved the manuscript. Further exploration of individual differences revealed that, as the reviewer predicted, differences in the magnitude of psilocybin’s effects on brain connectivity were closely related to differences in ratings of psilocybin’s subjective effects (see new Fig. 1f-h). Of note, they were not strongly related to head motion (LME model predicting MEQ score: $T\text{-stat}_{FC\text{Change}} = 7.68, P = 3.5 \times 10^{-6}, T\text{-stat}_{FD} = -1.26, P = 0.23$).

New manuscript Fig. 1f-h.

Te

R2R Figure 3. FC Change, Mystical Experience Questionnaire (MEQ) score, and FD for all psilocybin doses (as in Extended Data Figure 2).

For P1, there is no noticeable difference in head motion between the first and second psilocybin dose (R2R Fig 1).

*note - P1 individual FC change maps for PSIL doses 1 and 2 were switched in Extended Data Figure 2 of the initial submission. The first dose of psilocybin had a larger subjective effect and larger FC change in P1. The swap only occurred in E.D. Figure 2 (notice that P1 psilocybin dose 1 has larger effects in Figure 2 and Extended Data Figure 3). We apologize for this error.

4) Supplemental Figure 1. The effect of global signal regression on the magnitude of FC change on PSIL (vs no drug) is pronounced (Supplemental Figure 1E; distance drops ~40% from 4.58 with no GSR to 2.77 with GSR). I raise this point because it is suggestive of increased global signal fluctuations (which are primarily driven by changes in respiration, but also heart rate, modulating cerebral blood flow or volume and in turn the fMRI signal) when on PSIL, and consistent with my concern raised earlier about the role that altered physiology could have the fMRI signals.

As inspection of Supplemental Figure 1d reveals, GSR did not substantially impact the psilocybin-associated changes reported in the manuscript. The reviewer is correct however that the magnitude of FC change is smaller following GSR. Though importantly, the effect of psilocybin is still much larger than the effects of MTP, head motion, or task. As we report in the supplement, the effects of psilocybin on pulse and respiratory rate were very well matched by the methylphenidate control (see supplemental Fig. 4 reproduced below). Further, the results are unchanged by PhysIO-based regression of potential respiratory confounds (R2R Fig. 6). Thus, we assert that the effect of GSR on the results is not merely a consequence of confound.

Supplemental Figure 4. Pulse and respiratory rates across conditions. Mean rate from continuous measurement during fMRI scans is given for each participant and all participants (average). Asterisks indicate rejecting the null that a given condition was not significantly different from baseline at $p < 0.05$, uncorrected. At the individual participant level, a non-parametric (Mann-Whitney U test) comparison was used to compare each condition (PSIL, MTP, Between, After) with baseline. At the group level, a linear mixed effects model was used, as described previously.

5) Opting to not perform multi-echo denoising. The authors collected multi-echo fMRI data, but did not perform multi-echo denoising, which is curious as it is particularly well-suited for this kind of study where the effect of interest (PSIL vs. no drug) will covary with various kinds of artifacts (increased head motion, physiological artifacts). The multi-echo denoising approach is unique in its ability to disentangle BOLD and non-BOLD (artifactual) signals (see Kundu et al. 2017 Neuroimage for review).

Can the authors confirm that the effects reported in Figures 1-4 remain when using multi-echo ICA denoising ("ME-ICA"; currently implemented within the Tedana python library; <https://github.com/ME-ICA/tedana>) with or without subsequent global signal regression, instead of regressing the motion parameters, signals from nuisance compartments (CSF, WM), and the physiological recordings via the PhysIO toolbox. If the main findings (specifically, Figure 1A, Figure 3A-C, Extended Data Figure 2A and Extended Data Figure 3A) remain when using ME-ICA or ME-ICA + GSR alone (and to be clear, I mean without NORDIC, or regression of physiology or any other nuisance variables, e.g., motion parameters), I would be satisfied that that the FC change and desynchronization findings are not artifactual or introduced by their particular denoising procedures.

We can confirmed that the acute effects reported in the manuscript remain when using multi-echo ICA denoising ("ME-ICA", implemented within the Tedana python library). We applied MEICA to all P1 scans and P1R psilocybin replication scans. In cases where automated component selection was flawed, we relied on manual reclassification of components. Outside of MEICA, all other elements of preprocessing were unchanged from the previously reported methods, and included mode 1000 normalization, bandpass filtering, projection to grayordinate space, spatial smoothing at 4mm FWHM, and our normal FC analyses reported in the paper. As requested by the reviewer, we did not do any additional nuisance regression (neither motion parameters nor signal-based regressors).

Results after Tedana ME-ICA (S1 + S1R)

Results after our PFM pipeline (S1 + S1R)

R2R Figure 4. Primary results remain unchanged by the ME-ICA pipeline. T-test comparing NGSC between psilocybin and no-drug scans: $P_{(Psilocybin, MEICA\ pipeline)} = 0.013$, $P_{(Psilocybin, our\ pipeline)} = 3 \times 10^{-7}$.

MEICA did not alter the main findings of the paper, indicating that previously unaccounted for noise is highly unlikely to drive the main effects in this manuscript. The magnitude of FC change (normalized to day-to-day change minus within-session change, as reported in paragraph 3 of the results) actually got larger (increased to 5.23). This is because within-session (scan-to-scan) FC variability increased considerably with ME-ICA (33.8 [our pipeline] vs 49.9 [ME-ICA]).

We chose not to use Tedana for this study because doing ME-ICA denoising (selection of which features of the data to include/exclude) separately run-by-run could introduce additional confounds into the repeated sampling intervention paradigm by compromising the ability to compare across runs. This concern is supported by the observation that within-session (scan-to-scan) variability was considerably higher in ME-ICA data. Moreover, ME-ICA often requires hand selection of which components to exclude, making the analyses non-deterministic and therefore more difficult to replicate.

Technical concerns & minor issues

6) Auditory-visual matching task. Was this task was administered before or after the resting-state scan? Is it possible that the reduced FC change (Figure 4A) is due to subjects acclimating to the drug?

This is something that we considered when we first observed the drug-task interaction phenomenon. We did not perfectly counterbalance the order of task/rest across subjects. For most participants, the task scan occurred following the two resting scans - at roughly 2-hours after ingestion, which should not be past t_{max} for oral psilocybin. However, we realized that this was suboptimal part way through data collection - and so we added additional rest scans after task in some subjects (e.g. P7, P1R) to rule out that any effect of task might be confounded by order effects. For example, in P7, the task session was completed after rest 1 & 2 and before rest 4, 5, & 6 (R2R Figure 5).

R2R Figure 5. Whole-brain FC change from every scan.

The task session still had a lower FC change score than either rest scan preceding/following it. This is illustrated in the new Supplemental Fig. 2, and highlighted below.

We have included the new Supplemental Fig. 2 and added the following text to acknowledge this limitation: *While the order of rest and task scans was not perfectly counterbalanced across subjects, it is notable that in some psilocybin sessions (e.g. P7, P1R) task scans had lower FC change than either of the flanking rest sessions (Supplemental Fig. 2).*

7) Physiology. P1 and P3 had only “partial” physiological recordings. It is not clear what this means. For example, did they not have any physiology recorded during their T1 PSIL scans? Coincidentally,

these are the two subjects that did not show the effect at T1 but did at T2. I am trying to gauge the likelihood that regression physiological recordings (which are likely more “erratic” when subjects are on PSIL) could introduce the kinds of effects reported in this paper.

In the primary analyses and main figures, we did not use physiological recording-based regressors. This methodological point has now been stated more clearly in the manuscript. Part of the reason for this was because HR and RR data were not collected for two participants (P1, P3).

To assess if physiological regressors generated by the PhysIO Toolbox (Kasper et al., 2017) would alter the main findings presented in the manuscript, we selected the two participants with the highest quality pulse and respiratory data (P4 and P5) and added PhysIO-generate regressors (19 regressors generated from pulse-ox, respiratory belt, and their combination) during the nuisance regression step. As demonstrated below, results were not altered by the inclusions of PhysIO-generate regressors (R2R Fig. 6 and new Supplemental Figure 5).

R2R Figure 6. Primary results remain unchanged by the addition of PhysIO-based nuisance regression (in a subset of participants with the highest quality pulse and respiratory traces).

8) Subject IDs. The subject identifiers in Supplemental Figure 3 do not match those in the main text. This information is needed to help understand if the subjects showing the strongest effects in Extended Data Figure 2 are also individuals with largest changes in breathing or heart rate. Also, there are 8 total subjects in Supplemental Figure 3, but only 7 reported elsewhere. Please clarify this discrepancy.

Thank you for highlighting this. P1 and P3 had pulse and respiratory data collected on replication scans only. Thus, the 8 data points in Supplemental Figure 3 correspond to P4-7 and the 4 replication protocols (P1R, P3R, P4R, P5R). These are now correctly labeled in the revised Supplemental Figure 3.

9) Multi-echo fMRI acquisition and processing. The fMRI processing description in supplement do not adequately describe how the multi-echo fMRI data were handled.

For example, were the preprocessing steps currently listed (correcting for odd/even slice intensity differences, whole brain intensity normalization, removal of thermal noise via NORDIC) are applied before or after the optimal combination of echoes. This is important information because for the multi-echo modeling (fitting $T2^*$ decay at each voxel to obtain weights for optimal combination) to be valid, one must avoid rescaling values within the different echoes in a manner that does not maintain the relative differences in signal intensity.

We thank the Reviewer for asking for this important point of clarification in the multi-echo processing stream. The optimal combination procedure was performed after slice-time interpolation and after NORDIC thermal noise suppression, but before intensity normalization (see revised “Resting-state functional MRI processing, and surface projection” in the supplemental materials). Thus, each echo individually underwent slice-time interpolation and unstructured noise suppression by NORDIC to improve signal-to-noise. Intensity normalization to mode 1000 was computed after the 5 echoes had been optimally-combined (Posse et al., 1999) into a single BOLD timeseries.

The Reviewer is understandably likely most concerned about the possibility that the novel NORDIC thermal denoising strategy may be rescaling the echoes with respect to each other. Theoretically, the maneuver is meant to suppress unstructured noise and, thus, should be expected not to bias the measured signal at each voxel in a systematic way. However, to address the Reviewer’s concern empirically, we have processed multi-echo data both with and without the NORDIC step. As the outcome measure, we report the mean $T2^*$ at each voxel estimated from the 5 echo times across 10x15 minute resting state runs. The results are displayed in R2R Figure 7. The estimated $T2^*$ values within a whole brain with and without NORDIC processing are nearly identical (see distributions in R2R Figure 7). CSF voxels, which are expected to have high $T2^*$ values, exhibited modest increases in $T2^*$ with NORDIC; high susceptibility regions tended to have very modest decreases in $T2^*$ estimates with NORDIC. Otherwise, differences between the $T2^*$ estimates with and without NORDIC processing within gray matter were near zero, suggesting no meaningful systematic bias in the $T2^*$ value estimate in these parts of the brain. These results demonstrate that the NORDIC processing should not systematically bias relevant multi-echo results and is thus unlikely to account for the observations reported in this manuscript.

-

[REDACTED]

R2R Figure 7. A) Distribution of mean $T2^*$ values computed within a brain mask for a single subject averaged across 10 runs, with and without NORDIC noise suppression processing step. B) Difference in $T2^*$ values within a brain mask, non-NORDIC data subtracted from NORDIC data. The average difference in $T2^*$ in gray matter was extremely small (-0.4^* ms), suggesting no meaningful systematic bias in $T2^*$ values with and without NORDIC.

10) Treatment guess. In the supplemental methods and results section, it is reported that subjects were asked to guess if they had received PSIL or MTP. I do not see any data in the main text or supplement showing what percentage of subjects correctly guessed PSIL vs MTP.

The supplement now reads: "After each drug session (in the initial blinded cross-over portion of the study), participants were asked to guess if they had received psilocybin or MTP. 6/7 participants correctly guessed which dose was psilocybin. Curiously, P3, the only participant who guessed incorrectly, showed a smaller FC change during psilocybin than any other subject (with the exception of P5 replication dose, in which the participant vomited 30 minutes after swallowing the capsule)."

Referee #2 (Remarks to the Author):

In this study, Siegel and colleagues studied acute and persistent changes in brain network organization after a single dose of psilocybin. They found that psilocybin changes functional connectivity 3 times more than methylphenidate. In particular, psilocybin led to desynchronization of brain networks throughout association cortex, with effects strongest in the default network. Performing a perceptual task reduced psilocybin-driven FC changes and desynchronization. Psilocybin induced persistent decreases in functional connectivity between the anterior hippocampus and cortex. Overall I think this is a very strong study.

We thank the reviewer for the positive feedback.

Major comments

1) Maybe I am misreading the supplement, but the main text states that “Psilocybin-associated FC change was greatest in the default mode network ($P_{spin} < 0.001$; $P_{spin} > 0.05$ for all other networks; Fig. 1e)”. I assume this is only referring to the cortex since the spin test was used? ‘

However, according to Supplementary Figure 1b, it seems that during PSIL1 (which is presumably first psilocybin session), the largest cortical FC change was not DMN, although largest subcortical FC changes do correspond to DMN (Supplementary Figure 1c)? On the other hand, the largest cortical FC change during PSIL2 is DMN (Supplementary Figure 1b). I am confused by this inconsistency.

Correct, the spin test presented in the main text (Fig. 1e) is based only on cortical FC Change and was conducted using all psilocybin sessions for all participants (both PSIL1 and PSIL2, for those who returned for the replication protocol). We have now more clearly indicated this in the Figure 1 caption and the following sentence in main text: “In the cortex, the largest FC change (average across all psilocybin sessions) was in the default mode network.”

2) “Participants were required to have had at least one previous lifetime psychedelic exposure (e.g., psilocybin, mescaline, ayahuasca, LSD)” - What is the rationale for this?

Our protocol required individuals to have had prior exposure to psychedelics and excluded individuals with serious adverse response (based on the Challenging Experience Questionnaire). This was done to increase the likelihood that participants would be able to tolerate the high dose of psilocybin safely. This has not be clarified in the supplemental methods.

3) I found the description of the LME model hard to follow. For example, what does “i-th group” refers to? Is it referring to the i-th subject? Another example is whether “j-observation” refers to the “j-th visit” or something else? Since there are 4 drug conditions (“pre-psilocybin, PSIL, MTP, post-psilocybin”), are these coded using 3 one-hot vectors or something else?

Thank you for pointing this out. We have clarified the description of the LME. The method description now more clearly reads :”Let y_{ij} be the rs-fMRI metric for the j -th observation (15-minute fMRI scan) within the i -th participant... u_{0j} represents the random intercept for the i -th participant, accounting for individual-specific variability. v_{0j} represents the random intercept for the j -th observation within the i -th participant, capturing scan-specific variability... In Matlab (Wilkinsonian notation), this model is expressed for every vertex $Y(\text{vertex}) = \text{fitlme}(\text{groupd}, \text{FC_Change}(\text{vertex}) \sim \text{drug} + \text{FD} + \text{task} + \text{task}*\text{drug} + (1 | \text{SubID}) + (1 | \text{session}))$.

4) Furthermore, I am also not sure about the asymmetry in which PSIL and MTP was treated. For example, drug condition were split into “pre-psilocybin, PSIL, MTP, post-psilocybin”, but there was no pre-MTP and post-MTP. I am not sure how the asymmetry influences the result. For example, if drug conditions were coded as “pre-MTP, PSIL, MTP, post-MTP” instead, how will the results (Figure 1) change?

We test pre-MTP and post-MTP differences as a control for persistent effects of psilocybin and no change was observed. In the methods (line 569 of initial submission) we state “As a control, we tested anterior hippocampus ‘FC change’ pre- versus post-methylphenidate using both the LME model, and an equivalence test ...” In the presentation of persistent hippocampal-cortical FC change (line 284 of initial submission), we state “No persistent FC change was observed post-methylphenidate (see Methods – Persisting effect analysis; LME ‘FC change’ 90% CI = -0.056 – 0.080; Equivalence $\delta = +/- 0.086$, $P_{\text{pre-MTP:post-MTP}} = 0.77$).”

5) Page 4 (word document) - “At the group level, psilocybin caused large acute FC change across most of the cerebral cortex ($P < 0.05$ based on linear mixed effects model and permutation testing)” - Unclear how and what permutations were performed for the LME modeling. There was a lot of focus on the wild bootstrap in the methods section, but I don’t see an explanation of the permutation test. What was permuted?

Wild bootstrapping was used for permutation testing. Wild bootstrapping generates permutations (in our case 1000) by taking the residual error terms that result from the model, and randomly

inverting them, and recomputing the LME model. As described by Mammen et al., 1993 (ref 112), this approach was devised to assess likelihood of a result occurring by change in high dimensional linear models. Particularly for models that are not independent and identically distributed (non-i.i.d.) and heteroscedastic. It was optimal for our analysis because partially nested structure (with dimensions of task/participant/drug/session) and imperfectly balance (different numbers of sessions, usable scans per session for each participant) of our dataset makes more straightforward permutation of labels impossible (or at best, extremely complex).

6) “distribution of these effects closely matched PET-based maps of 5HT-2A receptor density (serotonin 2A) (Fig. 3e)” -> Is there any quantification of the similarity?

We did not have access to PET result registered to the same surface atlas, nor did we feel that the effort to take those steps was necessary. We now say “the distribution of these effects appear to match PET-based maps of 5HT-2A receptor density (serotonin 2A) (Fig. 3f).” We cite Preller et al., 2020, where quantitative comparison was made to mRNA expression maps. However, we also cite and show the PET-based map as this is likely a more biologically valid means of measuring 5HT2A signaling.

7) In Extended Data Figure 5, we actually see increased FC between many pairs of brain regions (many more red color than blue color), especially involving association networks. Given the increased FC, how do we reconcile this with the “desynchronization” narrative, which suggests a decrease in FC?

This is a great question that raises some important nuances. The manuscript could alternatively be titled “psilocybin desynchronizes brain activity and disorganizes brain networks” – although that is a bit more wordy. The term ‘desynchronization’ was borrowed from the observation of decreased local and global NGSC (and the fact that this is consistent with desynchronization observed at the micro- and meso-scale in animal research). The consequence of psilocybin induced desynchronization is a loss of normal segregated network structure. Thus, within network FC (boxes along the diagonal in ED Fig. 5) is decreased, while between network FC ((boxes off the diagonal in ED Fig. 5, which is negative or 0 at baseline) is increased. To make this point more clear, have added visualizations of average changes in within network and between network FC in Fig. 2d.

8) I also think that portion of Extended Data Figure 5 should be brought into the main text. Right now, the main figures show some rather derived (downstream) measures like multidimensional distance (Figure 2), overall FC changes at each location (Figure 1) and normalized global spatial complexity (Figure 3) that do not provide any insights into intra and inter-network FC changes.

Extended Data Figure 5 is important because it shows the underlying change in brain FC that give rise to the “derived (downstream)” measures.

We agree that more clearly highlighting the pattern of intra and inter-network FC provides a useful replication of past studies reporting that psychedelics increase inter-network FC, and also more clearly connect our result, derived from multi-dimensional scaling, to this phenomenon. We have included intra and inter-network FC in Figure 2d. The corresponding Correlation matrices, which are less interpretable to a non-neuroimager audience, are provided in Extended Data, which is published as a part of the main manuscript.

9) If I understand correct, whole brain FC was computed using group-level ROIs instead of individualized ROIs? This is probably ok, but seems a bit of a waste given the amount of data collected per participant.

You are correct. The full FC distance matrix (shown in Extended Data Fig 3) was computed on FC matrices generated from the Gordon-Laumann parcellation. The FC distance matrix was used for Fig. 1d (whole brain FC change) and all of the multi-dimensional scaling analysis shown in Fig. 2.

We decided that this was appropriate because whole brain FC change (i.e. comparisons to baseline from this FC distance matrix) produced very similar results when computed at the vertex level versus using group ROIs. Note the similarity between overall FC change in figure 1a/b/c and Fig. 1d (e.g. MTP produces slightly larger FC change scores than day-to-day PSIL produces dramatically larger. HOWEVER, computing the full similarity matrix between >100 scan using FC from >90,000 vertices (which was needed to determine FC change between person) was not computationally practical, thus group-level ROIs were used.

Connectivity was computed in multiple ways in this manuscript (vertex, group parcellation, individual parcellation) depending on what was most appropriate to the each analysis. Thus, the following approaches were taken:

- The primary FC change analysis was done at the vertex level.
- Group-level ROIs (the Gordon-Laumann parcellation) were used in analyses that required a comparison of brain wide connectivity between individuals (e.g. Fig. 1d, Fig 2, Extended Data Fig 4 and 5).
- Individualized infomap-based network detection (as shown in Extended Data Fig 2) was used in compared vertex-wise results across participants at the level of resting state networks (e.g. Fig 1e).
- In the case of ‘local NGSC’, we compute this measure using individual-specific regions on interest (discussed further immediately below).

10) “Normalized global spatial complexity” should yield one number per brain. I am not sure how the spatial maps (Figures 3d and Figure 3e) were obtained.

We computed both whole brain and ‘local’ (parcel) NGSC (as indicated in Fig. 3b/c). Local NGSC we computed using **individual-defined regions on interest** (using procedures as in Laumann et al., 2015 and Gordon et al., 2016) defined using all of a subjects scans. This was done because the parcellation approach is specifically designed to generate homogenous regions. In figure 3d, we simply mapped individual-defined parcel NGSC values (psilocybin vs baseline) for each participant and then averaged the vertex maps. This is why there is a slightly splotchy appearance in Fig. 3d. We now describe the procedures more accurately in the figure, caption, and methods.

11) Perhaps the most important question the study seeks to address is the persistent effects of psilocybin. Yet, I am worried about circularity/double dipping when looking at persistent changes due to psilocybin. More specifically, in the supplemental material, it states that “This analysis indicated that the limbic system was the only system to show FC change after psilocybin that exceeded chance ($P < 0.05$, uncorrected)... Following this observation, we probed FC change in the constituent parts of the limbic system. We separated the limbic system into five bilateral regions of interest - anterior hippocampus, posterior hippocampus, ventromedial thalamus, amygdala, and nucleus accumbens – and repeated the FC change permutation testing approach 500 times. Here, the anterior hippocampus was the only region for which pre- and post-drug FC change exceeded all 500 label permutations.” - Since was no correction for multiple comparisons in the first round of “system selection”, this should result in potential double dipping / circularity in the second round of analysis focusing on the regions within the limbic system.

We thank the reviewer for pointing out the potential issues with the reporting of statistics for this supplemental analysis. After discussion with our neuroimaging biostatistician collaborators Taki Shinohara and Yong Chang, we have decided to remove the supplemental RSN-wise analysis entirely. Here is our reasoning:

The primary analysis of persistent effects (Fig. 5) was based an *a priori* hypothesis. Our investigational new drug protocol submitted to the FDA in 2021 states “Aim 1: Examine persisting changes in blood flow, brain activity, and functional connectivity after psilocybin... H2: change in FC between hippocampus and cingulate cortex... will be tested on the entire cohort using a mixed-effects model”. As indicated in the text, we hypothesized persistent changes in hippocampus connectivity on the basis of multiple prior studies reporting persisting neurotrophic effects in the hippocampus in rodent and porcine models (Hesselgrave et al., 2021; Mason et al., 2020; Raval et al., 2021).

The supplemental RSN-wise analysis was conducted later to explore if specific networks showed persistent effects. The RSN-wise analysis of persistent effects did not use a linear mixed effects model (as was used throughout the rest of the paper), and thus did not optimally control for fixed and random effects (such as subject, head motion, session). Moreover, the testing of limbic regions (after observing $P < 0.05$ for the whole limbic network) was exploratory and indeed violates the assumptions of the permutation test.

The analyses were listed in the opposite way (from how they were conducted) in the main text for the purpose of the narrative. But, in order to remain truer to our pre-specified hypothesis and avoid the issue raised by the reviewer, we feel the best option is to remove the supplemental RSN-wise analysis.

Minor comments:

1) In Supplementary Figure 1b (second column), are you actually showing dissimilarity rather than “similarity: $z(r)$ ”? Because if it’s similarity, then shouldn’t larger values indicate less change, so I would expect a rough reverse ordering between first column (Euclidean distance) and second column (Similarity: $z(r)$)? But ordering in the two columns seem somewhat consistent.

You are absolutely right. Similarity changes (decreases) were inverted. The caption stated “[decrease in] bivariate correlation” but this wording was not specific enough. The text now more clearly states that decreases in similarity were inverted to facilitate comparison with distance values on the left (in the caption for Supp. Fig 1b and d).

2) Figure 1 captions: “Euclidian “ -> “Euclidean”

Corrected. Thanks.

3) Page 16 (word document) - “To compensate for the implementations of this LME model on multiple rs-fMRI-related dependent variables. differences were highlighted when $P < 0.005$. “ -> “To compensate for the implementations of this LME model on multiple rs-fMRI-related dependent variables, differences were highlighted when $P < 0.005$. “

Corrected. Thanks.

3) "Extended Data Figure 5" - "Right shows the effect of psilocybin" - should be bottom left. "Bottom left shows effect of methylphenidate" - should be right.

Corrected. Thanks.

4) Page 19 (word document) - Under "Normalized global spatial complexity" in the methods section, "We used a n approach previously" should be "We used an approach previously"

Corrected. Thanks.

5) Page 20 (word document) - "We assessed change in anterior hippocampus 'FC change' per- versus post-psilocybin" -> Should be "pre-" not "per-"

Corrected. Thanks.

6) The extended data figures do not match the ordering of the main text.

Corrected. Thanks.

Referee #3 (Remarks to the Author):

In this manuscript by Siegel, et al the authors expand on previous work that has examined acute changes in signal power and functional connectivity (FC) during acute psychedelic administration. They do this by longitudinally imaging subjects dosed with psilocybin using a method they previously developed, precision functional mapping (Gordon, et al 2017), that utilizes extensive within-individual data collection to characterize brain function in specific individuals. Six participants were dosed psilocybin and methylphenidate 1-2 weeks apart and underwent several MRI sessions (~18/participant). The main findings of the paper are that: 1) psilocybin causes profound changes in FC in cortex and in subcortical areas, and mostly in the default mode network (DMN) 2) reduced segregation between the DMN and other networks explained the variance between psilocybin and other groups, and that this held consistent for other data analyzed with in humans dosed with other psychedelics 3) that brain activity across regions is overall less synchronous with psilocybin 4) participation in a perceptual task weakens these effects and 5) some effects are persistent over time.

We thank the reviewer for taking the time to understand and summarize our findings.

The longitudinal precision functional mapping approach is potentially powerful here, as it should allow for a detailed understanding of individual differences with psilocybin use. In addition, longitudinal imaging with psychedelics seems important given long lasting effects on humans. With this approach, however, most of the analyses presented in the paper focused on group differences, rather than individual ones.

We thank the reviewer for the positive feedback and thoughtful review. Further exploration of individual differences (as prompted by your review) revealed that, differences in the magnitude of psilocybin's effects on brain connectivity were closely related to differences in ratings of psilocybin's subjective effects (new manuscript Fig. 1f-h reproduced below for convenience). As mentioned above, we added a section entitled **individual differences** in the revised manuscript. Specifically, the following paragraph has been added "Acquiring large amounts of data with multiple doses of psilocybin in each subject allowed us to assess not only common effects of psilocybin but also individual differences. The magnitude of psilocybin-associated FC changes showed marked variability (Fig. 1d-g, Supplemental Fig. 1b), larger than would be expected based on other sources of variability alone (likelihood ratio test for variance of random slopes for a participant-specific response to psilocybin, $P = 0.00245$)^{55,56}. When four participants returned 6 months later for a second psilocybin dose, the focality of psilocybin's effects were moderately similar (Pearson correlation between change maps; $r = 0.67$, two-sided $P < 0.001$) (Extended Data Fig. 2 & 3), but the magnitude of FC change varied widely in (Fig. 1f). Variability in FC change, but not head movement, was strongly related to subjective rating of drug effects across sessions (Fig. 1g; LME model predicting MEQ score: $T\text{-stat}_{FC\text{Change}} = 7.68$, $P = 3.5 \times 10^{-6}$, $T\text{-stat}_{FD} = -1.26$, $P = 0.23$). Many parts of association cortex

demonstrated a relationship between brain change and subjective experience (Fig. 1h). Yet, comparing psilocybin's effect across individuals also revealed commonalities, described below.”

New manuscript Fig. 1f-h

The main finding in both figures 1 and 2, corroborates previous results, that psilocybin desynchronizes cortical and subcortical activity and is therefore not entirely novel.

We have made a number of changes to the manuscript to acknowledge prior fMRI studies attempting to measure effects of psychedelics but also more clearly point out the conceptual advances of our own work (as well as replication of prior findings). This includes additional description of prior fMRI studies in the introduction, better linking our results to prior fMRI studies in the discussion, and a lengthy review in the supplement of prior efforts to measure persisting effects.

In figure 3, I am not sure the authors' conclusion that they have presented evidence for 'grounding.' What they have shown is that at this dose of psilocybin, participants can still engage task associated networks, when performance is maintained at a high rate but this does not seem entirely surprising – higher doses of the drug might be helpful here to prove this point.

The dose used in our study (25mg) is considered a high dose. It is the identical drug and dose used in the recent Raison et al., 2023 clinical trial and equivalent to the highest dose tested in the recent Compass clinical trial (Goodwin et al., 2022). Regarding grounding, see a longer response to Reviewer #4s comment about grounding.

Finally, and what is perhaps the most novel finding of the paper is the persistent decrease of FC of the anterior hippocampus and DMN. The authors assert that this their study is the first to study persistent effects of psilocybin however I can think of at least 1 other (Barrett, et al 2020).

It was not our intention to assert that we are the first to study persistent effects of psilocybin. The introduction now states “Preliminary efforts to identify network changes in the weeks after psilocybin have yielded mixed results (see supplement for a review)^{32–34}. Moreover, persistent effects of psilocybin on hippocampal-cortical circuits have yet to be characterized in humans.” Supplemental section “Further discussion of persistent effects” now provides a full review of Barrett and other studies attempting to characterize persistent effects.

We do assert that our study design offers a substantial advance over prior efforts and that the persistent decrease of FC that we report is novel and compelling.

Minor points:

Discussion is repetitive and includes a lot of points that were also included in the introduction. It might be good to have more thorough citations and review of literature in the introduction and use the discussion for interpretation of their results and next steps/outlook.

We have substantially altered both the introduction and discussion. The revised version includes more thorough citations and review of literature and removes repetition.

The layout of the figures and the reference to them in the text is often confusing. This is especially the case for figure 5. It was unclear to me what was plotted in panel 5b – and I could not find reference to this in the main text. It seems the authors are plotting numerous scans of each patient of the FC connectivity of the area highlighted in figure 5a? It was also unclear why this is going up in 5b but is a negative value in 5c. A clearer description of the methods used to quantify FC change in these panels as well as an explicit description of what is plotted in each subpanel.

We thank the reviewer for pointing out the need for better explanation of figure 5. We have modified the caption and text in hopes of providing greater clarity to readers.

Referee #4 (Remarks to the Author):

In this study, Siegel et al. use precision functional mapping to study within-subject acute and persistent changes in functional connectivity (FC) induced by 25 mg of oral psilocybin versus a psychoactive placebo (40 mg of oral methylphenidate) in 7 healthy adults (6 with useable fMRI data with psilocybin) administered in a blinded cross-over fashion. Their resting-state fMRI data collection protocol is rigorous and includes: (1) repeated scanning over multiple days to collect hours of data for establishing stable FC baselines for each participant, (2) scanning at approximately the same time each day to reduce diurnal variation in FC, (3) multi-echo data collection to reduce head motion, (4) real-time monitoring software, (5) field map unwarping, and (6) cardiovascular as well as respiratory monitoring to remove physiological noise from their fMRI data.

To characterize the acute effects of psilocybin, the team scanned participants during peak drug intoxication. During the scan, subjects passively stared at a fixation cross for 30 to 120 minutes and also completed an auditory-visual matching task. This was followed by hours of resting-state scanning collected in the days following psilocybin administration to evaluate for any persistent effects on FC. 4 of the participants also completed an open-label replication protocol with psilocybin 6-12 months after the blinded portion of the study.

Using this approach, Siegel et al. made a number of findings: (F-1) psilocybin acutely disrupts normal FC patterns globally, but most pronounced, in the default mode network (DMN), (F-2) psilocybin acutely reduces segregation between the DMN and other networks (fronto-parietal, dorsal attention, salience, and cingulo-opercular) that are normally anti-correlated with it, (F-3) psilocybin-associated disruptions in FC are spatially associated with 5HT_{2A} receptor density in humans, (F-4) individuals are still able to effectively engage task-positive networks on 25 mg of oral psilocybin and their FC patterns appear less de-synchronized while engaged in a task compared to rest, and (F-5) they observed persistent decreases in functional connectivity between the anterior hippocampus and DMN for up to 3 weeks following psilocybin.

This was a well-conceived resting-state fMRI study with oral psilocybin in healthy volunteers and serves to corroborate a number of important findings that have already been made and reported in the literature. For example, Preller et al. (*Biological Psychiatry* 2020;88:197-207)¹, Mason et al. (*Neuropsychopharmacology* 2020;45:2003-11)², and Madsen et al. (*European Neuropsychopharmacology* 2021;50:121-32)³ have previously demonstrated acute-phase findings (F-1), (F-2), and (F-3) with oral psilocybin in healthy volunteers. Persistent effects on FC in the weeks and months following oral psilocybin administration in healthy volunteers has also been examined and reported on by Barrett et al. (*Scientific Reports* 2020;10:2214)⁴ and McCulloch et al. (*Journal of Psychopharmacology* 2022;36:74-84)⁵. These papers should be discussed in the manuscript and serve as points of comparison to the findings reported here.

While this work is an important replication, it is not adequately novel to justify publication in Nature.

Moreover, I would like to provide some comments that I hope will clarify and strengthen the manuscript.

We are extremely grateful for the reviewers careful scrutiny of the manuscript and many helpful comments below. This was truly above and beyond the expectations of peer review and has helped to strengthen our manuscript considerably.

As mentioned above, the revised manuscript now provides a more extensive discussion of previous efforts to characterize acute and persistent effects of psilocybin using fMRI. We contend that, while some fMRI studies have attempted to measure persistent effects, most of these studies were limited in power and scope and that our approach and results offers a substantial advance. We have taken effort to review and justify this perspective in the revised manuscript. See specifics below.

Major Comments:

Supplementary Methods (lines 112-114): Low-pass filtering the motion time courses at 0.1 Hz prior to computing framewise displacement allows data collected during excessive head motion to enter your analyses and can lead to spurious findings. It would be prudent to repeat the main analyses of the manuscript without the low-pass filter (i.e. after removing all data containing excessive head motion) and ensure that similar results are still obtained. The results without the low pass filter should be reported.

To clarify, the actual rsfMRI timecourses themselves were not filtered at this point in preprocessing, just motion parameters. And more specifically, only the Y-translational measure was filtered, for the purposes of removing oscillations in the phase encoding direction that are believed to not actually represent true head motion (but rather small magnetic field shifts). The phenomenon of high-frequency oscillations contaminating y-translation motion estimates is described in Fair et al., 2020 and Gratton et al., 2020 "Removal of high frequency contamination from motion estimates in single-band fMRI saves data without biasing functional connectivity" - this has now been cited in the manuscript.

Empirically, this maneuver had a very small effect of FD measures and censoring (and no effect on subsequent analyses of rsfMRI data). To demonstrate this, we have included a supplemental video showing motion traces for every scan both with and without the Y-translation low-pass filter (Supplemental Video).

The supplemental methods text has been modified to read: “Thus, we low-pass filtered the y-translation estimate time courses at 0.1Hz in all participants prior to computing FD to prevent superfluous data loss^{126–128}. We observed that this had minimal effect on the computation of FD (see FD computation with and without y-translation filtering in Supplemental video 2).

Supplementary Methods (lines 169-170): It is not clear why the impulse response was modeled 19.37 seconds after each trial.

The fMRI BOLD sequence had a TR of 1.761s, therefore we modeled the finite impulse response over $11TR \times 1.761s/TR = 19.37$ seconds after each trial. As can be seen in the figure below, this is how long the hemodynamic response takes to return to baseline in the cortex. This is now more clearly stated in Extended Data Figure 4 caption and in the supplemental methods.

Moreover, it appears that all other task-related variance was unmodeled – specifically stimulus duration, response, conflict, error, etc.

We note, for the purposes of this response, that when we recompute the timecourses using the SPM 3-parameter hemodynamic response function (with canonical, dispersion, and time parameters - which instead last 22.89s) plus a parameter for button press, the results (with respect to timecourses and drug differences) are unchanged.

R2R Figure 8. Timecourses from the perceptual (auditory-visual matching) task, computed using SPM's 3-parameter hemodynamic response function.

The goal of our analysis of evoked responses (Extended Data Fig. 4) was to assess if psilocybin affects the hemodynamic response to neural activity. To maximize power to answer this question, it was most reasonable to compare the 'main effect' across all trials. We did experiment with modeling button press. Modeling other parameters (stimulus duration, conflict, error) was not appropriate for the following reasons: the visual stimulus duration was identical for every trial (500ms), and all participants had greater than 94% accuracy (Extended Data Fig. 4b) so there were not a sufficient number of error trials (or sufficient differentiation between conflict and button press conditions) to be able to separately model those factors.

Without regressing out the full effect of the task, it is not possible to appreciate the underlying FC.

In the primary FC analysis presented in Figure 4, evoked response was not removed from timecourses prior to computing FC. To address the reviewer's concern, we have added to the supplement analyses of FC change and NGSC conducted on data for which the same 3-parameter hemodynamic response function was used as a regressors (along with other tissue- and movement-based nuisance regressors) (see "**Regression of evoked response in preprocessing**" in supplement). Regressing out the effect of the task had negligible impact on the primary observations (e.g. the interaction of task with psilocybin on NGSC and FC change). This has been added to the manuscript in Supplemental Fig. 3.

Main Text (lines 222-228): I disagree with the conclusion being drawn here. At 25 mg of oral psilocybin most individuals maintain a connection to reality as is evidenced by the 100% performance on the auditory-visual matching task. What I believe the results are showing is that individuals are still able to effectively engage task-positive networks following ingestion of 25 mg of oral psilocybin, which appears as less de-synchronized brain activity compared to the brain at rest. I would not equate this to discovering a neurobiological basis for 'grounding'.

We agree with your interpretation of the results that "individuals are still able to effectively engage task-positive networks following ingestion of 25 mg of oral psilocybin, which appears as less de-synchronized brain activity compared to the brain at rest." This seems to be in line with the interpretation put forth in the manuscript. We do not assert that the evidence presented is sufficient to prove a causal relationship between decreased desynchronization and the psychological experience of grounding. Indeed, we did not ask participants to report intensity of drug effects during the perceptual task, nor did we mechanistically alter synchronization and assess effects. The manuscript only discusses that engaging in a task appears as less de-synchronized brain activity and this may provide an explanation for why focusing attention externally can decrease the perceived intensity of psychedelic effects. Since the inference to grounding is indirect, the wording of the section has been changed to read "*may parallel the psychological principle ...*"

Expansion of Discussion: The manuscript would benefit significantly from an expanded discussion section that compares the results of this study to the work done by others who have studied the acute (Preller et al.1, Mason et al.2, Madsen et al.3) and persistent (Barrett et al.4 and McCulloch et al.5) effects of oral psilocybin in healthy volunteers.

As mentioned above, the revised manuscript now provides a more extensive discussion of previous efforts to characterize acute and persistent effects of psilocybin using fMRI – including reference to all of the articles listed, as well as some others.

Minor Comments:

Summary (lines 35-36): The way this sentence currently reads implies that everyone experiences distortions in space-time perception and ego-dissolution on psilocybin. These subjective effects are dose dependent, and even at the highest doses, do not unanimously occur in all individuals who consume psilocybin.

Updated (Added 'can')

Summary (line 36): In addition to having persistent anti-depressant effects, psilocybin has been shown to have anti-addictive properties⁶ and more. Perhaps the term “transdiagnostic” better captures the effects of psilocybin?

We agree. We reference results in substance use clinical trials in the first paragraph of the introduction. We have added the Bogenschutz reference there.

Summary (lines 37-39): Preller et al.¹ demonstrated reduced global connectivity in associative brain regions in their resting-state fMRI study on the acute effects of oral psilocybin in healthy volunteers. Mason et al.² and Madsen et al.³ also demonstrated reduced DMN integrity in their resting-state fMRI studies on the acute effects of oral psilocybin in healthy volunteers. These studies should be cited given their similarities to the present work.

These studies have now been cited in the summary. In addition, as mentioned above, prose connecting our results to these important studies has been added to the discussion.

Summary (lines 39-40): It is still unclear how brain network changes underlie the variety(ies) of subjective psychedelic experience(s) (ego dissolution, sense of unity, feelings of love, synesthesia, oceanic boundlessness, etc.)

We have taken this suggestion. Thanks.

Summary (lines 40): There have been some studies evaluating enduring effects on functional connectivity in healthy volunteers following oral psilocybin administration. For example, Barrett et al.⁴ scanned 12 healthy volunteers 1 week and 1 month after oral psilocybin ingestion and McCulloch et al.⁵ scanned 10 healthy volunteers 1 week and 3 months after oral psilocybin ingestion.

To give richer context to the aims and the findings of our work, we have added a review of studies reporting persistent effects of psychedelic in the supplement. While we are familiar with, and appreciate, these studies. A part of the impetus for this study, was literature reviews indicating that a consensus had not yet been reached on persistent effects and that many of the early attempts have been underpowered. Thus, we stated “it remains unclear... whether connectivity is altered long term.”

Summary (line 47): The anterior hippocampus is not considered part of the default mode network.

We point the reviewer to Zheng et al., 2021 - “we found the anterior hippocampus (head and body) to be preferentially functionally connected to the default mode network (DMN), as expected.” This has been replicated across multiple FC studies, and is also consistent with anatomical connectivity in the macaque (for review, see Kahn et al., 2008).

As Zheng and colleagues further state “Group-averaged RSFC studies have found the hippocampus to be functionally connected to the default mode network (DMN) (25–28). The DMN is deactivated by attention-demanding tasks and thought to be important for self-referential processes, such as autobiographical memory, introspection, emotional processing, and motivation (26). Other group-averaged RSFC studies have reported the anterior hippocampus to be preferentially functionally connected to anterior parts of the DMN, while the posterior hippocampus was more strongly connected to the posterior DMN via the perirhinal and parahippocampal gyri (29–32).”

The theory and findings of that study provided much of the basis for our hypothesis that psilocybin would reduce connectivity between the anterior hippocampus and the DMN. Figure 5 in our manuscript, in which we show parallel hippocampal cortical circuits for self- and goal-oriented processing, was taken from Zheng et al., 2021.

Summary (48-50): You have demonstrated that individuals are still able to activate task-positive networks on 25 mg of oral psilocybin. I would not equate this to discovering a neurobiological basis for grounding.

- See answer to this concern above. As with the main text, we have softened the wording in the Summary.

Summary (52-53): It is pure speculation that persistent reduction in hippocampal-DMN connectivity represents a mechanistic correlate of psilocybin’s anti-depressant effect. I think this should be removed from the summary.

- The language in the summary has been updated

Main Text (line 59): Consider citing Bogenschutz et al.6 as an example of psilocybin’s anti-addictive potential.

- Done. Thanks

Main Text (line 69): Change network to networks

- Done. Thanks

Main Text (line 72): Remove citation 28 – Timmerman et al. studied DMT, not psilocybin.

- Done. Thanks

Main Text (line 73-74): This is not true – there have been at least two studies evaluating persisting effects of oral psilocybin on FC in healthy volunteers, namely Barret et al.⁴ and McCulloch et al.⁵

- This has been changed.

Main Text (line 75-76): The anterior/middle hippocampus are not considered core parts of the default mode network. However, there is some evidence suggesting they are functionally connected.

- Our wording is based on the empirical result of Infomap that associate cortical DMN and regions of a number of subcortical structures into a single system. However, we understand that this can cause confusion, thus the wording has been changed (to say ‘functionally connected to the default mode network’)

Main Text (lines 109-111): “default mode network parts of the thalamus, basal ganglia, cerebellum, and hippocampus” is very confusing. Most imagers and neuroscientists consider the DMN to consist of 3 parts: 1. medial prefrontal cortex, 2. posterior cingulate cortex / precuneus, and 3. inferior parietal lobule. It might be more clarifying to say something like -- regions of subcortical structures that have been shown to be functionally connected with the DMN.

The wording has been changed.

Main Text (lines 114-115): As above, “In the cerebellum the largest FC changes were seen in the default mode network” is very confusing. Consider rephrasing.

The wording has been changed.

Main Text (line 119): Can you clarify high head motion = 1.5.

We have added the following sentence to the methods: In the 'high head motion' comparison ('hi:lo motion' in Supplemental Figure 1d), the two non-drug scans with the highest average FD were labeled and compared against all other baseline scans.

Main Text (line 123): It might be helpful to define the DMN since you've been referring to many structures in the paper that aren't traditionally considered part of this network.

Network definition based on Infomap is now explicitly stated.

Main Text (line 192): Preller et al.¹ have already shown that psilocybin induced disruptions in FC are spatially associated with 5HT2A receptor density in humans and this manuscript should be cited in the main text.

Done.

Main Text (line 202): I believe you mean to write Fig 3b-c not Fig 3b-d?

Corrected. Thanks.

Main Text (line 204): I believe you mean to write Fig 3d not Fig 3b-d?

Corrected. Thanks.

Main Text (line 205): I believe you mean to write Fig 3e not Fig 3d?

Corrected. Thanks.

Main Text (line 206): I believe you mean to write Fig 3f not Fig 3e?

Corrected. Thanks.

Main Text (line 274): Atypical cortico-hippocampal connectivity has been associated with affective symptoms in patients with multiple sclerosis. The relevant patient population should be mentioned.

Corrected. Thanks.

Main Text (line 294): Consider changing the header of this section – it is unclear to me how the 2 paragraphs (lines 295-309) relate to the linking of acute and persistent effects.

Thank you for catching this! The two subheadings were accidentally switched. This has been corrected.

Main Text (lines 321-322): Please add citations to corroborate these statements.

Done. Thanks.

Main Text (line 326): The DMN and hippocampus are also highly relevant to addiction and significant work is being done to study psilocybin for the treatment of alcohol abuse and other addictive disorders. It might be worthwhile to add a sentence or two expanding the discussion on the relevance of your finding to this line of work.

This is a good insight. But given tight space limitations of the format, we are not able to add this to the main text.

Methods (line 382): Please provide some additional information as to how the multi-echo data was analyzed.

See revised “Resting-state functional MRI processing, and surface projection” in the supplemental materials

Methods (line 393): Add a reference for the previously validated event-related fMRI task.

The fMRI task was collected and validated as a part of a large stroke study but, to our knowledge, has not been published in a peer reviewed manuscript.

Methods (line 400): Were stimuli presented in random or pseudorandom order?

Stimulus order in the two trials did not vary across sessions. So participants' familiarity with the stimulus order may have had some impact on evoked responses. An analysis to investigate this possibility (R2R Figure 9) suggested that practice effects were mild, potentially because of the simple perceptual nature of the task.

R2R Figure 9. Peak activation magnitude (% of signal) over subsequent sessions repeating the same task. MTP session is blue, PSIL session is red.

Methods (line 413): Change achieves to achieve

Corrected. Thanks.

Methods (line 416): Change hippocampu to hippocampus

Corrected. Thanks.

Methods (line 420): Did you mean that you calculated the correlation between time series and then transformed each correlation using fisher z for group comparisons?

Yes. Corrected. Thanks.

Methods (line 440): Change advance to advantage

Corrected. Thanks.

Methods (line 442): Was there a post-MTP label?

Yes. This is described in the “Persistent effects analysis” section of the Methods.

Methods (line 448): Please define y – presumably this is the session/subject specific connectome?

Done. Thanks.

Methods (line 459): Change ‘.’ to ‘,’

Done. Thanks.

Methods (line 560-561): This sentence should be in the results, not the methods.

It is in the results (line 276 in first submission) but repeated here for clarity.

References (line 631 and 651): Citation 3 and citation 14 are both Carhart-Harris et al. Psilocybin versus Escitalopram for Depression. Please consolidate them to a single citation.

Corrected. Thanks.

Extended Data Figure 1: Please ensure that the range of MRIs depicted in panel (a) matches the imaging visits in panel (b). For example, I don't see any participant who had 9 'after' MRI visits in panel (b) – P1 comes close at 8 'after' visits. I only see a max of 4 'between' MRI visits in panel (b) but the range in (a) says up to 5.

Panel a depicts the study protocol. This has been clarified in the caption.

Extended Data Figure 1: It would be helpful to be consistent with your nomenclature for days. For example, in panel (b) psilo = day 0, however, in panel (d) baseline 1 = day 0. Furthermore, based on panel (b) it appears that P1 has 8 'after' visits, but in panel (d) there are only 7 listed. Please clarify.

Corrected. Thanks.

Extended Data Figure 1: The open-label replication protocol is described as "included one scan each of baseline, psilocybin, and after drug" in the figure legend but in panel (d) participant P1 appears to have 2 baseline and 2 after scans. Please clarify.

Corrected to say "The open label replication protocol 6-12 months later included one or two scans each of baseline, psilocybin, and after drug." Thanks.

Extended Data Figure 1: It might make sense to depict individual-specific MEQ scores in panel (e) since the rest of the paper reports within-subject effects.

Done. Thanks.

Extended Data Figure 2: Add Euclidian distance to the color spectrum in panel (a) to clarify units.

Done. Thanks.

Extended Data Figure 2: Consider adding an asterisk to the FC change brain for P3 on MTP to explain why their change map looks more pronounced on MTP than PSIL1.

We do not have a clear answer for why P3 had a larger FC change during MTP. However it is interesting to note that the largest changes were within the sensory-motor cortex, consistent with stimulant-induced FC changes in other participants and in ABCD data (Extended Data Fig. 5). We now report MEQ scores in Extended Data Figure 2.

Extended Data Figure 6: Consider re-labeling extended figure 6 to 4 to match the ordering presented in the manuscript. And visa-versa.

Supplemental Table 1: Why is the age of each participant depicted as a 2- or 4-year range?

Nature formatting guidelines specifically ask for age to be given by 5 year intervals to limit identifying information. P5 was listed as '18-20' because 18 was the minimum age allowed in our study.

Supplemental Table 1: Are you reporting baseline personality characteristics of each participant? Please clarify since Mini-IPIP was collected at 3 timepoints.

Done. Thanks.

Supplemental Table 1: I would clarify that '+' in MRI visits represents the additional scans done as part of the replication protocol.

Supplemental Table 1: Why is the rsfMRI data with methylphenidate not reported?

This information has been added to supplemental table 1. In so doing, we also noticed an error in the reported number of PSIL rsfMRI scans and corrected it. We thank the reviewer for bringing the omission and error to our attention.

Supplemental Table 1: Why are 2 participants missing respiratory and pulse data?

In the section titled *Physiological Monitoring during MRI*, we state “Recording of pulse and respiration recordings were using Siemens Trio was amended to the protocol prior to enrolling P4.” As noted above (and illustrated in supplemental Fig. 4), every participant had physiological recording done both on and off of psilocybin (because P1 and P3 had physiological recording during their replication protocol).

Reviewer Reports on the First Revision:

Referee #1 (Remarks to the Author):

The authors have been very responsive to my concerns. The new analyses and various clarifications have improved the manuscript and bolstered confidence in the results considerably. In my view, the revised manuscript is suitable for publication.

Minor point, but in Extended Data Figure 7, the authors have removed an outlier to emphasize the correlation between delta NGSC and MEQ scores (top left panel). I would like to see them treat the outlier in the lower right panel (delta respiratory rate) the same way (it appears to be close to ~2-3 SD from mean).

Also, I am unable to access / review the code, I get a 404 page not found error when using the link provided by the authors.

Referee #1 (Remarks on code availability):

I am unable to access / review the code, I get a 404 page not found error when clicking the link provided by the authors.

Referee #2 (Remarks to the Author):

The authors have addressed all my technical concerns.

Referee #2 (Remarks on code availability):

It says page not found when I try to go to the page.

Referee #3 (Remarks to the Author):

In this revised manuscript, the authors perform additional analyses and address many of the critiques brought up by my fellow reviewers. It is clear they spent time thinking about the critiques and made a good faith effort to address the majority of them. I particularly appreciate that in Figure 1, authors added some analyses to relate individual differences in FC change to mystical experience ratings. However, I still feel the authors have not fully leveraged their methods to understand time course of long-term changes across individuals and how this relates to the acute effects.

Also, the discussion is still quite repetitive. See examples below for illustration of this.

From intro:

In rodent models, transient activation of the 5HT_{2A} receptors by a psychedelic can alter neuronal communication in 5HT_{2A}-rich regions (e.g. the medial frontal lobe) and induce persistent plasticity-related phenomena^{5,8,9}...

Yet, inherent limitations of rodent models, and imperfect homology to the human 5HT_{2A} receptor²⁵, limit the strength of these assertions. Understanding the effects of psychedelics on human brain networks is critical to unlocking their therapeutic mechanisms.

Discussion:

Psychedelics rapidly induce synaptogenesis in the hippocampus and cortex, effects that appear to be necessary for rapid antidepressant-like effects in animal models^{9,24}. However, to understand the underpinnings of psychedelics' unique effects, human studies are needed.

Why restate the same thing twice esp. when it is review of what is known? Also, I hardly think the authors needs to state that human work is needed in this field at all, let alone two times in one manuscript. Maybe instead authors could expand on what they mean in the last sentence: "Novel methods to measure neurotrophic markers in the human brain¹⁰² will provide a critical link between mechanistic observations at the cellular, brain-networks and psychological levels." Or perhaps they can talk more about how their results may be relevant to psychiatric populations.

Finally with regard to "grounding" I think this sentence in the abstract is too much of an overstep: "Performing a perceptual task reduced psilocybin-driven FC changes and desynchronization, suggesting a neurobiological basis for grounding – connecting with physical reality during psychedelic therapy," given that the inference to grounding is indirect at best, in the actual experiment that they performed.

Overall, while the authors have addressed many of the comments and the manuscript is stronger overall, I remain ambivalent about its publication in this particular journal given conceptual advance of the study.

Referee #4 (Remarks to the Author):

Siegel et al. have done a great job responding to most of my comments. I have some additional items for them to consider, which I hope will further strengthen and clarify the manuscript prior to publication.

Comments:

- Main Text (line 79): Timmerman et al. studied DMT (citation 30). While DMT and psilocybin are chemically related, I would not cite this particular study as an example of psilocybin decreasing the power of electrophysiological signals.
- Figure 1 Panel D: There appear to be fewer open circles for the DMN column. I'm assuming that some circles are superimposed and therefore giving the impression of there being fewer?
- Main Text (line 133): Change average to averaged
- Main Text (line 147): I would specify that you're using MEQ-30 and not MEQ-43. You may also want to present MEQ-30 data as a percentage of maximum possible score instead of using 150.
- Figure 1 Panel F: I'm guessing 1st session and 2nd session refer to PSIL1 and PSIL2 (i.e. one average FC change brain is shown from the main experiment and one average FC change brain from the replication protocol)? It would be nice to know exactly when the resting state scans for each participant were acquired relative to when they consumed the oral psilocybin for both sessions. This might inform why these two participants are having such dramatic FC change differences from the same dosage of psilocybin on different sessions.
- Main Text (line 152) and Panel G: There are 16 data points here. I expected to see 10 data points: 6 participants who completed PSIL1 + 4 participants who completed PSIL2. Then a correlation between MEQ30 at the end of each session with whole brain FC change normed. How did you end up with 16 data points? Consider color coding each data point by participant contribution to the model. It would also be nice to know where in the time course following oral psilocybin consumption these points are coming from (possibly as a separate panel?).
- Figure 1 Panel G: Consider adding a line at 60% maximum possible score on the MEQ-30 (90 out of 150) to show threshold for "complete" mystical experience. Looks like 2.8 FC change is needed for achieving a "complete" mystical experience.
- Main Text (lines 157-159): "Acquiring multiple scans at multiple psilocybin sessions, enabled us to determine that the variability in psilocybin's brain effects was more likely due to differences in subjective experience than noise" – consider rephrasing to -- differences in subjective experience are more likely caused by variability in psilocybin's brain effects (probably as a function of time since ingestion, but also participant weight/drug metabolism/etc.).
- Main Text (Figure 1 F-H): The MEQ-30 has 4 dimensions: (1) mystical, (2) transcendence of time and space, (3) positive mood, and (4) ineffability. While you've presented the results for MEQ-30 total score, I think it would be informative to show the results of this analysis for the 4 dimensions of the MEQ-30. Are there certain FC changes that are more or less specific to these 4 dimensions? How do individual differences in FC change map to differences in the phenomenology of the psychedelic experience?
- Main Text (Figure 2, panel b): Why does one connection change appear to be occurring outside the brain?
- Main Text (lines 268-270): "and the distribution of these effects appear to match a map of 5HT-2A

receptor density” – this is not convincing with just a qualitative assessment of Figure 3 panels D-F. I think it would be helpful to quantitate this claim. LSD binds more tightly to the 5HT-2A receptor than psilocybin. Does LSD-induced desynchronization overly more closely with 5HT-2A expression than psilocybin?

- Main Text (line 410): Change prepare to prepared.
- Main Text (line 432): I would be more specific and state that you assessed subjective effects using the total score of the MEQ-30. You may also want to provide some context as to why MEQ-30 was used and not 5D-ASC or 11D-ASC for measuring acute subjective effects.
- Main Text (line 433): Where are the results for personality change? Was there any change pre vs post psilocybin in these subjects?
- Extended Data Figure 1 Panel D: Was the number of resting-state scans acquired for PSIL1 session for this participant 2 x 15 minutes (panel label) or 6 x 15 minutes (supplementary table 1)?
- Extended Data Figure 1 Panel E: Might be worthwhile to also show possible individual differences in psychedelic experience of the 7 participants instead of just group differences between psilocybin and methylphenidate.
- Extended Data Figure 3 – Participant 3 MTP Session: It’s very interesting to see that this individual had an MEQ-30 score of 80/150 on methylphenidate and more robust FC change on MTP than in PSIL1. What do you make of this? I think this should be commented on in the legend of the figure. (Also, please double check that this wasn’t human error, i.e. mislabeling PSIL1 for MTP and vice versa for example...)
- Extended Data Figure 7: Label the outlier a different color so it’s easier for the reader to observe the point you removed in your revised Pearson correlation (top left panel).
- Supplementary Table 1: Might be helpful to know the approximate weight of each participant to get a sense for what the mg/kg dosing of their psilocybin experience was.
- Supplementary Table 1: Might be helpful to add timestamps for when the resting-state PSIL data was collected for each participant relative to when the oral psilocybin was consumed.
- Supplementary Figure 2: Please add SD or SEM to these average points. It’s puzzling to me that P6 had a lower heart rate while on psilocybin than baseline... How is this possible?
- Supplement (Resting-state functional MRI processing and surface projection): Was MCFLIRT used for motion correction? If so, linear spatial registration was used and not affine? Please clarify.
- Supplement (Task fMRI Analyses): What was the assumed HRF, i.e. was it the canonical double gamma HRF? Please provide details of the task GLM -- How many regressors? What was the shape/duration of the regressors? Is each trial represented by an impulse or epoch?

Author Rebuttals to First Revision:

Referees' comments:

Referee #1 (Remarks to the Author):

The authors have been very responsive to my concerns. The new analyses and various clarifications have improved the manuscript and bolstered confidence in the results considerably. In my view, the revised manuscript is suitable for publication.

Thank you very much for the positive feedback.

Minor point, but in Extended Data Figure 7, the authors have removed an outlier to emphasize the correlation between delta NGSC and MEQ scores (top left panel). I would like to see them treat the outlier in the lower right panel (delta respiratory rate) the same way (it appears to be close to ~2-3 SD from mean).

The outlier in the upper left panel (NGSC) is 2.3 SD below the mean. The datapoint in question in the lower right panel (respiratory rate) is 1.9 SD below the mean, which would not be considered an outlier. Thus, it would not be appropriate to remove. Note, however, with it removed, the correlation for the lower right remains not significant (Pearson correlation of respiratory rate vs MEQ: $r = 0.53$, $P = 0.11$).

Also, I am unable to access / review the code, I get a 404 page not found error when using the link provided by the authors.

This was a mistake. The page has now been made accessible to the public.

Referee #1 (Remarks on code availability):

I am unable to access / review the code, I get a 404 page not found error when clicking the link provided by the authors.

This was a mistake. The page has now been made accessible to the public.

Referee #2 (Remarks to the Author):

The authors have addressed all my technical concerns.

Referee #2 (Remarks on code availability):

It says page not found when I try to go to the page.

This was a mistake. The page has now been made accessible to the public.

Referee #3 (Remarks to the Author):

In this revised manuscript, the authors perform additional analyses and address many of the critiques brought up by my fellow reviewers. It is clear they spent time thinking about the critiques and made a good faith effort to address the majority of them. I particularly appreciate that in Figure 1, authors added some analyses to relate individual differences in FC change to mystical experience ratings. However, I still feel the authors have not fully leveraged their methods to understand time course of long-term changes across individuals and how this relates to the acute effects.

Also, the discussion is still quite repetitive. See examples below for illustration of this.

From intro:

In rodent models, transient activation of the 5HT_{2A} receptors by a psychedelic can alter neuronal communication in 5HT_{2A}-rich regions (e.g. the medial frontal lobe) and induce persistent plasticity-related phenomena^{5,8,9}...

Yet, inherent limitations of rodent models, and imperfect homology to the human 5HT_{2A} receptor²⁵, limit the strength of these assertions. Understanding the effects of psychedelics on human brain networks is critical to unlocking their therapeutic mechanisms.

Discussion:

Psychedelics rapidly induce synaptogenesis in the hippocampus and cortex, effects that appear to be necessary for rapid antidepressant-like effects in animal models^{9,24}. However, to understand the underpinnings of psychedelics' unique effects, human studies are needed.

Why restate the same thing twice esp. when it is review of what is known? Also, I hardly think the authors needs to state that human work is needed in this field at all, let alone two times in one manuscript. Maybe instead authors could expand on what they mean in the last sentence: "Novel methods to measure neurotrophic markers in the human brain¹⁰² will provide a critical link between mechanistic observations at the cellular, brain-networks and psychological levels." Or perhaps they can talk more about how their results may be relevant to psychiatric populations.

The intro paragraph describing what is known from rodent models has been shortened so that it is less redundant with discussion points. Specifically, this sentence has been

removed: “Persistent effects of psychedelics observed days to weeks later include increases in the expression of genes that contribute to synaptic plasticity (c-Fos, BDNF, Arc) and neurite and synapse growth, in vitro²¹, and in vivo.”

Finally with regard to “grounding” I think this sentence in the abstract is too much of an overstep: “Performing a perceptual task reduced psilocybin-driven FC changes and desynchronization, suggesting a neurobiological basis for grounding – connecting with physical reality during psychedelic therapy,” given that the inference to grounding is indirect at best, in the actual experiment that they performed.

We now simply state “Performing a perceptual task reduced psilocybin-driven FC changes.” This both avoids over-interpretation of the results and also shortens the summary so that it fits the 230 word limit.

Overall, while the authors have addressed many of the comments and the manuscript is stronger overall, I remain ambivalent about its publication in this particular journal given conceptual advance of the study.

Referee #4 (Remarks to the Author):

Siegel et al. have done a great job responding to most of my comments. I have some additional items for them to consider, which I hope will further strengthen and clarify the manuscript prior to publication.

Comments:

- Main Text (line 79): Timmerman et al. studied DMT (citation 30). While DMT and psilocybin are chemically related, I would not cite this particular study as an example of psilocybin decreasing the power of electrophysiological signals.

Corrected.

- Figure 1 Panel D: There appear to be fewer open circles for the DMN column. I’m assuming that some circles are superimposed and therefore giving the impression of there being fewer?

Yes, there are 3 circles nearly on top of each other (values 0.187, 0.183, 0.22).

- Main Text (line 133): Change average to averaged

Corrected.

- Main Text (line 147): I would specify that you're using MEQ-30 and not MEQ-43. You may also want to present MEQ-30 data as a percentage of maximum possible score instead of using 150.

Indicated that the 30 question version of the MEQ was used in line 139. Percentage was not used, because the scale is not linear.

- Figure 1 Panel F: I'm guessing 1st session and 2nd session refer to PSIL1 and PSIL2 (i.e. one average FC change brain is shown from the main experiment and one average FC change brain from the replication protocol)? It would be nice to know exactly when the resting state scans for each participant were acquired relative to when they consumed the oral psilocybin for both sessions. This might inform why these two participants are having such dramatic FC change differences from the same dosage of psilocybin on different sessions.

For consistency/clarity, we changed labels to PSIL1 and PSIL2 (as used in Extended Data Fig 3 and Supplementary Fig 1). We also added clarification to the Study Design section: "Participants underwent imaging during drug sessions (with MRI starting 1 hour after drug ingestion) ..."

- Main Text (line 152) and Panel G: There are 16 data points here. I expected to see 10 data points: 6 participants who completed PSIL1 + 4 participants who completed PSIL2. Then a correlation between MEQ30 at the end of each session with whole brain FC change normed. How did you end up with 16 data points? Consider color coding each data point by participant contribution to the model. It would also be nice to know where in the time course following oral psilocybin consumption these points are coming from (possibly as a separate panel?).

The 16 data points are 10 psilocybin (6 from initial protocol, 4 replication), and 6 methylphenidate sessions. To explain this more clearly, the figure caption now reads “g) Relationship between whole brain FC change and mystical experience rating is plotted for all drug sessions (psilocybin and methylphenidate).”

- Figure 1 Panel G: Consider adding a line at 60% maximum possible score on the MEQ-30 (90 out of 150) to show threshold for “complete” mystical experience. Looks like 2.8 FC change is needed for achieving a “complete” mystical experience.
- Main Text (Figure 1 F-H): The MEQ-30 has 4 dimensions: (1) mystical, (2) transcendence of time and space, (3) positive mood, and (4) ineffability. While you’ve presented the results for MEQ-30 total score, I think it would be informative to show the results of this analysis for the 4 dimensions of the MEQ-30. Are there certain FC changes that are more or less specific to these 4 dimensions? How do individual differences in FC change map to differences in the phenomenology of the psychedelic experience?

We appreciate the reviewer drawing our attention to these interesting points about the MEQ30. In some literature, the criteria for ‘complete mystical experience’ on the MEQ30 appears to be at least 60% in all 4 subscales (as opposed to 60% of total score), (e.g. Barrett et al., 2015). It is not clear how distinguishing complete mystical experience would fit into the comparison between FC change and total MEQ30 score.

Above is the correlation between the four dimensions of the MEQ30 and FC Change. Transcendence is marginally stronger than other domains. However, in our dataset (16 drug doses across 6 individuals) the four dimensions of the MEQ30 are highly correlated (all with $r > 0.8$ and upper bounds of 95% CI > 0.93), thus differences in between dimensions in the relationships between subjective experience and brain changes are not statistically significant.

- Main Text (lines 157-159): “Acquiring multiple scans at multiple psilocybin sessions, enabled us to determine that the variability in psilocybin’s brain effects was more likely due to differences in subjective experience than noise” – consider rephrasing to -- differences in subjective experience are more likely caused by variability in psilocybin’s brain effects (probably as a function of time since ingestion, but also participant weight/drug metabolism/etc.).

Thank you for this insightful point. The sentence now reads “Acquiring multiple scans at multiple psilocybin sessions, enabled us to determine that the inter-individual variability in psilocybin’s brain effects was more likely related to real differences in drug effects than measurement error (likelihood ratio test of participant-specific response to psilocybin, $P = 0.00245$)^{51,52}.” This way, it honestly describes the results of the specific statistical test (likelihood ratio test) which tests the null hypothesis that differences in participant-specific FC change on psilocybin are a consequence of random variance or measurement error, without implying that directionality/causality between brain and subject experience.

As a side note, it is surely true that psilocybin’s brain effects vary as a function of weight and drug metabolism. However, it is interesting to note that an implication of the likelihood ratio test is that there is an effect of FC change that is specific to a given psilocybin session and therefore fairly robust to time/drug metabolism. The low p-value on the likelihood ratio test indicates that there were inter-individual differences that were preserved across a participants drug scans (regardless of whether those scans happened at the beginning or end of their time in the scanner).

- Main Text (Figure 2, panel b): Why does one connection change appear to be occurring outside the brain?

That is a connection with the cerebellum (which is not shown in the traslucent ‘glass-brain’ overlay). The caption for Fig. 2b now includes the sentence “Cerebellar connections are included although the structure is not shown.”

- Main Text (lines 268-270): “and the distribution of these effects appear to match a map of 5HT-2A receptor density” – this is not convincing with just a qualitative assessment of Figure 3 panels D-F. I think it would be helpful to quantitate this claim. LSD binds more tightly to the 5HT-2A receptor than psilocybin. Does LSD-induced desynchronization overly more closely with 5HT-2A expression than psilocybin?

We have now quantified the association of both local desynchronization maps (psilocybin and LSD) with 5HT-2A receptor binding. The main text now states “Global and local desynchronization replicated in an LSD dataset (Fig. 3e)⁵⁴ and the distribution of these effects correlated with serotonin 2A receptor (5HT-2A) density (Fig. 3f; Pearson correlation NGSC_{PSIL} to Cimi-36 binding, $r = 0.39$, $P = 1.9 \times 10^{-13}$; NGSC_{LSD} to Cimi-36 binding, $r = 0.32$, $P = 4.5 \times 10^{-9}$)^{35,66}.”

- Main Text (line 410): Change prepare to prepared.

Corrected.

- Main Text (line 432): I would be more specific and state that you assessed subjective effects using the total score of the MEQ-30. You may also want to provide some context as to why MEQ-30 was used and not 5D-ASC or 11D-ASC for measuring acute subjective effects.

In the “Assessing subjective experience” section we now more clearly state: “Subjective experience was assessed for drug sessions using the 30-item mystical experience questionnaire⁵¹ (MEQ30; see Supplementary methods). The MEQ30 is designed to capture the core domains of the subjective effects of psychedelics (as compared to the altered states of consciousness rating scales which more broadly assess effects of psychoactive drugs¹²⁴)”

In supplement: “MEQ has shown an association to persistent effects and to symptom reduction across various conditions, including cancer-related distress, substance use disorder, and depressive disorders (Ko et al., 2022). The 5D-ASC is considerably longer (96 vs 30 questions) and captures a wider variety of altered states of consciousness. Oceanic Boundlessness (OBN), one of five subdimensions of 5D-ASC, which includes subdimensions of experience of unity, spiritual experience, blissful state, and insightfulness, correlates strongly with MEQ (Liechti et al., 2017). OBN was shown to therapeutic efficacy of psilocybin in treatment-resistant depression (Roseman et al., 2018). We chose the MEQ30 because it has greater specificity to the effects of psychedelics and reduced participant burden.”

- Main Text (line 433): Where are the results for personality change? Was there any change pre vs post psilocybin in these subjects?

We collected personality assessments using the International Personality Item Pool (IPIP) Five-Factor Model. However we did not end up acquiring personality assessments on the majority of participants after drug doses. Measurement personality after drug doses was abandoned after the 3rd participant in order to shorten the protocol and because it was clear that we would not be powered to detect personality change with $n = 7$. We now state “Changes in pulse rate and respiratory rate during psilocybin and placebo were later added as secondary outcome measures and personality change was abandoned because it was clear that we would not be powered to detect personality change.”

- Extended Data Figure 1 Panel D: Was the number of resting-state scans acquired for PSIL1 session for this participant 2 x 15 minutes (panel label) or 6 x 15 minutes (supplementary table 1)?

We have removed “2 x 15” from Extended Data Figure 1 Panel D, since more than two scans were often acquired.

- Extended Data Figure 1 Panel E: Might be worthwhile to also show possible individual differences in psychedelic experience of the 7 participants instead of just group differences between psilocybin and methylphenidate.

MEQ30 score for every subject for every drug session is shown Extended Data Figure 3.

- Extended Data Figure 3 – Participant 3 MTP Session: It’s very interesting to see that this individual had an MEQ-30 score of 80/150 on methylphenidate and more robust FC change on MTP than in PSIL1. What do you make of this? I think this should be commented on in the legend of the figure. (Also, please double check that this wasn’t human error, i.e. mislabeling PSIL1 for MTP and vice versa for example...)

We state in the supplement text, “6/7 participants correctly guessed which dose was psilocybin. Curiously, P3, the only participant who guessed incorrectly, showed a smaller FC change during psilocybin than any other participant (with the exception of P5 replication dose, in which the participant vomited 30 minutes after swallowing the capsule).” It is notable that this participant did report substantial psychoactive effects (MEQ30 = 38.5) on the second drug (which they incorrectly guessed to be MTP). It is also notable that the pattern of FC change after the MTP dose was primarily in somato-motor (consistent with MTP changes seen in others participants/datasets (Extended Data Fig. 3).

An experienced clinical trial coordinator prepared the pills based on RedCap-generated pseudorandomization. I (Josh Siegel) was blinded and was one of the facilitators for P3. I suspected, for a number of reasons including those listed above, that the participant may have guessed wrong. Only later after data collection was complete and analysis was under way did I see that my suspicion was correct.

- Extended Data Figure 7: Label the outlier a different color so it's easier for the reader to observe the point you removed in your revised Pearson correlation (top left panel).

For simplicity, we labeled the outlier with a gray arrow and indicated in the caption “In the case of Δ NGSC, statistics are reported before and after the removal of an outlier point (> 2 SD lower than mean, indicated by the gray arrow).”

- Supplementary Table 1: Might be helpful to know the approximate weight of each participant to get a sense for what the mg/kg dosing of their psilocybin experience was.

Great point. This has been added to Supplementary Table 1.

- Supplementary Table 1: Might be helpful to add timestamps for when the resting-state PSIL data was collected for each participant relative to when the oral psilocybin was consumed.

This information is provided in Extended Data Figure 1, Panel b.

- Supplementary Figure 2: Please add SD or SEM to these average points. It's puzzling to me that P6 had a lower heart rate while on psilocybin than baseline... How is this possible?

It is indeed unusual. But HR was measured by two separate systems (the in-scanner physio recording system, and a separate BP/HR monitoring system that was required as part of safety protocol) and the same result was seen for P6. On the dosing day, P6 pre-dose HR

was 93 and 88BPM. Post-dose HR's (measure independently of the data acquired from the in-scanner system) were 88 (30 min), 72 (60m), 74 (90m), 76 (120m), 89 (240m), 89 (360m).

Here are more complete distributions for every condition shown in Supplementary Figure 2.

- Supplement (Resting-state functional MRI processing and surface projection): Was MCFLIRT used for motion correction? If so, linear spatial registration was used and not affine? Please clarify.

Thank you for pointing out this ambiguity. We now more clearly state: “Preprocessing of fMRI data was done using an inhouse MRI processing pipeline and included: 1) removal of thermal noise using NORDIC (a local PCA approach in which temporal components of an fMRI signal that are indistinguishable from Gaussian noise are eliminated)¹⁰⁹; 2) compensation for asynchronous slice acquisition using sinc interpolation; 3) compute linear spatial registration of all volumes within a run (using 4dfp tools); 4) elimination of odd/even slice intensity differences resulting from interleaved acquisition (debanding); 5) compute affine spatial registration across fMRI runs; 6) compute an run volume mean (of all low-noise volumes); 7) computation of field distortion on the basis of a spin echo field maps using FSL top-up¹³⁰; and 8) gain field correction using FSL fast¹³¹ (computed on the run volume mean);

Resampling in MNI152 2 mm³ atlas space was accomplished for all echoes in one step combining (i) motion correction of volumes within a visit; (ii) distortion correction; (iii) gain field correction; (iv) affine spatial registration of average volumes across visits; and (v) non-linear MNI152 atlas registration via the fsl fnirt¹³²..”

- Supplement (Task fMRI Analyses): What was the assumed HRF, i.e. was it the canonical double gamma HRF? Please provide details of the task GLM -- How many regressors? What was the shape/duration of the regressors? Is each trial represented by an impulse or epoch?

In the Supplementary section “Task fMRI Analyses”, we now more clearly state “A generalized linear model was computed in two different ways: 1) vertexwise GLM, using an assumed hemodynamic response function to visualize the magnitude of task-evoked responses, 2) parcel-wise GLM, using a finite impulse response model to model evoked response for 11 TRs (19.37 seconds) after each trial. For (1), the canonical HRF from SPM (double gamma function) was used. For (2), a set of a priori regions of interest (ROIs) relevant to the task were selected from the Gordon-Laumann parcellation. These included: left/right calcarine sulcus (V1), left/right auditory cortex (A1), left language (Wernicke’s area), left hand knob, left angular gyrus, and right angular gyrus (default mode). Trial conditions (congruent, incongruent; button press, no button press) were collapsed to model a main effect of task. For both (1) and (2), additional regressors for button response, demean and detrend terms, and 6 movement parameters were added to generate a general linear model (GLM). This GLM was solved to estimate beta weights separately for each task visit.”